# VARIATIONAL IMAGE COMPRESSION WITH A SCALE HYPERPRIOR

**Johannes Ballé**[*]
jballe@google.com

**David Minnen**[*]
dminnen@google.com

**Saurabh Singh**[*]
saurabhsingh@google.com

**Sung Jin Hwang**[*]
sjhwang@google.com

**Nick Johnston**[*]
nickj@google.com

[*]Google
Mountain View, CA 94043, USA

## ABSTRACT

We describe an end-to-end trainable model for image compression based on variational autoencoders. The model incorporates a hyperprior to effectively capture spatial dependencies in the latent representation. This hyperprior relates to side information, a concept universal to virtually all modern image codecs, but largely unexplored in image compression using artificial neural networks (ANNs). Unlike existing autoencoder compression methods, our model trains a complex prior jointly with the underlying autoencoder. We demonstrate that this model leads to state-of-the-art image compression when measuring visual quality using the popular MS-SSIM index, and yields rate–distortion performance surpassing published ANN-based methods when evaluated using a more traditional metric based on squared error (PSNR). Furthermore, we provide a qualitative comparison of models trained for different distortion metrics.

## 1 INTRODUCTION

Recent machine learning methods for lossy image compression have generated significant interest in both the machine learning and image processing communities (e.g., Ballé et al., 2017; Theis et al., 2017; Toderici et al., 2017; Rippel and Bourdev, 2017). Like all lossy compression methods, they operate on a simple principle: an image, typically modeled as a vector of pixel intensities $\boldsymbol{x}$, is quantized, reducing the amount of information required to store or transmit it, but introducing error at the same time. Typically, it is not the pixel intensitites that are quantized directly. Rather, an alternative (latent) representation of the image is found, a vector in some other space $\boldsymbol{y}$, and quantization takes place in this representation, yielding a discrete-valued vector $\hat{\boldsymbol{y}}$. Because it is discrete, it can be losslessly compressed using *entropy coding* methods, such as arithmetic coding (Rissanen and Langdon, 1981), to create a bitstream which is sent over the channel. Entropy coding relies on a prior probability model of the quantized representation, which is known to both encoder and decoder (the *entropy model*).

In the class of ANN-based methods for image compression mentioned above, the entropy model used to compress the latent representation is typically represented as a joint, or even fully factorized, distribution $p_{\hat{\boldsymbol{y}}}(\hat{\boldsymbol{y}})$. Note that we need to distinguish between the actual marginal distribution of the latent representation $m(\hat{\boldsymbol{y}})$, and the entropy model $p_{\hat{\boldsymbol{y}}}(\hat{\boldsymbol{y}})$. While the entropy model is typically assumed to have some parametric form, with parameters fitted to the data, the marginal is an unknown distribution arising from both the distribution of images that are encoded, and the method which is used to infer the alternative representation $\boldsymbol{y}$. The smallest average code length an encoder–decoder pair can achieve, using $p_{\hat{\boldsymbol{y}}}$ as their shared entropy model, is given by the Shannon cross entropy between the two distributions:

$$R = \mathbb{E}_{\hat{\boldsymbol{y}} \sim m}[-\log_2 p_{\hat{\boldsymbol{y}}}(\hat{\boldsymbol{y}})]. \tag{1}$$

Note that this entropy is minimized if the model distribution is identical to the marginal. This implies that, for instance, using a fully factorized entropy model, when statistical dependencies exist in the actual distribution of the latent representation, will lead to suboptimal compression performance.

One way conventional compression methods increase their compression performance is by transmitting *side information*: additional bits of information sent from the encoder to the decoder, which

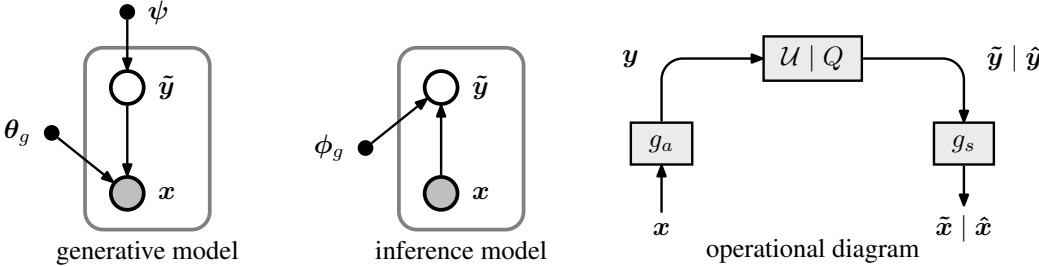

Figure 1: Left: representation of a transform coding model as a generative Bayesian model, and a corresponding variational inference model. Nodes represent random variables or parameters, and arrows indicate conditional dependence between them. Right: diagram showing the operational structure of the compression model. Arrows indicate the flow of data, and boxes represent transformations of the data. Boxes labeled $\mathcal{U} \mid Q$ represent either addition of uniform noise applied during training (producing vectors labeled with a tilde), or quantization and arithmetic coding/decoding during testing (producing vectors labeled with a hat).

signal modifications to the entropy model intended to reduce the mismatch. This is feasible because the marginal for a particular image typically varies significantly from the marginal for the ensemble of images the compression model was designed for. In this scheme, the hope is that the amount of side information sent is smaller, on average, than the reduction of code length achieved in eq. (1) by matching $p_{\hat{y}}$ more closely to the marginal for a particular image. For instance, JPEG (1992) models images as independent fixed-size blocks of $8 \times 8$ pixels. However, some image structure, such as large homogeneous regions, can be more efficiently represented by considering larger blocks at a time. For this reason, more recent methods such as HEVC (2013) partition an image into variable-size blocks, convey the partition structure to the decoder as side information, and then compress the block representations using that partitioning. That is, the entropy model for JPEG is always factorized into groups of 64 elements, whereas the factorization is variable for HEVC. The HEVC decoder needs to decode the side information first, so that it can use the correct entropy model to decode the block representations. Since the encoder is free to select a partitioning that optimizes the entropy model for each image, this scheme can be used to achieve more efficient compression.

In conventional compression methods, the structure of this side information is hand-designed. In contrast, the model we present in this paper essentially learns a latent representation of the entropy model, in the same way that the underlying compression model learns a representation of the image. Because our model is optimized end-to-end, it minimizes the total expected code length by learning to balance the amount of side information with the expected improvement of the entropy model. This is done by expressing the problem formally in terms of variational autoencoders (VAEs), probabilistic generative models augmented with approximate inference models (Kingma and Welling, 2014). Ballé et al. (2017) and Theis et al. (2017) previously noted that some autoencoder-based compression methods are formally equivalent to VAEs, where the entropy model, as described above, corresponds to the prior on the latent representation. Here, we use this formalism to show that side information can be viewed as a prior on the parameters of the entropy model, making them *hyperpriors* of the latent representation.

Specifically, we extend the model presented in Ballé et al. (2017), which has a fully factorized prior, with a hyperprior that captures the fact that spatially neighboring elements of the latent representation tend to vary together in their scales. We demonstrate that the extended model leads to state-of-the-art image compression performance when measured using the MS-SSIM quality index (Wang, Simoncelli, et al., 2003). Furthermore, it provides significantly better rate–distortion performance compared to other ANN-based methods when measured using peak signal-to-noise ratio (PSNR), a metric based on mean squared error. Finally, we present a qualitative comparison of the effects of training the same model class using different distortion losses.

## 2 COMPRESSION WITH VARIATIONAL MODELS

In the *transform coding* approach to image compression (Goyal, 2001), the encoder transforms the image vector $x$ using a parametric analysis transform $g_a(x; \phi_g)$ into a latent representation $y$, which

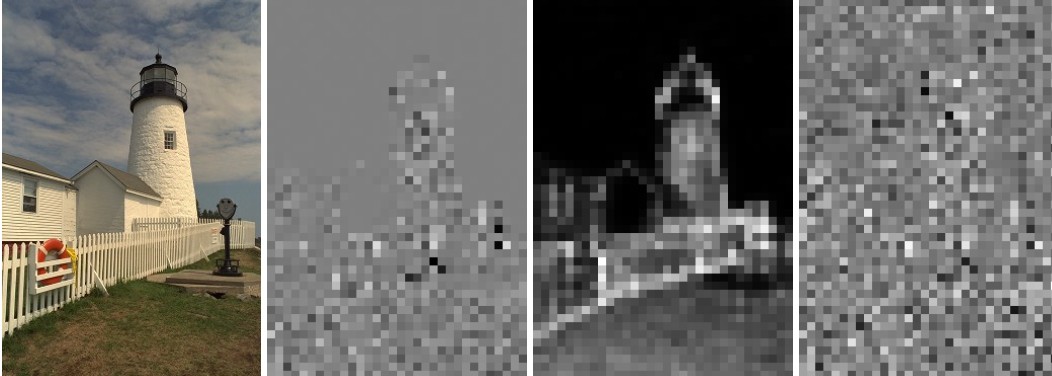

Figure 2: Left: an image from the Kodak dataset. Middle left: visualization of a subset of the latent representation $y$ of that image, learned by our factorized-prior model. Note that there is clearly visible structure around edges and textured regions, indicating that a dependency structure exists in the marginal which is not represented in the factorized prior. Middle right: standard deviations $\hat{\sigma}$ of the latents as predicted by the model augmented with a hyperprior. Right: latents $y$ divided elementwise by their standard deviation. Note how this reduces the apparent structure, indicating that the structure is captured by the new prior.

is then quantized to form $\hat{y}$. Because $\hat{y}$ is discrete-valued, it can be losslessly compressed using entropy coding techniques such as arithmetic coding (Rissanen and Langdon, 1981) and transmitted as a sequence of bits. On the other side, the decoder recovers $\hat{y}$ from the compressed signal, and subjects it to a parametric synthesis transform $g_s(\hat{y}; \boldsymbol{\theta}_g)$ to recover the reconstructed image $\hat{x}$. In the context of this paper, we think of the transforms $g_a$ and $g_s$ as generic parameterized functions, such as artificial neural networks (ANNs), rather than linear transforms as in traditional compression methods. The parameters $\boldsymbol{\theta}_g$ and $\boldsymbol{\phi}_g$ then encapsulate the weights of the neurons, etc. (refer to section 4 for details).

The quantization introduces error, which is tolerated in the context of lossy compression, giving rise to a *rate–distortion optimization* problem. Rate is the expected code length (bit rate) of the compressed representation: assuming the entropy coding technique is operating efficiently, this can again be written as a cross entropy:

$$R = \mathbb{E}_{\boldsymbol{x} \sim p_{\boldsymbol{x}}} \big[ - \log_2 p_{\hat{y}} \big( Q(g_a(\boldsymbol{x}; \boldsymbol{\phi}_g)) \big) \big], \tag{2}$$

where $Q$ represents the quantization function, and $p_{\hat{y}}$ is the entropy model, as described in the introduction. In this context, the marginal distribution of the latent representation arises from the (unknown) image distribution $p_{\boldsymbol{x}}$ and the properties of the analysis transform. Distortion is the expected difference between the reconstruction $\hat{x}$ and the original image $\boldsymbol{x}$, as measured by a norm or perceptual metric. The coarseness of the quantization, or alternatively, the warping of the representation implied by the analysis and synthesis transforms, affects both rate and distortion, leading to a trade-off, where a higher rate allows for a lower distortion, and vice versa. Various compression methods can be viewed as minimizing a weighted sum of these two quantities. Formally, we can parameterize the problem by $\lambda$, a weight on the distortion term. Different applications require different trade-offs, and hence different values of $\lambda$.

In order to be able to use gradient descent methods to optimize the performance of the model over the parameters of the transforms ($\boldsymbol{\theta}_g$ and $\boldsymbol{\phi}_g$), the problem needs to be relaxed, because due to the quantization, gradients with respect to $\boldsymbol{\phi}_g$ are zero almost everywhere. Approximations that have been investigated include substituting the gradient of the quantizer (Theis et al., 2017), and substituting additive uniform noise for the quantizer itself during training (Ballé et al., 2016b). Here, we follow the latter method, which switches back to actual quantization when applying the model as a compression method. We denote the quantities derived from this approximation with a tilde, as opposed to a hat; for instance, $\tilde{y}$ represents the "noisy" representation, and $\hat{y}$ the quantized representation.

The optimization problem can be formally represented as a variational autoencoder (Kingma and Welling, 2014); that is, a probabilistic generative model of the image combined with an approximate

inference model (figure 1). The synthesis transform is linked to the generative model ("generating" a reconstructed image from the latent representation), and the analysis transform to the inference model ("inferring" the latent representation from the source image). In variational inference, the goal is to approximate the true posterior $p_{\tilde{\boldsymbol{y}}|\boldsymbol{x}}(\tilde{\boldsymbol{y}} \mid \boldsymbol{x})$, which is assumed intractable, with a parametric variational density $q(\tilde{\boldsymbol{y}} \mid \boldsymbol{x})$ by minimizing the expectation of their Kullback–Leibler (KL) divergence over the data distribution $p_{\boldsymbol{x}}$:

$$\mathbb{E}_{\boldsymbol{x} \sim p_{\boldsymbol{x}}} D_{\mathrm{KL}}[q \parallel p_{\tilde{\boldsymbol{y}}|\boldsymbol{x}}] = \mathbb{E}_{\boldsymbol{x} \sim p_{\boldsymbol{x}}} \mathbb{E}_{\tilde{\boldsymbol{y}} \sim q} \Big[ \underbrace{\log q(\tilde{\boldsymbol{y}} \mid \boldsymbol{x})}_{0} \underbrace{- \log p_{\boldsymbol{x}|\tilde{\boldsymbol{y}}}(\boldsymbol{x} \mid \tilde{\boldsymbol{y}})}_{\text{weighted distortion}} \underbrace{- \log p_{\tilde{\boldsymbol{y}}}(\tilde{\boldsymbol{y}})}_{\text{rate}} \Big] + \text{const.} \quad (3)$$

By matching the parametric density functions to the transform coding framework, we can appreciate that the minimization of the KL divergence is equivalent to optimizing the compression model for rate–distortion performance. We have indicated here that the first term will evaluate to zero, and the second and third term correspond to the weighted distortion and the bit rate, respectively. Let's take a closer look at each of the terms.

First, the mechanism of "inference" is computing the the analysis transform of the image and adding uniform noise (as a stand-in for quantization), thus:

$$q(\tilde{\boldsymbol{y}} \mid \boldsymbol{x}, \boldsymbol{\phi}_g) \quad = \quad \prod_i \mathcal{U}\big(\tilde{y}_i \mid y_i - \tfrac{1}{2}, y_i + \tfrac{1}{2}\big) \quad (4)$$
$$\text{with } \boldsymbol{y} = g_a(\boldsymbol{x}; \boldsymbol{\phi}_g),$$

where $\mathcal{U}$ denotes a uniform distribution centered on $y_i$. Since the width of the uniform distribution is constant (equal to one), the first term in the KL divergence technically evaluates to zero, and can be dropped from the loss function.

For the sake of argument, assume for a moment that the likelihood is given by:

$$p_{\boldsymbol{x}|\tilde{\boldsymbol{y}}}(\boldsymbol{x} \mid \tilde{\boldsymbol{y}}, \boldsymbol{\theta}_g) \quad = \quad \mathcal{N}\big(\boldsymbol{x} \mid \tilde{\boldsymbol{x}}, (2\lambda)^{-1}\mathbf{1}\big) \quad (5)$$
$$\text{with } \tilde{\boldsymbol{x}} = g_s(\tilde{\boldsymbol{y}}; \boldsymbol{\theta}_g).$$

The log likelihood then works out to be the squared difference between $\boldsymbol{x}$ and $\tilde{\boldsymbol{x}}$, the output of the synthesis transform, weighted by $\lambda$. Minimizing the second term in the KL divergence is thus equivalent to minimizing the expected distortion of the reconstructed image. A squared error loss is equivalent to choosing a Gaussian distribution; other distortion metrics may have an equivalent distribution, but this is not guaranteed, as not all metrics necessarily correspond to a normalized density function.

The third term in the KL divergence is easily seen to be identical to the cross entropy between the marginal $m(\tilde{\boldsymbol{y}}) = \mathbb{E}_{\boldsymbol{x} \sim p_{\boldsymbol{x}}} q(\tilde{\boldsymbol{y}} \mid \boldsymbol{x})$ and the prior $p_{\tilde{\boldsymbol{y}}}(\tilde{\boldsymbol{y}})$. It reflects the cost of encoding $\tilde{\boldsymbol{y}}$, as produced by the inference model, assuming $p_{\tilde{\boldsymbol{y}}}$ as the entropy model. Note that this term represents a differential cross entropy, as opposed to a Shannon (discrete) entropy as in eq. (2), due to the uniform noise approximation. Under the given assumptions, however, they are close approximations of each other (for an empirical evaluation of this approximation, see Ballé et al., 2017). Similarly to Ballé et al. (2017), we model the prior using a non-parametric, fully factorized density model (refer to appendix 6.1 for details):

$$p_{\tilde{\boldsymbol{y}}|\boldsymbol{\psi}}(\tilde{\boldsymbol{y}} \mid \boldsymbol{\psi}) \quad = \quad \prod_i \Big( p_{y_i|\boldsymbol{\psi}^{(i)}}\big(\boldsymbol{\psi}^{(i)}\big) * \mathcal{U}\big(-\tfrac{1}{2}, \tfrac{1}{2}\big) \Big)(\tilde{y}_i) \quad (6)$$

where the vectors $\boldsymbol{\psi}^{(i)}$ encapsulate the parameters of each univariate distribution $p_{y_i|\boldsymbol{\psi}^{(i)}}$ (we denote all these parameters collectively as $\boldsymbol{\psi}$). Note that we convolve each non-parametric density with a standard uniform density. This is to enable a better match of the prior to the marginal – for more details, see appendix 6.2. As a shorthand, we refer to this case as the *factorized-prior* model.

The center panel in figure 2 visualizes a subset of the quantized responses ($\hat{\boldsymbol{y}}$) of a compression model trained in this way. Visually, it is clear that the choice of a factorized distribution is a stark simplification: non-zero responses are highly clustered in areas of high contrast; i.e., around edges, or within textured regions. This implies a probabilistic coupling between the responses, which is not represented in models with a fully factorized prior. We would expect a better model fit and, consequently, a better compression performance, if the model captured these dependencies. Introducing a hyperprior is an elegant way of achieving this.

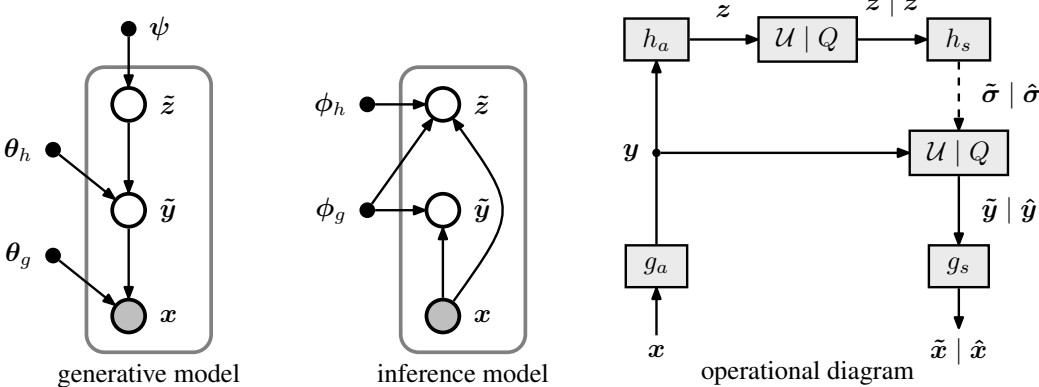

Figure 3: As in figure 1, but extended with a hyperprior.

## 3 INTRODUCTION OF A SCALE HYPERPRIOR

As evident from the center panel of figure 2, there are significant spatial dependencies among the elements of $\hat{\boldsymbol{y}}$. Notably, their scales appear coupled spatially. A standard way to model dependencies between a set of target variables is to introduce latent variables conditioned on which the target variables are assumed to be independent (Bishop, 1999). We introduce an additional set of random variables $\tilde{\boldsymbol{z}}$ to capture the spatial dependencies and propose to extend the model as follows (figure 3).

Each element $\tilde{y}_i$ is now modeled as a zero-mean Gaussian with its own standard deviation $\sigma_i$, where the standard deviations are predicted by applying a parametric transform $h_s$ to $\tilde{\boldsymbol{z}}$ (as above, we convolve each Gaussian density with a standard uniform; see appendix 6.2):

$$p_{\tilde{\boldsymbol{y}}|\tilde{\boldsymbol{z}}}(\tilde{\boldsymbol{y}} \mid \tilde{\boldsymbol{z}}, \boldsymbol{\theta}_h) = \prod_i \Big( \mathcal{N}\big(0, \tilde{\sigma}_i^2\big) * \mathcal{U}\big(-\tfrac{1}{2}, \tfrac{1}{2}\big) \Big)(\tilde{y}_i) \qquad (7)$$

$$\text{with } \tilde{\boldsymbol{\sigma}} = h_s(\tilde{\boldsymbol{z}}; \boldsymbol{\theta}_h).$$

We extend the inference model simply by stacking another parametric transform $h_a$ on top of $\boldsymbol{y}$, effectively creating a single joint factorized variational posterior, as follows:

$$q(\tilde{\boldsymbol{y}}, \tilde{\boldsymbol{z}} \mid \boldsymbol{x}, \boldsymbol{\phi}_g, \boldsymbol{\phi}_h) = \prod_i \mathcal{U}\big(\tilde{y}_i \mid y_i - \tfrac{1}{2}, y_i + \tfrac{1}{2}\big) \cdot \prod_j \mathcal{U}\big(\tilde{z}_j \mid z_j - \tfrac{1}{2}, z_j + \tfrac{1}{2}\big) \qquad (8)$$

$$\text{with } \boldsymbol{y} = g_a(\boldsymbol{x}; \boldsymbol{\phi}_g), \boldsymbol{z} = h_a(\boldsymbol{y}; \boldsymbol{\phi}_h).$$

This follows the intuition that the responses $\boldsymbol{y}$ should be sufficient to estimate the spatial distribution of the standard deviations. As we have no prior beliefs about the hyperprior, we now model $\tilde{\boldsymbol{z}}$ using the non-parametric, fully factorized density model previously used for $\tilde{\boldsymbol{y}}$ (appendix 6.1):

$$p_{\tilde{\boldsymbol{z}}|\boldsymbol{\psi}}(\tilde{\boldsymbol{z}} \mid \boldsymbol{\psi}) = \prod_i \Big( p_{z_i|\boldsymbol{\psi}^{(i)}}\big(\boldsymbol{\psi}^{(i)}\big) * \mathcal{U}\big(-\tfrac{1}{2}, \tfrac{1}{2}\big) \Big)(\tilde{z}_i), \qquad (9)$$

where the vectors $\boldsymbol{\psi}^{(i)}$ encapsulate the parameters of each univariate distribution $p_{z_i|\boldsymbol{\psi}^{(i)}}$ (collectively denoted as $\boldsymbol{\psi}$). The loss function of this model works out to be:

$$\mathbb{E}_{\boldsymbol{x} \sim p_{\boldsymbol{x}}} D_{\mathrm{KL}}[q \parallel p_{\tilde{\boldsymbol{y}}, \tilde{\boldsymbol{z}}|\boldsymbol{x}}] = \mathbb{E}_{\boldsymbol{x} \sim p_{\boldsymbol{x}}} \mathbb{E}_{\tilde{\boldsymbol{y}}, \tilde{\boldsymbol{z}} \sim q} \Big[ \log q(\tilde{\boldsymbol{y}}, \tilde{\boldsymbol{z}} \mid \boldsymbol{x}) - \log p_{\boldsymbol{x}|\tilde{\boldsymbol{y}}}(\boldsymbol{x} \mid \tilde{\boldsymbol{y}})$$

$$- \log p_{\tilde{\boldsymbol{y}}|\tilde{\boldsymbol{z}}}(\tilde{\boldsymbol{y}} \mid \tilde{\boldsymbol{z}}) - \log p_{\tilde{\boldsymbol{z}}}(\tilde{\boldsymbol{z}}) \Big] + \text{const.} \quad (10)$$

Again, the first term is zero, since $q$ is a product of uniform densities of unit width. The second term (the likelihood) encapsulates the distortion, as before. The third and fourth term represent the cross entropies encoding $\tilde{\boldsymbol{y}}$ and $\tilde{\boldsymbol{z}}$, respectively. In analogy to traditional transform coding, the fourth term can be seen as representing side information.

The right-hand panel in figure 3 illustrates how the model is used as a compression method. The encoder subjects the input image $\boldsymbol{x}$ to $g_a$, yielding the responses $\boldsymbol{y}$ with spatially varying standard deviations. The responses are fed into $h_a$, summarizing the distribution of standard deviations

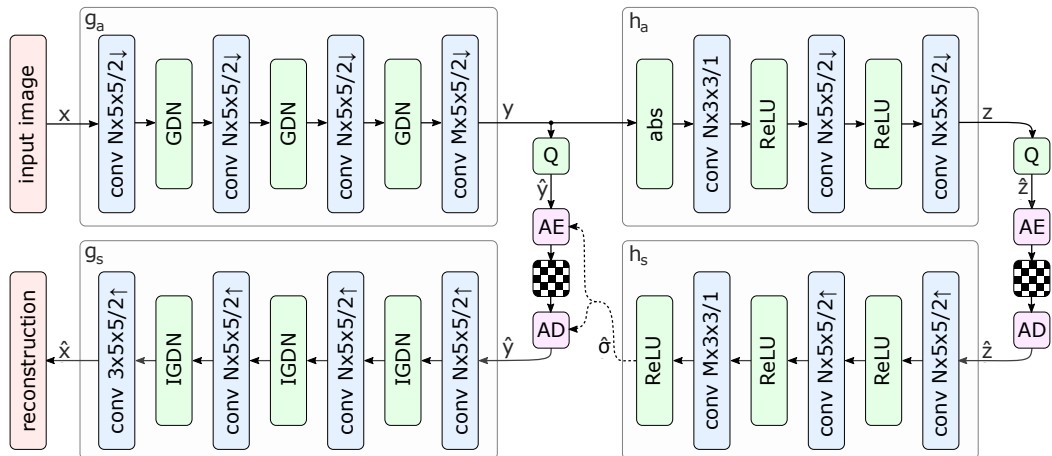

Figure 4: Network architecture of the hyperprior model. The left side shows an image autoencoder architecture, the right side corresponds to the autoencoder implementing the hyperprior. The factorized-prior model uses the identical architecture for the analysis and synthesis transforms $g_a$ and $g_s$. Q represents quantization, and AE, AD represent arithmetic encoder and arithmetic decoder, respectively. Convolution parameters are denoted as: number of filters $\times$ kernel support height $\times$ kernel support width / down- or upsampling stride, where $\uparrow$ indicates upsampling and $\downarrow$ downsampling. $N$ and $M$ were chosen dependent on $\lambda$, with $N = 128$ and $M = 192$ for the 5 lower values, and $N = 192$ and $M = 320$ for the 3 higher values.

in $\boldsymbol{z}$. $\boldsymbol{z}$ is then quantized, compressed, and transmitted as side information. The encoder then uses the quantized vector $\hat{\boldsymbol{z}}$ to estimate $\hat{\boldsymbol{\sigma}}$, the spatial distribution of standard deviations, and uses it to compress and transmit the quantized image representation $\hat{\boldsymbol{y}}$. The decoder first recovers $\hat{\boldsymbol{z}}$ from the compressed signal. It then uses $h_s$ to obtain $\hat{\boldsymbol{\sigma}}$, which provides it with the correct probability estimates to successfully recover $\hat{\boldsymbol{y}}$ as well. It then feeds $\hat{\boldsymbol{y}}$ into $g_s$ to obtain the reconstructed image.

## 4 EXPERIMENTS

To compare the compression performance of our proposed models, we conducted a number of experiments using the Tensorflow framework.

### 4.1 EXPERIMENTAL SETUP

We set up the transforms $g_a$, $g_s$, $h_a$, and $h_s$ as alternating compositions of linear and nonlinear functions, as is common in artificial neural networks (figure 4). Specifically, $g_a$ and $g_s$ are composed of convolutions and GDN/IGDN nonlinearities, which implement local divisive normalization, a type of transformation that has been shown to be particularly suitable for density modeling and compression of images (Ballé et al., 2016a; Ballé et al., 2017).[1] $h_a$ and $h_s$ are composed of convolutions and rectifiers (rectified linear units). To make the hyperprior model and the factorized-prior model comparable, we chose identical architectures for $g_a$ and $g_s$, as shown in figure 4.

To maintain translation invariance across the model, all elements of $\boldsymbol{z}$ with the same channel index are assumed to follow the same univariate distribution. This allows the model to be used with arbitrary image sizes. Arithmetic coding is implemented using a simple non-adaptive binary arithmetic coder. Each element of $\hat{\boldsymbol{y}}$ and $\hat{\boldsymbol{z}}$ is independently converted to its representation as a binary integer and arithmetically encoded from the most significant to the least significant bit. Since the spatial distribution of standard deviations ($\hat{\boldsymbol{\sigma}}$) is known to the decoder by the time decoding of $\hat{\boldsymbol{y}}$ is attempted, the arithmetic coder does not need to handle conditional dependencies. It also does not need to be separately trained, since the binary probabilities needed for encoding are a direct function of the probability mass functions of $\hat{\boldsymbol{y}}$ and $\hat{\boldsymbol{z}}$, and the probability mass functions in turn are direct functions

---

[1]We used the Tensorflow implementation of GDN/IGDN, as documented at https://www.tensorflow.org/versions/master/api_docs/python/tf/contrib/layers/GDN.

of their "noisy" counterparts $\tilde{\boldsymbol{y}}, \tilde{\boldsymbol{z}}$ by design (Ballé et al., 2017). This is particulary important for $\hat{\boldsymbol{y}}$. Since the prior is conditioned on $\hat{\boldsymbol{\sigma}}$, the probability mass functions $p_{\hat{y}_i}$ need to be constructed "on the fly" during decoding of an image:

$$p_{\hat{y}_i}(\hat{y}_i \mid \hat{\sigma}_i) = p_{\tilde{y}_i}(\hat{y}_i \mid \hat{\sigma}_i) = \left( \mathcal{N}(0, \hat{\sigma}_i) * \mathcal{U}\left(-\tfrac{1}{2}, \tfrac{1}{2}\right) \right)(\hat{y}_i) = \int_{\hat{y}_i - 1/2}^{\hat{y}_i + 1/2} \mathcal{N}(y \mid 0, \hat{\sigma}_i) \, \mathrm{d}y, \quad (11)$$

which can be evaluated in closed form.

The models were trained on a body of color JPEG images with heights/widths between 3000 and 5000 pixels, comprising approximately 1 million images scraped from the world wide web. Images with excessive saturation were screened out to reduce the number of non-photographic images. To reduce existing compression artifacts, the images were further downsampled by a randomized factor, such that the minimum of their height and width equaled between 640 and 1200 pixels. Then, randomly placed $256 \times 256$ pixel crops of these downsampled images were extracted. Minibatches of 8 of these crops at a time were used to perform stochastic gradient descent using the Adam algorithm (Kingma and Ba, 2015) with a learning rate of $10^{-4}$. Common machine learning techniques such as batch normalization or learning rate decay were found to have no beneficial effect (this may be due to the local normalization properties of GDN, which contain global normalization as a special case).

With this setup, we trained a total of 32 separate models: half of the models with a hyperprior and half without; half of the models with mean squared error as the distortion metric (as described in the previous section), and half on the MS-SSIM distortion index (Wang, Simoncelli, et al., 2003); finally, each of these combinations with 8 different values of $\lambda$ in order to cover a range of rate–distortion tradeoffs.

## 4.2    Experimental results

We evaluate the compression performance of all models on the publicly available Kodak dataset (Eastman Kodak, 1993). Summarized rate–distortion curves are shown in figure 5. Results for individual images, as well as summarized comparisons to a wider range of existing methods are provided in appendices 6.5 and 6.7. We quantify image distortion using peak signal-to-noise ratio (PSNR) and MS-SSIM. Each curve represents the rate–distortion tradeoffs for a given set of models, across different values of $\lambda$. Since MS-SSIM yields values between 0 (worst) and 1 (best), and most of the compared methods achieve values well above 0.9, we converted the quantity to decibels in order to improve legibility.

Interestingly, but maybe not surprisingly, results differ substantially depending on which distortion metric is used in the loss function during training. When measuring distortion in PSNR (figure 5, top), both our models perform poorly if they have been optimized for MS-SSIM. However, when optimized for squared error, the model with the factorized prior outperforms existing conventional codecs such as JPEG, as well as other ANN-based methods which have been trained for squared error (Theis et al., 2017; Ballé et al., 2017). Note that other published ANN-based methods not shown here underperform compared to the ones that are shown, or have not made their data available to us. Our factorized prior model does not outperform BPG (Bellard, 2014), an encapsulation of HEVC (2013) targeted at still image compression. When training our hyperprior model for squared error, we get close to BPG performance, with better results at higher bit rates than lower ones, but still substantially outperforming all published ANN-based methods.

When measuring distortion using MS-SSIM (figure 5, bottom), conventional codecs such as JPEG and BPG end up at the lower end of the performance ranking. This is not surprising, since these methods have been optimized for squared error (with hand-selected constraints intended to ensure that squared error optimization doesn't go against visual quality). To the best of our knowledge, the state of the art for compression performance in terms of MS-SSIM is Rippel and Bourdev (2017). Surprisingly, it is matched (with better performance at high bit rates, and slightly worse performance at low bit rates) by our factorized prior model, even though their model is conceptually much more complex (due to its multiscale architecture, GAN loss, and context-adaptive entropy model). The hyperprior model adds further gains across all rate–distortion tradeoffs, consistently surpassing the state of the art.

With the results differing so heavily depending on which training loss is used, one has to wonder if there are any qualitative differences in the image reconstructions. When comparing images com-

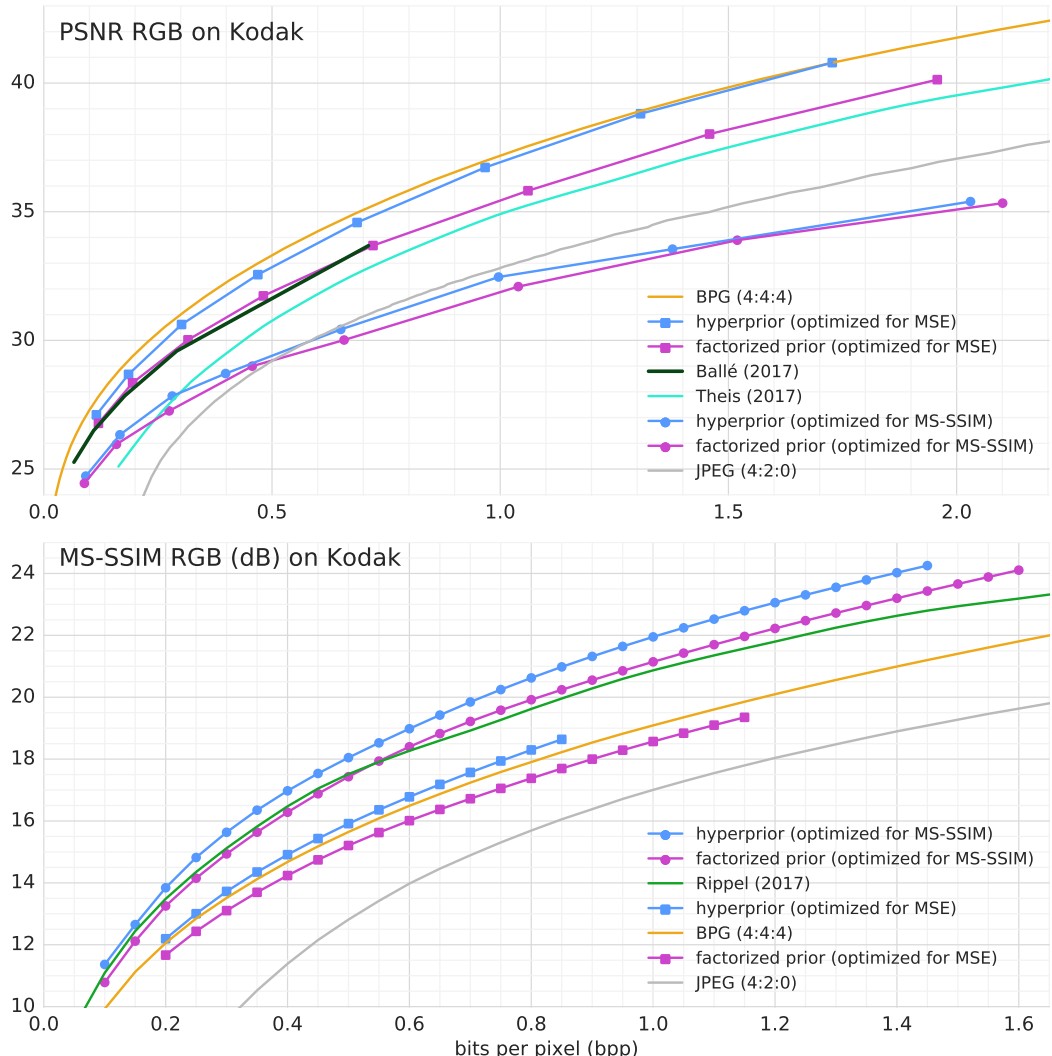

Figure 5: Rate–distortion curves aggregated over the Kodak dataset. The top plot shows peak signal-to-noise ratios as a function of bit rate ($10 \log_{10} \frac{255^2}{d}$, with $d$ representing mean squared error), the bottom plot shows MS-SSIM values converted to decibels ($-10 \log_{10}(1 - d)$, where $d$ is the MS-SSIM value in the range between zero and one). We observe that matching the training loss to the metric used for evaluation is crucial to optimize performance. Our hyperprior model trained on squared error outperforms all other ANN-based methods in terms of PSNR, and approximates HEVC performance. In terms of MS-SSIM, the hyperprior model consistently outperforms conventional codecs as well as Rippel and Bourdev (2017), the current state-of-the-art model for that metric. Note that the PSNR plot aggregates curves over equal values of $\lambda$, and the MS-SSIM plot aggregates over equal rates (with interpolation), in order to provide a fair comparison to both state-of-the-art methods. Refer to figures 11 and 12 in the appendix for full-page RD curves that include a wider range of compression methods.

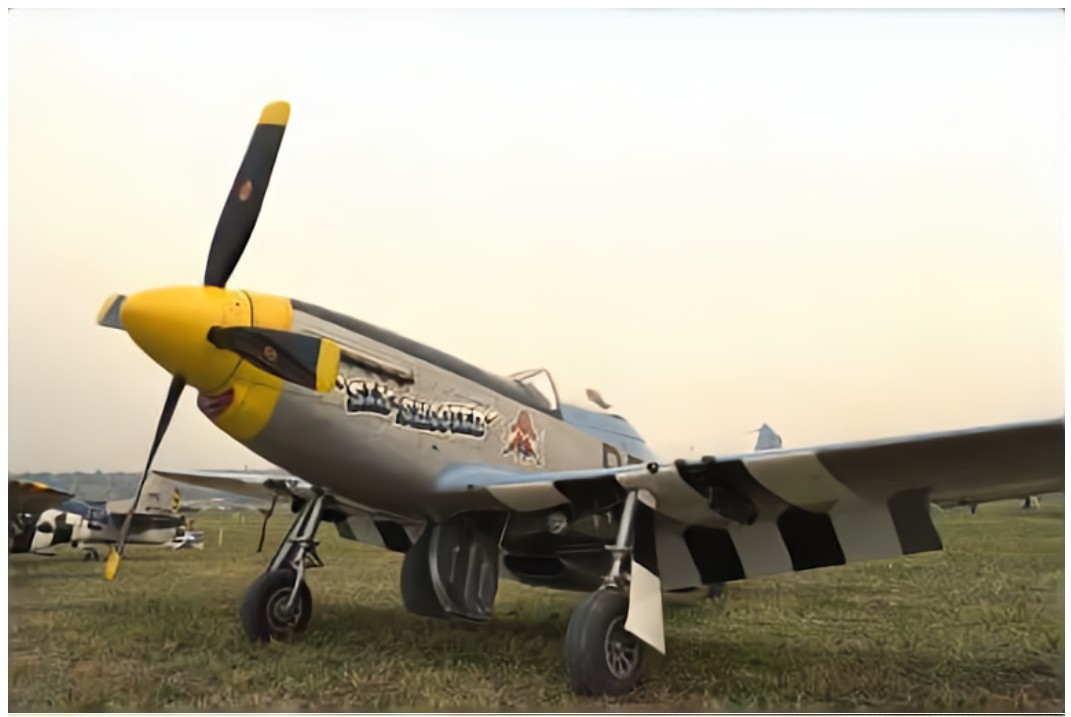

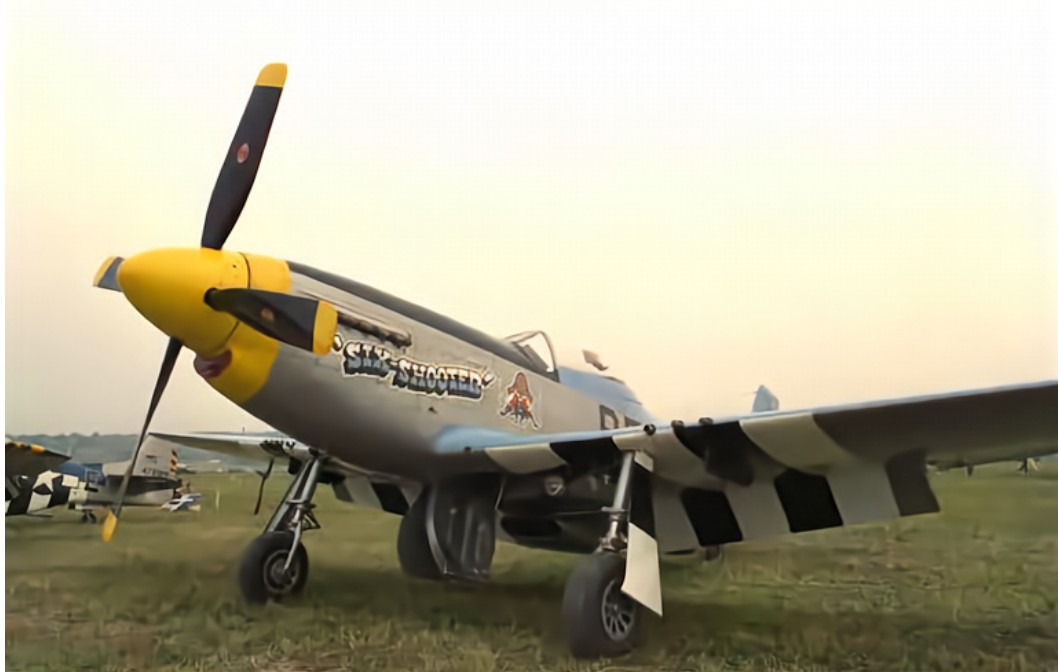

Figure 6: The visual artifacts generated at low bit rates depend on the training loss. The top figure (0.1864 bpp, PSNR=27.99, MS-SSIM=0.9803) was generated by the hyperprior model using an MS-SSIM loss, while the bottom figure (0.1932 bpp, PSNR=32.26, MS-SSIM=0.9713) was trained using squared loss.

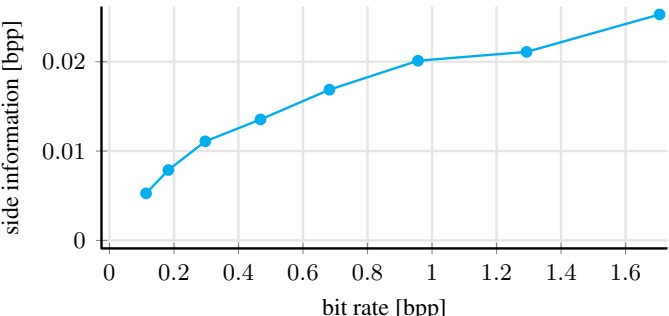

Figure 7: Amount of side information (encoding $\hat{z}$) as a function of total bit rate (encoding $\hat{y}$ and $\hat{z}$), for the hyperprior model optimized for squared error, averaged over the Kodak set, and normalized per pixel. Only a small fraction of the total bit rate is used for encoding $\hat{z}$.

pressed to similar bit rates by models optimized with an MS-SSIM distortion loss compared to a squared loss, we find that the overall fidelity in terms of how much detail is preserved appears similar. However, the spatial distribution of detail changes substantially. MS-SSIM, like its predecessor SSIM (Wang, Bovik, et al., 2004), is a metric designed to model human visual contrast perception. Compared to squared loss, it has the effect of attenuating the error in image regions with high contrast, and boosting the error in regions with low contrast, because the human visibility threshold varies with local contrast. This behavior yields good results for images containing textures with different local contrast (refer to examples provided in appendix 6.7). However, more frequently than expected, it can also produce results inconsistent with human expectations: for the image we show in figure 6, the compression model trained for MS-SSIM assigns more detail to the grass (low contrast), and removes detail from the text on the side of the airplane (high contrast). Because semantic relevance is often assigned to high-contrast areas (such as text, or salient objects), the squared-error optimized models produce subjectively better reconstructions in these cases. It is important to note that neither distortion metric is sophisticated enough to capture image semantics, which makes the choice of distortion loss a difficult one.

Prior work on ANN-based image compression has shown that extending the transform coding concept from linear to nonlinear transforms fundamentally improves the qualitative nature of compression artifacts (Ballé et al., 2017). It appears that nonlinear transforms with higher computational capacity adapt better to the statistics of natural images, imitating properties of the data distribution better than linear transforms. When comparing image reconstructions visually between models with or without the hyperprior, we find no changes to the qualitative nature of the artifacts. Rather, the hyperprior model simply tends to produce image reconstructions with improved detail and a lower bit rate than the corresponding model with a factorized prior.

Figure 7 shows how much of the total bit rate the hyperprior model uses as side information. The amount of side information grows with the total bit rate, but stays far below 0.1 bpp, even for the highest total bit rates. Still, the resulting improvement of the prior enables the performance gains over the factorized-prior model shown in figure 5. Note that the architecture of the models does not explicitly constrain the bit rates in any way. The illustrated trade-off in allocating bits for encoding $\hat{z}$ vs. $\hat{y}$ is simply the result of optimizing the loss function given in eq. (10).

## 5 DISCUSSION

We implement a variational image compression model, conceptually identical to the model presented by Ballé et al. (2017), and augment it with a more powerful entropy model by introducing a hyperprior on the local scale parameters of the latent representation. The hyperprior is trained end-to-end with the rest of the model.

Like all recent image compression methods based on ANNs, our method can be directly optimized for distortion losses that are more complex than pixel-wise losses such as mean squared error. As one of the first studies in this emerging field, we examine the effect of optimizing for one of the most popular perceptual metrics, MS-SSIM, and compare it to optimizing for squared loss. Note that Ballé et al. (2016b) compare models trained for different metrics, but their results are limited by the choice of transforms. Figure 6 demonstrates that the results can show significant variation in terms of visual quality, depending on image content, which implies that unless human rating

experiments are conducted to provide more reliable data, it is wise to compare methods based on more than a single type of metric.

Santurkar et al. (2017) formulate their compression method in a hybrid VAE-GAN framework, adopting a stepwise training scheme where a decoder is first trained using an adversarial loss. It is then fixed, and an encoder is trained to minimize the reconstruction error. Rippel and Bourdev (2017) also employ an adversarial approach, but use a weighted combination of an MS-SSIM and an adversarial loss. Baig and Torresani (2017) propose a compression scheme based on colorization, where color channels are predicted from the the luminance channel by making use of some model specific side information. The luminance channel is compressed using a traditional method. The proposed method exhibits significant color distortions at low bit rates, and is limited by the compression method used for the luminance channel.

An early exploration of hierarchical generative models for compression of small images is found in Gregor et al. (2016). However, the aspect of quantization is not thoroughly considered, and hence, no actual compression method is designed. Theis et al. (2017) approach the problem of generating gradient descent directions for quantization functions by replacing their (unhelpful) gradient with the identity function, and derive a differentiable upper bound for the discrete rate term. Ballé et al. (2016b) instead replace the quantizer with additive uniform noise during training, and the discrete rate term with a differential entropy. While this method doesn't offer a bound for the approximation, it establishes a direct relationship between the discrete and continuous prior distributions $p_{\hat{y}}$ and $p_{\tilde{y}}$, which enables direct evaluation of the discrete prior as a function of the latents $\hat{z}$ as in eq. (11), and hence makes use of a hyperprior feasible in practice. The quality of the approximation is verified empirically by Ballé et al. (2017).

Wainwright and Simoncelli (2000) observe that linear filter responses (i.e., wavelet coefficients obtained by filtering an image) follow heavy-tailed marginal distributions, but can be represented as conditionally Gaussian when groups of neighboring coefficients are linked by a common scale multiplier. That is, the distributions of the filter responses can be modeled as Gaussian scale mixtures. Lyu and Simoncelli (2009) extend this model from spatially localized groups of wavelet coefficients to a global image model. Our model can be seen as a further extension of this, where the filter responses are replaced with responses of a nonlinear transform, and an approximate inference model is added. Theis et al. (2017) directly use Gaussian scale mixtures, but in the form of a fully factorized prior. In the presented form, our variational model is perhaps most closely related to ladder VAEs (Sønderby et al., 2016). However, we choose different parametric forms to accommodate the approximation of the quantization and entropy coding process.

In classical transform coding methods, compression researchers have exploited statistical dependency in the latent variables (e.g., DCT or wavelet coefficients) by carefully hand-engineering entropy codes modeling the dependencies in the quantized regime (Taubman and Marcellin, 2002). This presents a much more difficult engineering problem than relying on a fully factorized entropy model; transitioning to nonlinear transforms whose parameters are determined through training (and thus may be different for each re-training) only complicates the problem. Toderici et al. (2017) model images directly with a binarized latent representation, which technically removes the need for a separate entropy coding step. However, this corresponds to a very inflexible entropy model (a uniform prior on a binary representation, with no trainable parameters). The model apparently compensates for this by using higher capacity transforms (e.g., based on recurrent networks). Johnston et al. (2017) improve the method by designing an adaptive entropy model. However, this entropy model is not included in the rate term while training the transforms, and hence no feedback (in terms of gradients) is returned from the entropy model back to the transforms during training. This breaks the paradigm of end-to-end optimization, and may stand in the way of better compression performance. Similarly, Rippel and Bourdev (2017) use a hand-designed energy function without trainable parameters as the prior for training the autoencoder, and design an adaptive entropy model post hoc. The fact that our factorized prior model matches the performance of their method, when optimized on the same metric, may point towards this disconnect. Ágústsson et al. (2017) extend the fully-factorized prior model by proposing to do vector quantization over small subtensors of the latent representation, which effectively relaxes the factorization. They train their method end-to-end.

All of the models presented here make use of GDN, a type of nonlinearity implementing local normalization. As part of a Gaussianizing transformation, GDN has been shown to be more efficient, in terms of number of parameters, at removing statistical dependencies in image data, than pointwise

nonlinearities (Ballé et al., 2016a). Furthermore, there has been a long history of generative models, starting with independent component analysis (Cardoso, 2003), which can successfully recover factorized representations just by maximizing likelihood assuming a fully factorized prior. Despite these facts, we observe that significant dependencies between neighboring elements remain in the latent representation of our compression models (figure 2), even though we took care not to impose constraints on the transforms which might reduce their capacity to factorize the representation (refer to appendix 6.3 for details). We attribute this to the fact that the rate–distortion loss, unlike a maximum likelihood loss, trades off the rate term against expected distortion. It is easy to see that for increasing values of $\lambda$, the rate term containing the factorized prior becomes less and less important. Hence, it is questionable whether rate–distortion optimality implies full independence of the representation, at least for arbitrary values of $\lambda$.

Regardless of this, the fact that the hyperprior models consistently outperform models with a factorized prior illustrate that it is important for any compression method to reduce mismatch between the prior and the marginal, as in eq. (2). Our model, when trained on the appropriate loss, has the capacity to surpass the state of the art on MS-SSIM, but does not quite reach the performance of a heavily optimized traditional method such as BPG on PSNR (while outperforming all other methods based on ANNs). This discrepancy may indicate that methods based on ANNs have not yet reached the expressive power of traditional methods. As such, the introduction of a hyperprior – or, in traditional terms, side information – is an elegant way of introducing more flexible priors, and a big step in the right direction.

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

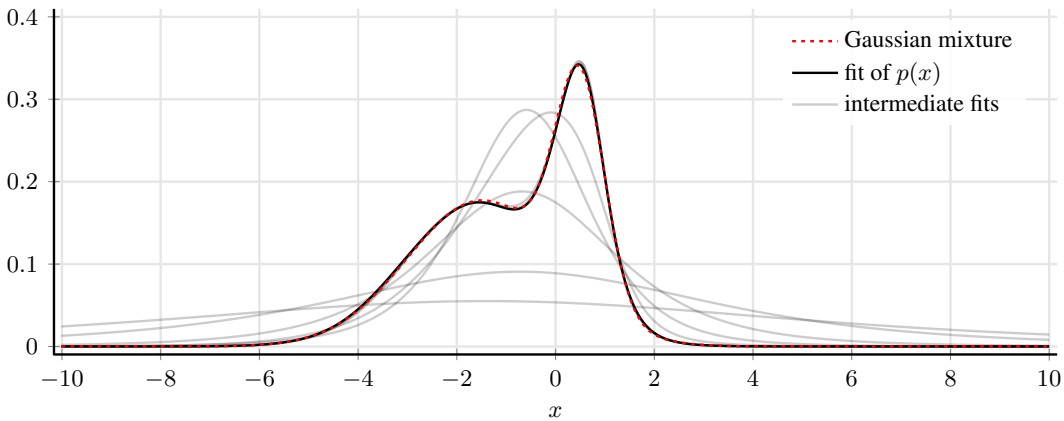

Figure 8: A fit of the non-parametric model $p$ (with $K = 3$) to a Gaussian mixture distribution. Gray plots illustrate convergence of the model. The non-parametric model is able to produce a good fit to the ground truth density.

## 6 APPENDIX

### 6.1 UNIVARIATE NON-PARAMETRIC DENSITY MODEL

Ballé et al. (2017) use a non-parametric piecewise linear density model to represent each factor of the fully factorized prior. By increasing the number of samples per unit interval, it can in principle be used to model any univariate density with arbitrary precision. However, it has two practical problems: The range of values with non-zero probability must be finite and known ahead of time, and its implementation is non-trivial with existing automatic differentiation frameworks, both due to numerical issues with normalizing the density and the fact that it typically relies on discrete operations such as array indexing. For the compression models presented in this paper, we instead use the following model based on the cumulative.

We define a density $p : \mathbb{R} \to \mathbb{R}^+$ using its cumulative $c : \mathbb{R} \to [0, 1]$ by satisfying the following constraints:

$$c(-\infty) = 0; \quad c(\infty) = 1; \quad p(x) = \frac{\partial c(x)}{\partial x} \geq 0 \tag{12}$$

Note that the monotonicity constraint of the cumulative is established by requiring the density function $p$ to be non-negative. Suppose the cumulative is a composition of functions. Then the density can be written using the chain rule of calculus:

$$c = f_K \circ f_{K-1} \cdots f_1 \tag{13}$$
$$p = f'_K \cdot f'_{K-1} \cdots f'_1 \tag{14}$$

where we write the derivative of $f_k$ as $f'_k$. We'll allow the $f_k$ to be vector functions:

$$f_k : \mathbb{R}^{d_k} \to \mathbb{R}^{r_k} \tag{15}$$

In general, the $f'_k$ are Jacobian matrices, and the dots are matrix multiplications. To ensure $p(x)$ is univariate, the domain of $f_1$ and the range of $f_K$ need to be one dimensional ($d_1 = r_K = 1$).

To guarantee that $p(x)$ is a density, we just need $f_K$ to map to the range between 0 and 1, and ensure that $p(x) \geq 0$. To do that, we require all the Jacobian elements to be non-negative. Then the matrix product computing $p(x)$ is non-negative as well, and we have defined a valid density.

An effective choice of $f_k$ is the following (as a shorthand, we define tanh, sigmoid, and softplus as elementwise functions, when applied to vectors or matrices):

$$f_k(\boldsymbol{x}) = g_k\big(\boldsymbol{H}^{(k)}\boldsymbol{x} + \boldsymbol{b}^{(k)}\big) \qquad\qquad 1 \leq k < K \tag{16}$$
$$f_K(\boldsymbol{x}) = \text{sigmoid}\big(\boldsymbol{H}^{(K)}\boldsymbol{x} + \boldsymbol{b}^{(K)}\big) \tag{17}$$

where $\boldsymbol{H}^{(k)}$ are matrices, $\boldsymbol{b}^{(k)}$ are vectors, and $g_k$ are nonlinearities defined as

$$g_k(\boldsymbol{x}) = \boldsymbol{x} + \boldsymbol{a}^{(k)} \odot \tanh(\boldsymbol{x}) \tag{18}$$

where $\boldsymbol{a}^{(k)}$ is a vector and $\odot$ denotes elementwise multiplication. The rationale behind this particular nonlinearity is that it allows to expand or contract the space near $x = 0$. $\boldsymbol{a}^{(k)}$ controls the rate of expansion (when positive) or contraction (when negative). If $\boldsymbol{a}^{(k)}$ were fixed to a positive value, "peaks" in the density would become easier to model than "troughs".

The derivatives work out as follows:

$$f'_k(\boldsymbol{x}) = \operatorname{diag} g'_k\big(\boldsymbol{H}^{(k)}\boldsymbol{x} + \boldsymbol{b}^{(k)}\big) \cdot \boldsymbol{H}^{(k)} \qquad 1 \leq k < K, \text{with} \tag{19}$$

$$g'_k(\boldsymbol{x}) = 1 + \boldsymbol{a}^{(k)} \odot \tanh'(\boldsymbol{x}) \qquad \text{and} \tag{20}$$

$$f'_K(\boldsymbol{x}) = \operatorname{sigmoid}'\big(\boldsymbol{H}^{(K)}\boldsymbol{x} + \boldsymbol{b}^{(K)}\big) \cdot \boldsymbol{H}^{(K)} \tag{21}$$

For the derivatives to be non-negative, we need to constrain $\boldsymbol{H}^{(k)}$ to have all non-negative elements, and the elements of $\boldsymbol{a}^{(k)}$ to be lower bounded by $-1$. This is easily done by reparameterization:

$$\boldsymbol{H}^{(k)} = \operatorname{softplus}\big(\hat{\boldsymbol{H}}^{(k)}\big) \tag{22}$$

$$\boldsymbol{a}^{(k)} = \tanh\big(\hat{\boldsymbol{a}}^{(k)}\big) \tag{23}$$

where the quantities with the hat are the actual parameters. A plot of a fit of this model to a "toy" mixture density is provided in figure 8. As a special case, setting $K = 1$ yields a logistic distribution:

$$c(x) = \operatorname{sigmoid}\big(hx + b\big) \tag{24}$$

$$p(x) = \frac{h}{2} \cdot \frac{1}{1 + \cosh(hx + b)} \tag{25}$$

It may seem odd to define a density function as an explicit derivative; however, in an automatic differentiation framework, this operation is very easy to implement, and the resulting density function is normalized by construction. We have found the model to fit well to arbitrary densities, and perform just as well as the piecewise linear model in the context of compression models. For all experiments in this paper, we used $K = 4$, with the dimensionalities $r_1 = r_2 = r_3 = 3$. Each univariate density model is associated with its own set of parameters $\boldsymbol{a}^{(k)}, \boldsymbol{b}^{(k)}, \boldsymbol{H}^{(k)}$ (which, together, form $\boldsymbol{\psi}^{(i)}$).

## 6.2 Modeling priors with added uniform noise

We model both the prior $p_{\tilde{\boldsymbol{y}}|\tilde{\boldsymbol{z}}}$ and the hyperprior $p_{\tilde{\boldsymbol{z}}}$ using densities that are convolved with a standard uniform density function. This is to ensure that the priors have enough flexibility to match the variational posterior $q$. To see this, consider that in some cases, it is beneficial in terms of rate–distortion performance for the model to "disable" part of the latent representation, leading to a lower effective dimensionality than the model architecture has been set up for. For simplicity of notation, let's assume that the variational posterior and the prior have just one dimension which has collapsed. In this case, $g_a$ converges to always producing a constant value for the corresponding dimensions:

$$y = g_a(\boldsymbol{x}) = c, \text{ independent of } \boldsymbol{x}. \tag{26}$$

When this happens, the marginal distribution of that element during training is a uniform density centered on $c$, due to the added uniform noise, and the variational posterior matches it exactly:

$$m(\tilde{y}) = q(\tilde{y} \mid \boldsymbol{x}) = \mathcal{U}\big(\tilde{y} \mid c - \tfrac{1}{2}, c + \tfrac{1}{2}\big). \tag{27}$$

The cross entropy of this element is given by:

$$\mathbb{E}_{\tilde{y} \sim m}[-\log_2 p_{\tilde{y}}]. \tag{28}$$

This entropy should evaluate to zero bits, as the quantized representation is deterministic (and hence, no information needs to be transmitted). For the cross entropy to evaluate to zero, however, the prior needs to be flexible enough to assume the shape of the posterior – a unit-width uniform density.

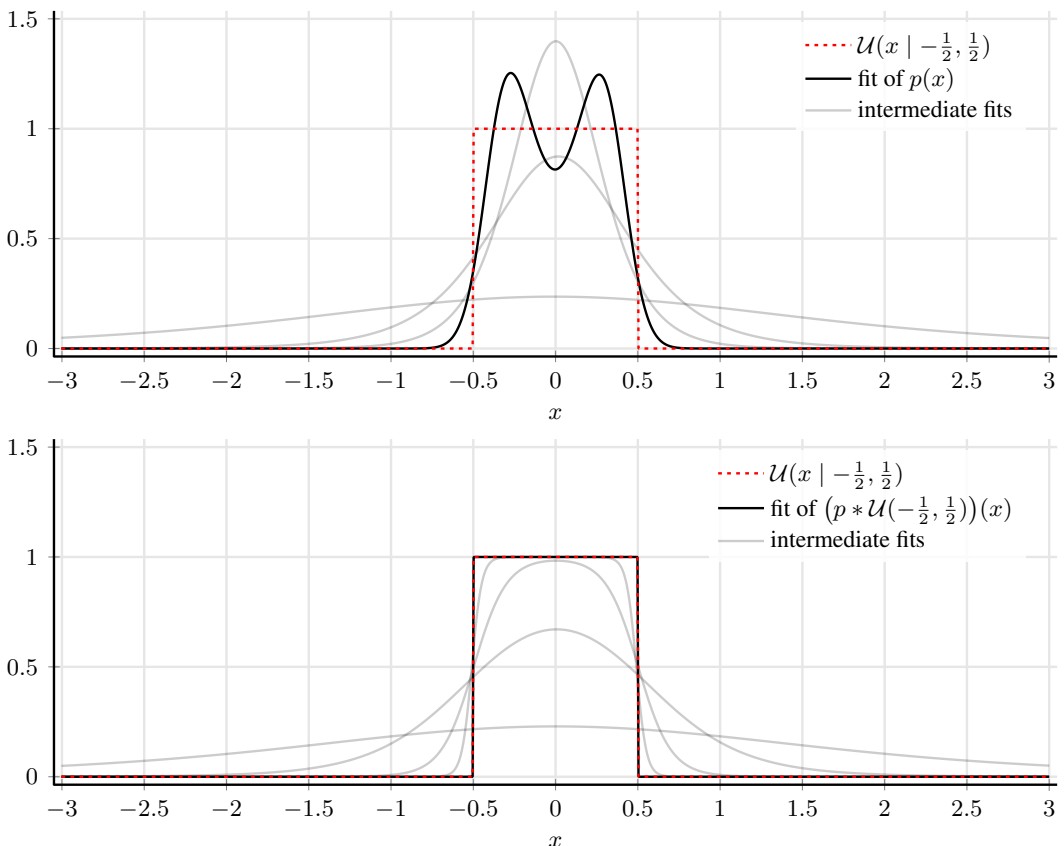

Figure 9: Fitting the density model described in the previous section to a uniform distribution, with and without convolving the model with a uniform density. Gray plots illustrate convergence of the model. While $p$ itself assumes smoothness and thus fails to find an adequate fit to the uniform with its steep edges, the augmented model fits almost perfectly.

Due to its infinitely steep edges, the uniform distribution is a corner case for not only the Gaussian density model, but also the non-parametric model described in appendix 6.1. To fix this, we incorporate the added noise directly into the prior/hyperprior by convolving the underlying density model $p$ with a standard uniform:

$$
\begin{aligned}
p_{\tilde{y}}(\tilde{y}) &= \Big(p * \mathcal{U}\big(-\tfrac{1}{2}, \tfrac{1}{2}\big)\Big)(\tilde{y}) \\
&= \int_{-\infty}^{\infty} p(y)\,\mathcal{U}\big(\tilde{y} - y \mid -\tfrac{1}{2}, \tfrac{1}{2}\big)\,\mathrm{d}y \\
&= \int_{\tilde{y}-\frac{1}{2}}^{\tilde{y}+\frac{1}{2}} p(y)\,\mathrm{d}y \\
&= c\big(\tilde{y} + \tfrac{1}{2}\big) - c\big(\tilde{y} - \tfrac{1}{2}\big),
\end{aligned}
\tag{29}
$$

where $c$ is the cumulative of the underlying density model. Now, whatever the underlying density $p$ is, letting its scale go towards zero makes $p_{\tilde{y}}$ approach a unit-width uniform density. Since the non-parametric model is defined via its cumulative, and the cumulative of a Gaussian is available in most computational frameworks, this solution is easy to implement in practice.

### 6.3 MODEL CAPACITY

Our results seem to indicate that a certain degree of statistical dependency in the latent image representation $\boldsymbol{y}$ is preferred by the rate–distortion objective, and that the hyperprior model performs better by embracing this. However, it is possible that dependencies remain simply because the analysis

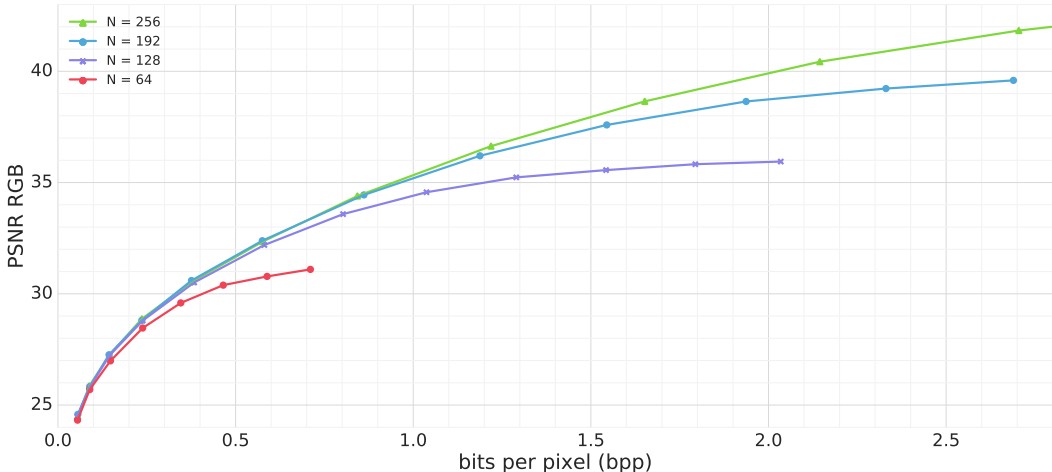

Figure 10: Rate–distortion curves for factorized-prior models only differing in their transform capacity (number of filters at each transform layer $N$). Note that performance gains with increased number of filters stagnates as a $\lambda$-dependent saturation point is reached. For example, moving from 64 to 128 filters makes a significant difference at 0.5 bpp, while moving from 128 to 192 only yields a negligible gain, and there is no benefit in going up to 256.

| CPU | Kodak | | Tecnick | | GPU | Kodak | | Tecnick | |
|-----|-------|------|---------|------|-----|-------|------|---------|------|
| $N$ | encode | decode | encode | decode | $N$ | encode | decode | encode | decode |
| 128 | 331.54 | 334.21 | 1003.73 | 1085.56 | 128 | 242.12 | 338.09 | 491.88 | 799.16 |
| 192 | 551.22 | 576.34 | 1852.10 | 1971.85 | 192 | 310.01 | 385.64 | 630.02 | 1018.35 |

Table 1: Average encoding and decoding runtimes for the proposed model in milliseconds.

and synthesis transforms $g_a$ and $g_s$ do not have enough capacity to factorize the image representation, or because the training algorithm did not succeed in finding the global optimum. Although it is impossible to fully control for this, we attempted to minimize the chances that capacity limitations in the transforms lead to the wrong conclusions, by carefully selecting the number of filters across layers of the transforms (as given by $N$ and $M$ in figure 4).

We established in previous experiments that, for a given $\lambda$, there exist a certain number of filters per layer at which performance saturates, and no gains can be achieved by further increasing it (figure 10; note that for these experiments, we set $N = M$). The optimal number of filters increases with $\lambda$, indicating that models with higher bit rates require higher transform capacities. Based on these previous experiments, we attempted to choose values close to the point of saturation, or a little higher, in order to control for capacity limitations while minimizing training time. Additionally, we found that allowing a somewhat wider bottleneck $M > N$ helps to achieve comparable performance with overall lower $N$, and we used this when choosing the model architectures.

## 6.4 COMPUTATIONAL COMPLEXITY

Table 1 lists encoding and decoding times of our method for a Python and TensorFlow implementation, for CPU as well as GPU and different number of filters per layer ($N$), averaged over the Kodak and Tecnick datasets. Note that no performance optimization was attempted. In particular, we did not optimize the metaparameter choices (number of filters, layers, etc.) for computational complexity. Rather, we chose the number of filters high enough to rule out bottlenecks in the transforms, as described in the previous section. Only the arithmetic coding was implemented as a customized operator in C++. Thus, these measurements represent proof that the method is feasible, but their utility for meaningful comparisons with other methods is limited. The average increase in runtime for the hyperprior model compared to the factorized-prior model was between 20% and 50%.

## 6.5 PERFORMANCE COMPARISONS FOR THE KODAK IMAGE SET

The plots in figures 11 and 12 show the same results as figure 5, but provide comparisons to a wider array of compression methods. Note that the method to aggregate rate–distortion points across images differs between the PSNR and MS-SSIM plots: in the latter, we interpolate the RD curves for each image (as shown in appendix 6.7) using cubic splines at a predefined set of bit rates, and then average across equal bit rates. In the former, no interpolation was used, averaging rate and distortion measurements across equal values of $\lambda$. As noted by Ballé et al. (2017), directly comparing RD curves with different methods of aggregation can give misleading results. Because of this, we match our aggregation method to the data available for the current state of the art ($\lambda$-aggregation for HEVC and PSNR, and rate aggregation for Rippel and Bourdev (2017) and MS-SSIM). Ultimately, a comparison based on individual images, as provided in section 6.7, should be considered more reliable; however, data on individual images for Rippel and Bourdev (2017) has not been available.

## 6.6 PERFORMANCE COMPARISONS FOR THE TECNICK IMAGE SET

For the sake of completeness, the plots in figures 13 and 14 show results over the Tecnick dataset (Asuni and Giachetti, 2014). Rate and distortion measurements were averaged across equal values of $\lambda$ for both PSNR and MS-SSIM plots.

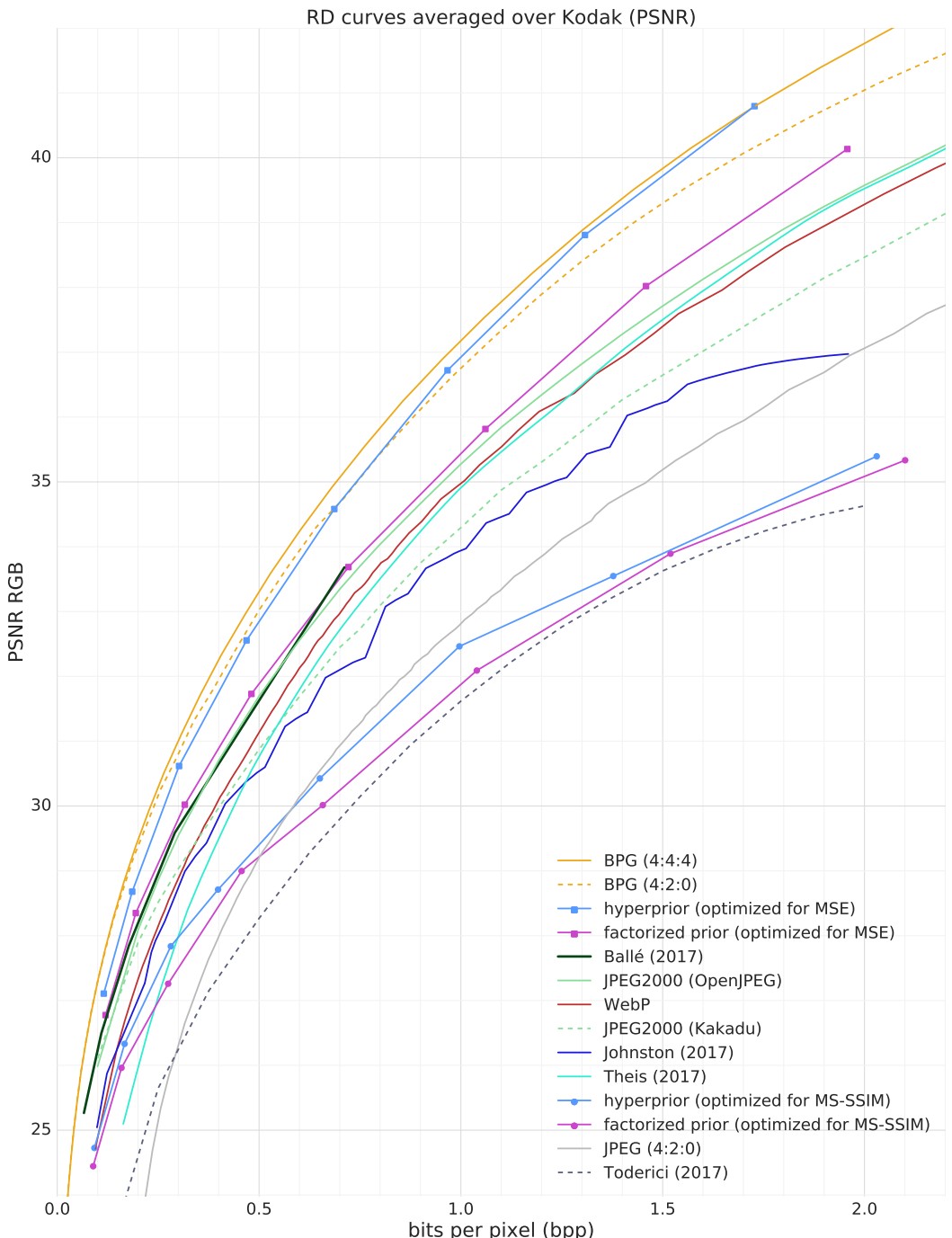

Figure 11: Rate–distortion curves for PSNR covering a wide range of conventional and ANN-based compression methods. We see that our hyperprior model (blue squares) outperforms most conventional codecs (JPEG, JPEG 2000, and WebP) as well as all ANN-based methods by a wide margin.

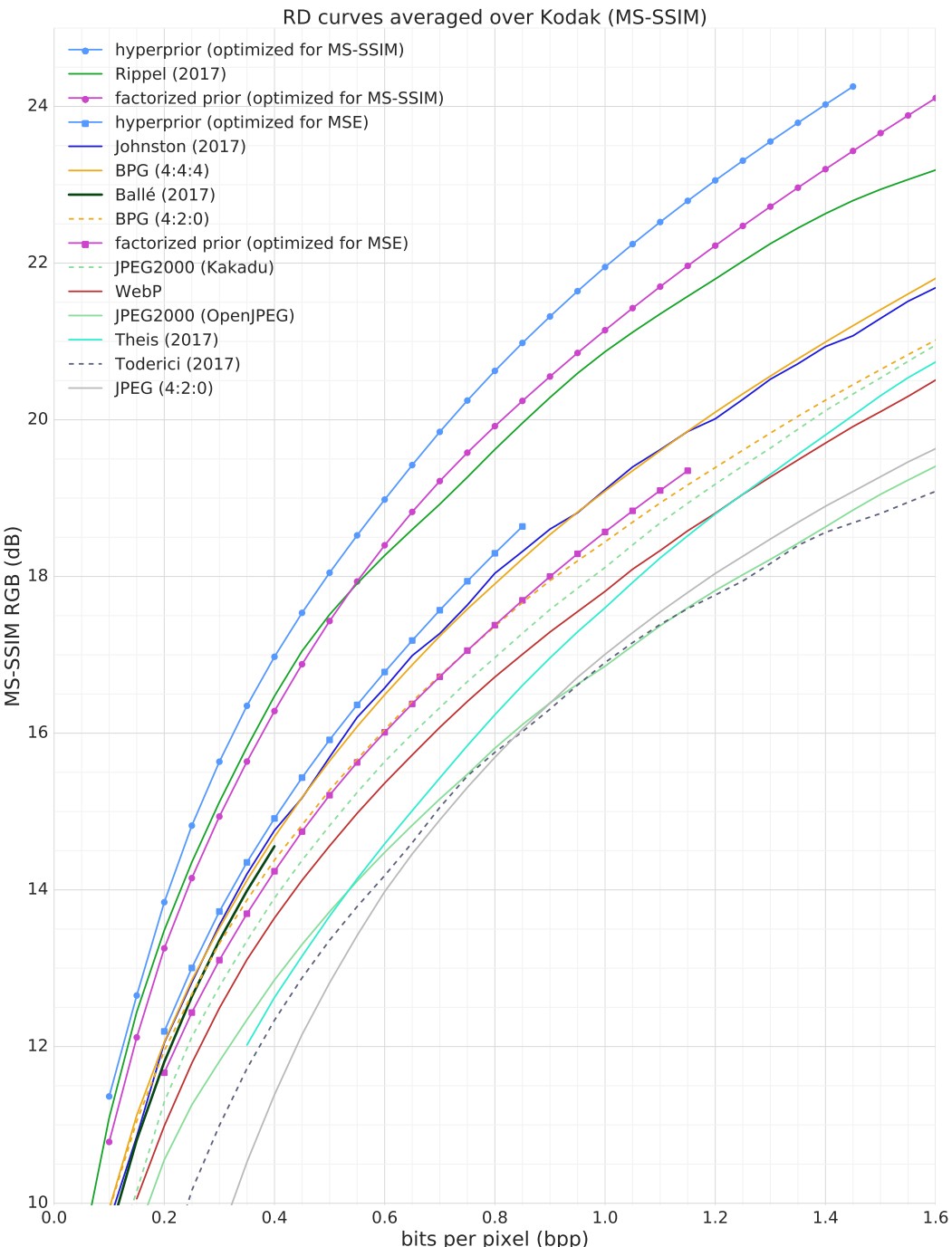

Figure 12: Rate–distortion curves for MS-SSIM covering a wide range of conventional and ANN-based compression methods. When trained on MS-SSIM, our hyperprior model outperforms Rippel and Bourdev (2017), the current state of the art, consistently across all bit rates. Note that even when trained using squared loss, our hyperprior model (blue squares) yields higher MS-SSIM scores than all of the conventional methods.

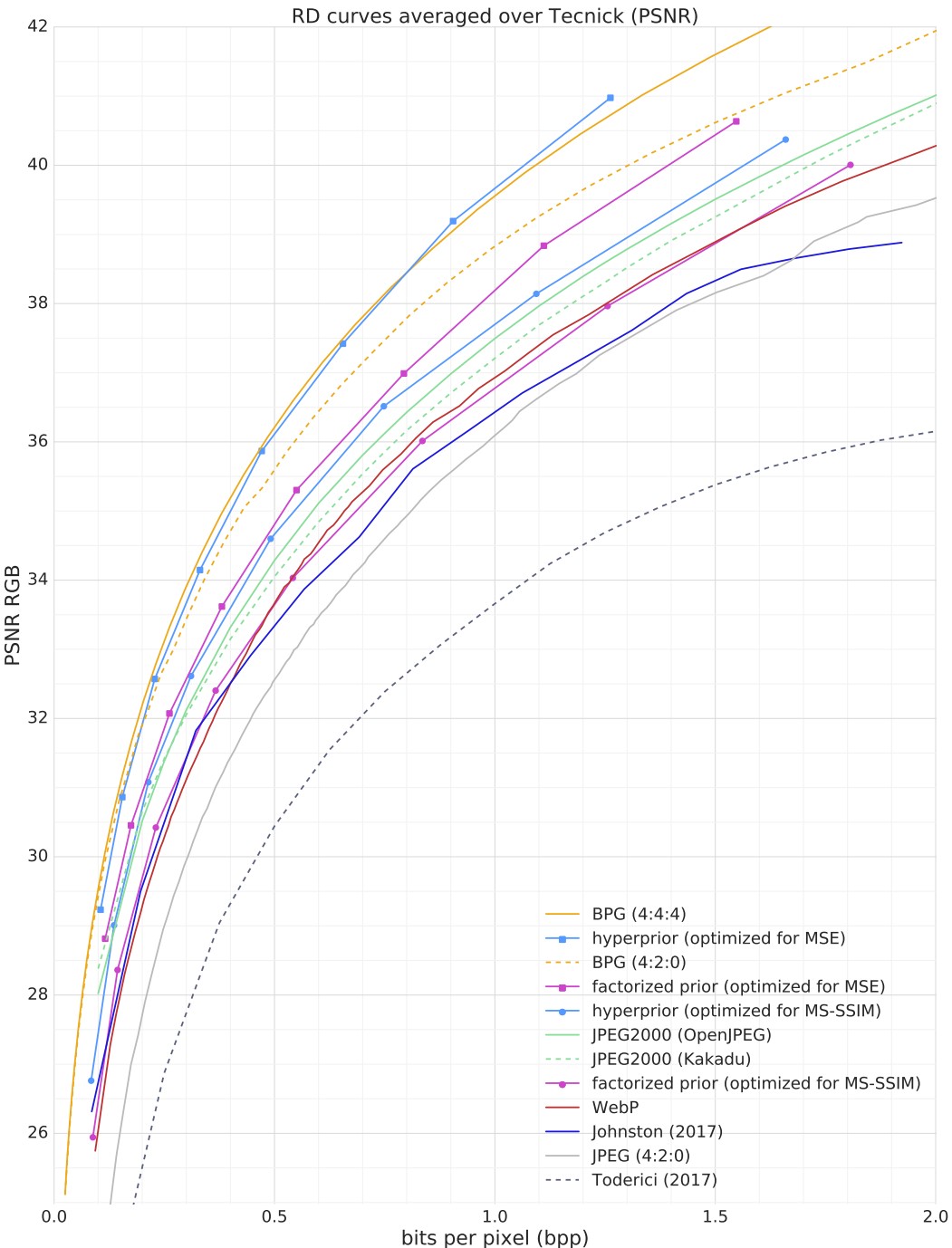

Figure 13: Rate–distortion curves for PSNR covering a wide range of conventional and ANN-based compression methods. Results are qualitatively similar to the results on Kodak.

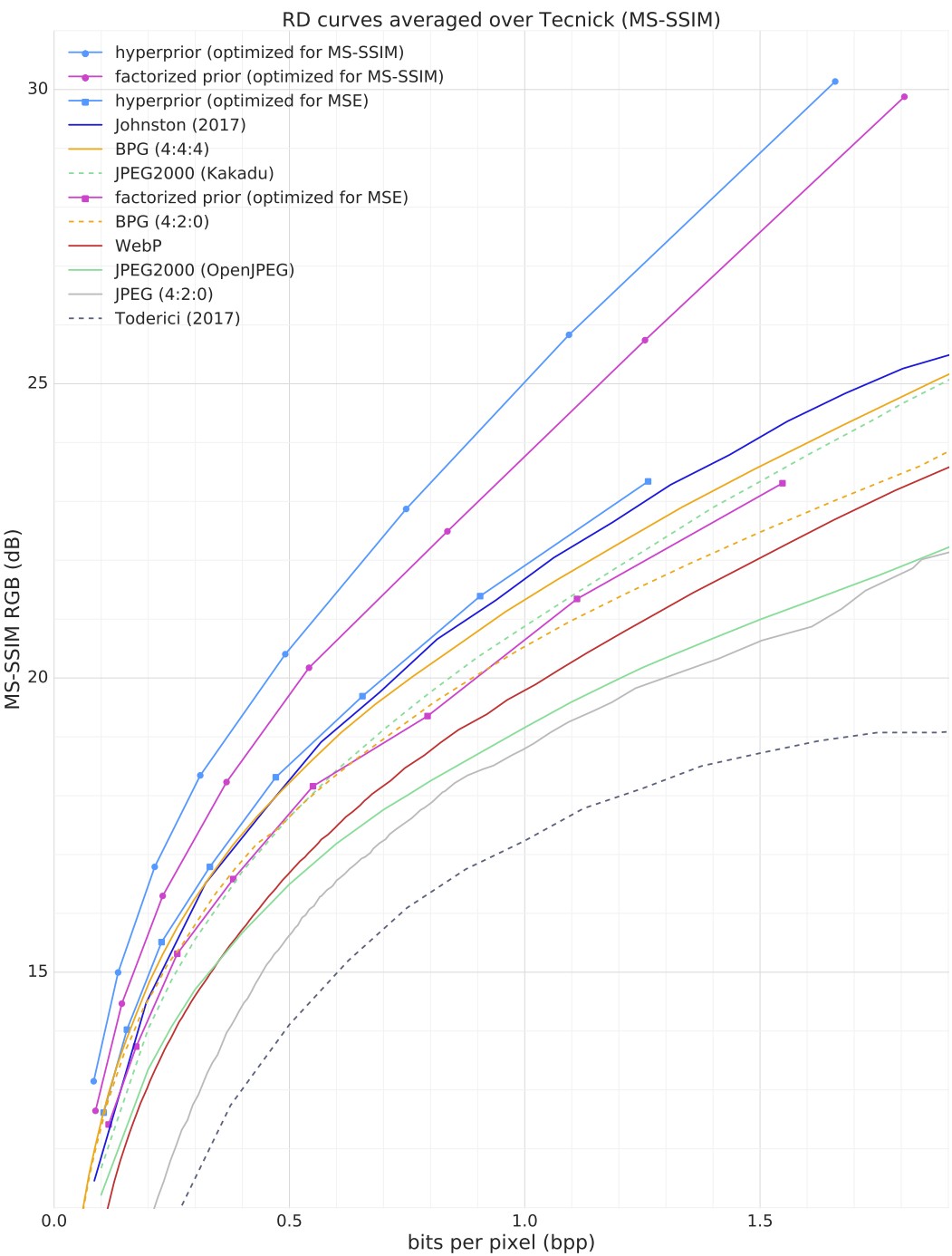

Figure 14: Rate–distortion curves for MS-SSIM covering a wide range of conventional and ANN-based compression methods. Results are qualitatively similar to the results on Kodak.

## 6.7 PERFORMANCE COMPARISONS FOR INDIVIDUAL KODAK IMAGES

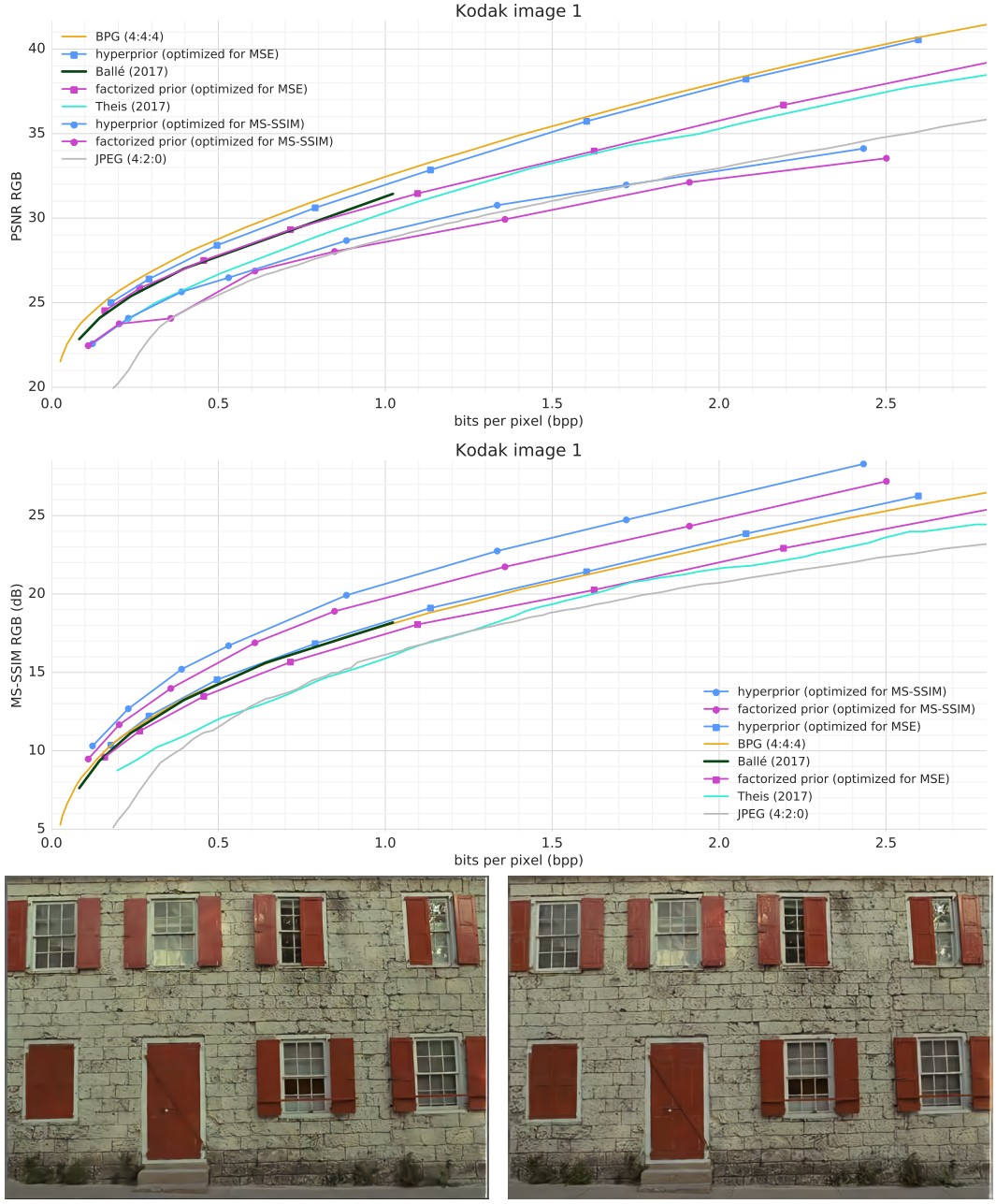

Figure 15: Results for Kodak image 01: PSNR and MS-SSIM rate-distortion curves (top), and example reconstructions for the hyperprior model optimized for squared error (bottom left) and MS-SSIM (bottom right). Images correspond to third rate–distortion point from the left of the blue curves (square and disc markers, respectively). Best viewed on a computer screen.

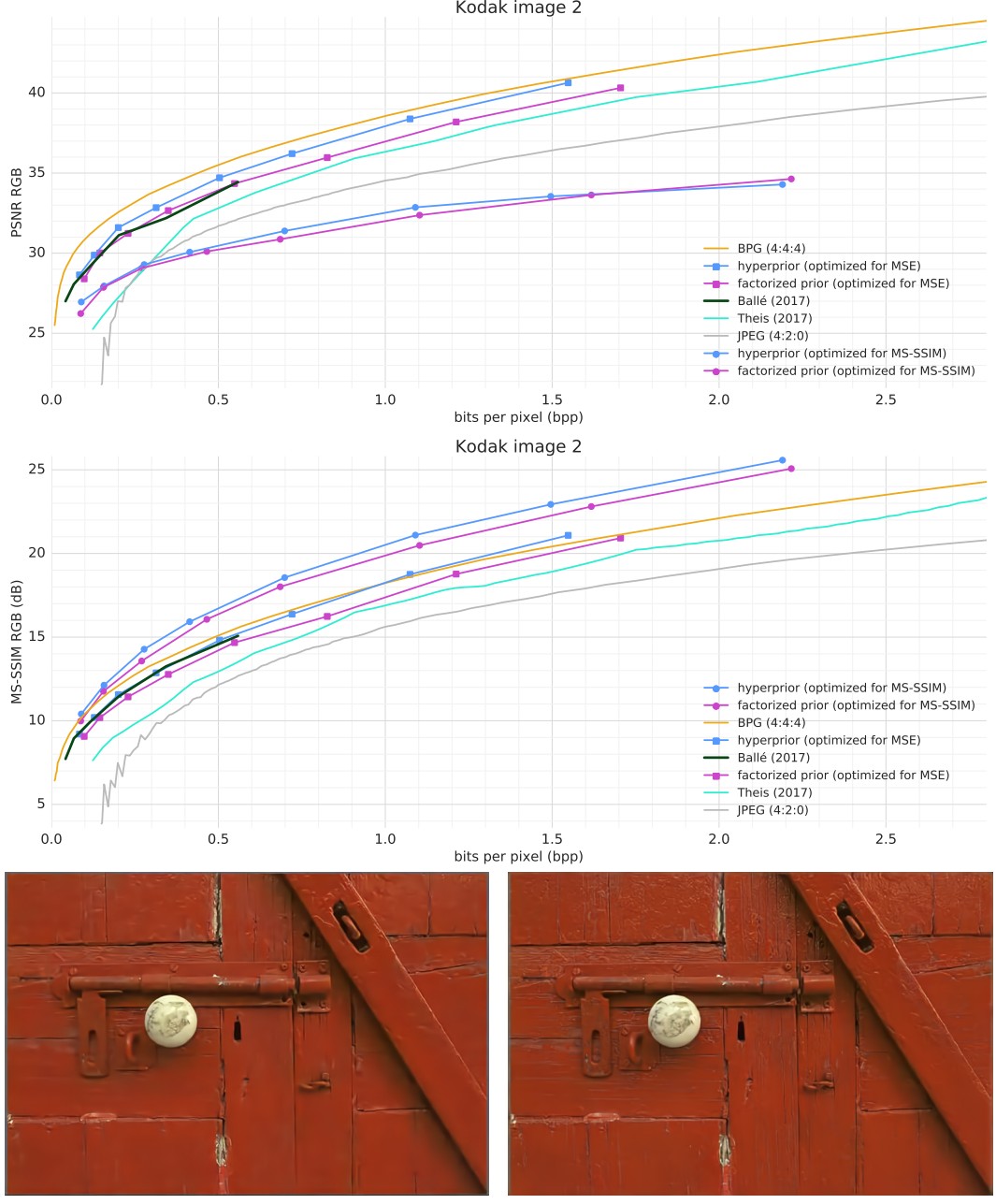

Figure 16: Results for Kodak image 02: PSNR and MS-SSIM rate-distortion curves (top), and example reconstructions for the hyperprior model optimized for squared error (bottom left) and MS-SSIM (bottom right). Images correspond to third rate–distortion point from the left of the blue curves (square and disc markers, respectively). Best viewed on a computer screen.

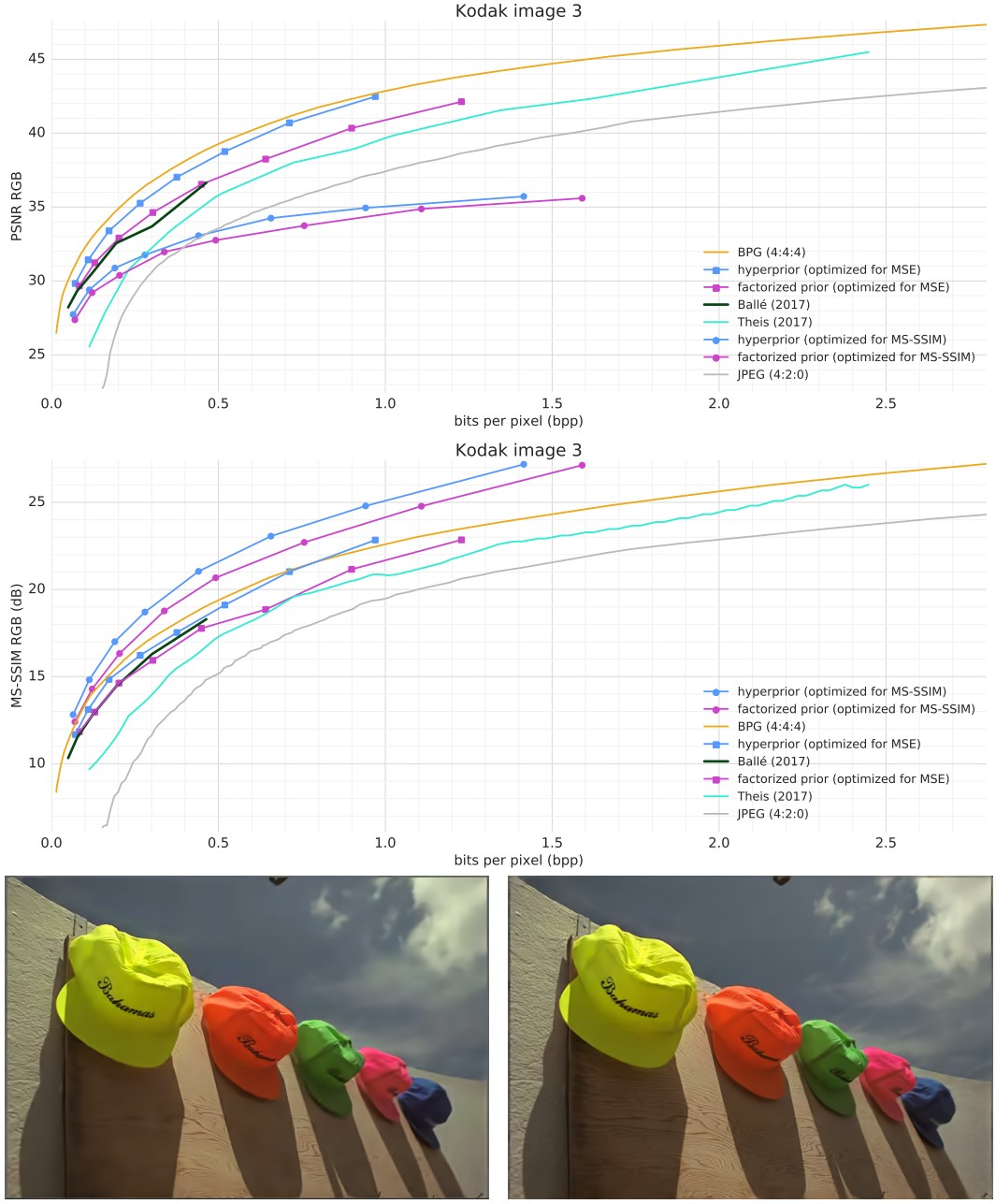

Figure 17: Results for Kodak image 03: PSNR and MS-SSIM rate-distortion curves (top), and example reconstructions for the hyperprior model optimized for squared error (bottom left) and MS-SSIM (bottom right). Images correspond to third rate–distortion point from the left of the blue curves (square and disc markers, respectively). Best viewed on a computer screen.

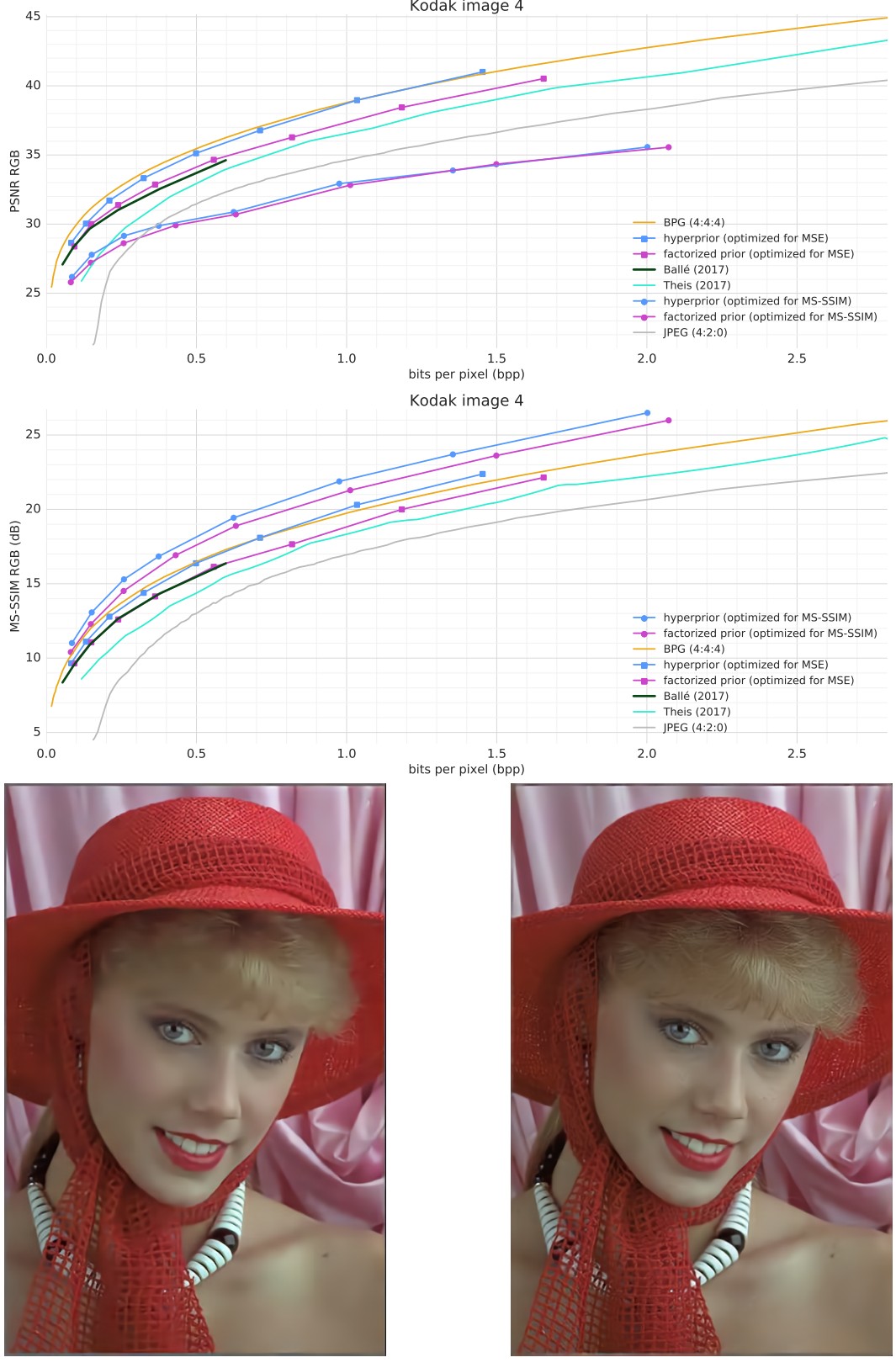

Figure 18: Results for Kodak image 04: PSNR and MS-SSIM rate-distortion curves (top), and example reconstructions for the hyperprior model optimized for squared error (bottom left) and MS-SSIM (bottom right). Images correspond to third rate–distortion point from the left of the blue curves (square and disc markers, respectively). Best viewed on a computer screen.

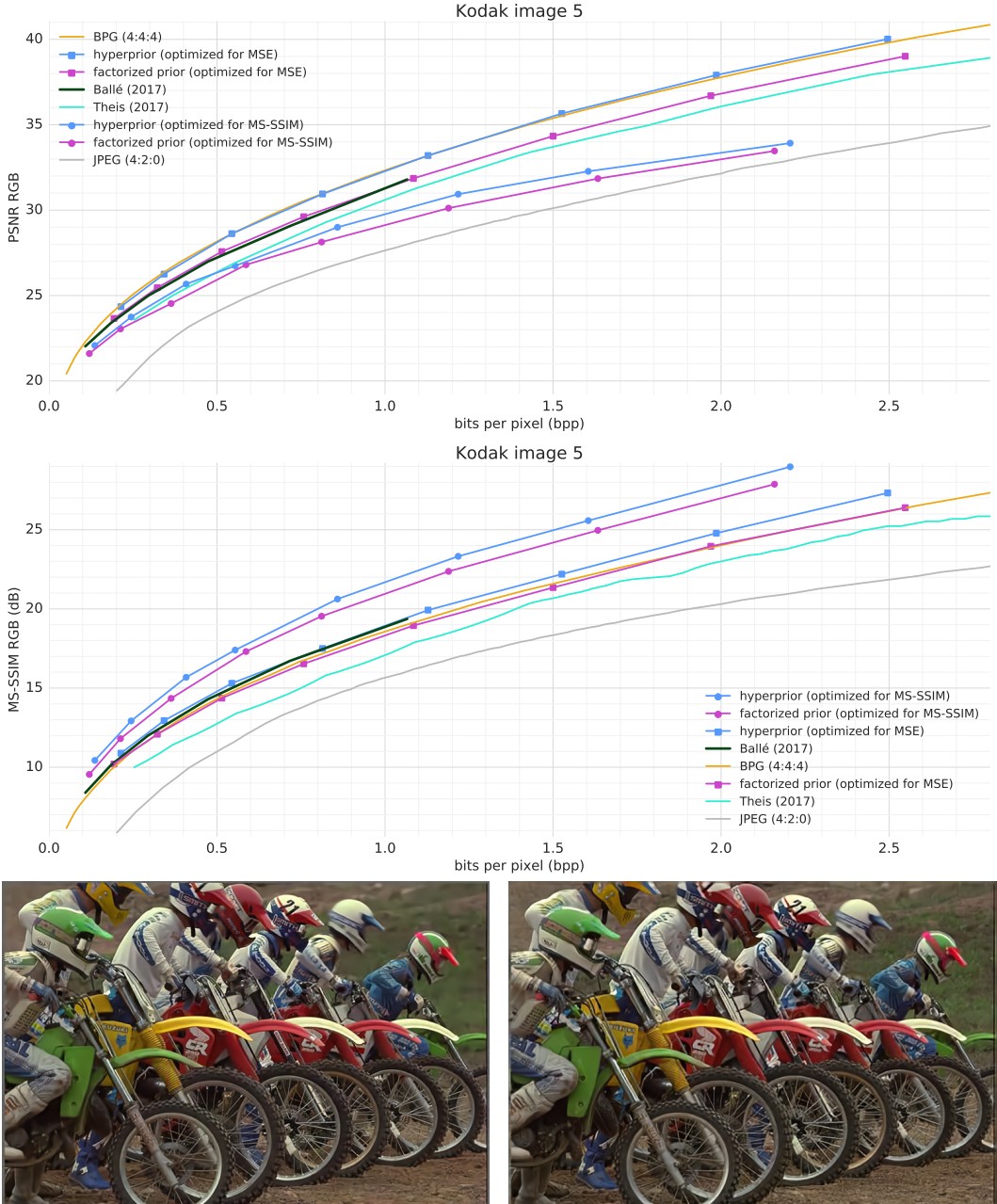

Figure 19: Results for Kodak image 05: PSNR and MS-SSIM rate-distortion curves (top), and example reconstructions for the hyperprior model optimized for squared error (bottom left) and MS-SSIM (bottom right). Images correspond to third rate–distortion point from the left of the blue curves (square and disc markers, respectively). Best viewed on a computer screen.

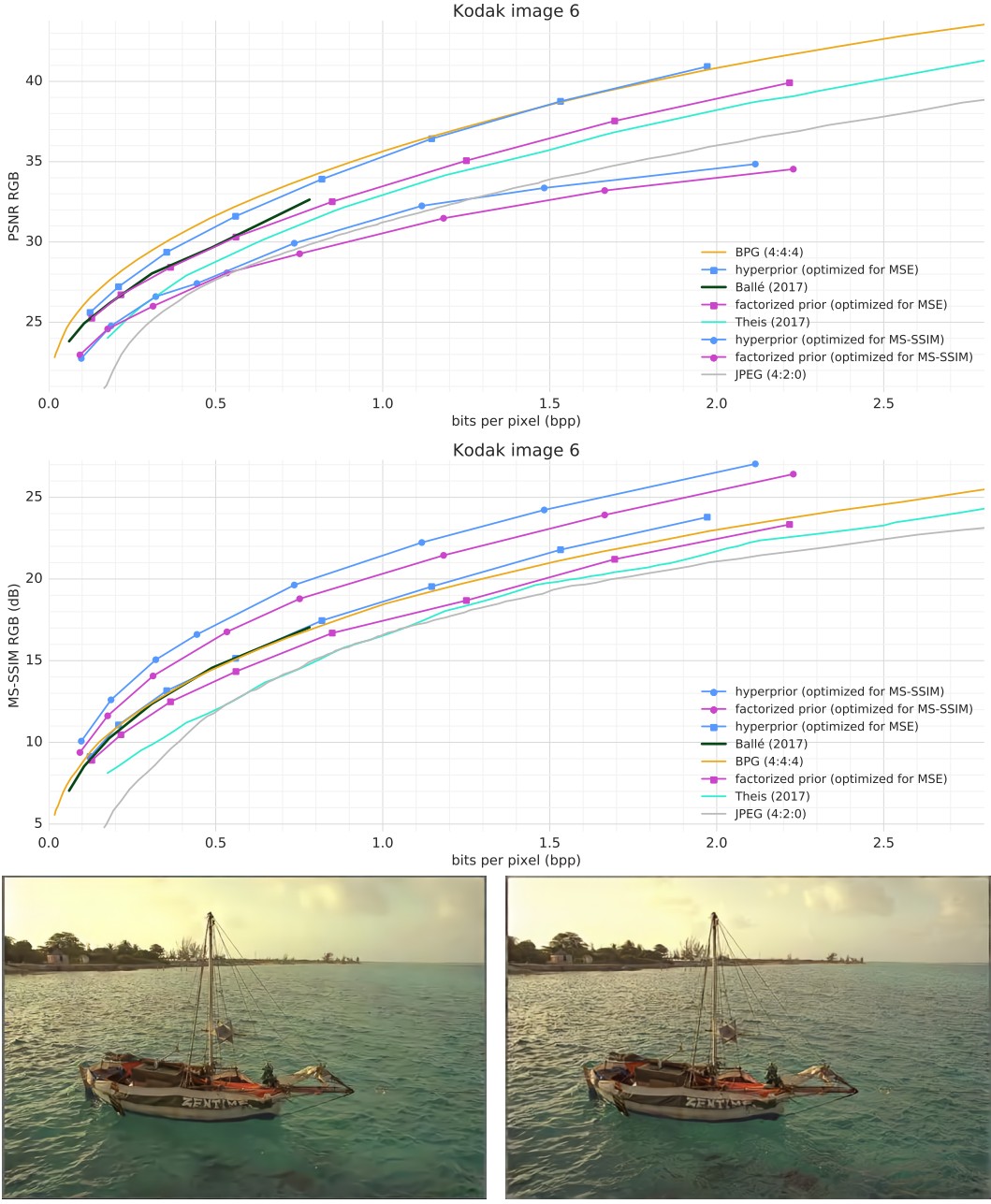

Figure 20: Results for Kodak image 06: PSNR and MS-SSIM rate-distortion curves (top), and example reconstructions for the hyperprior model optimized for squared error (bottom left) and MS-SSIM (bottom right). Images correspond to third rate–distortion point from the left of the blue curves (square and disc markers, respectively). Best viewed on a computer screen.

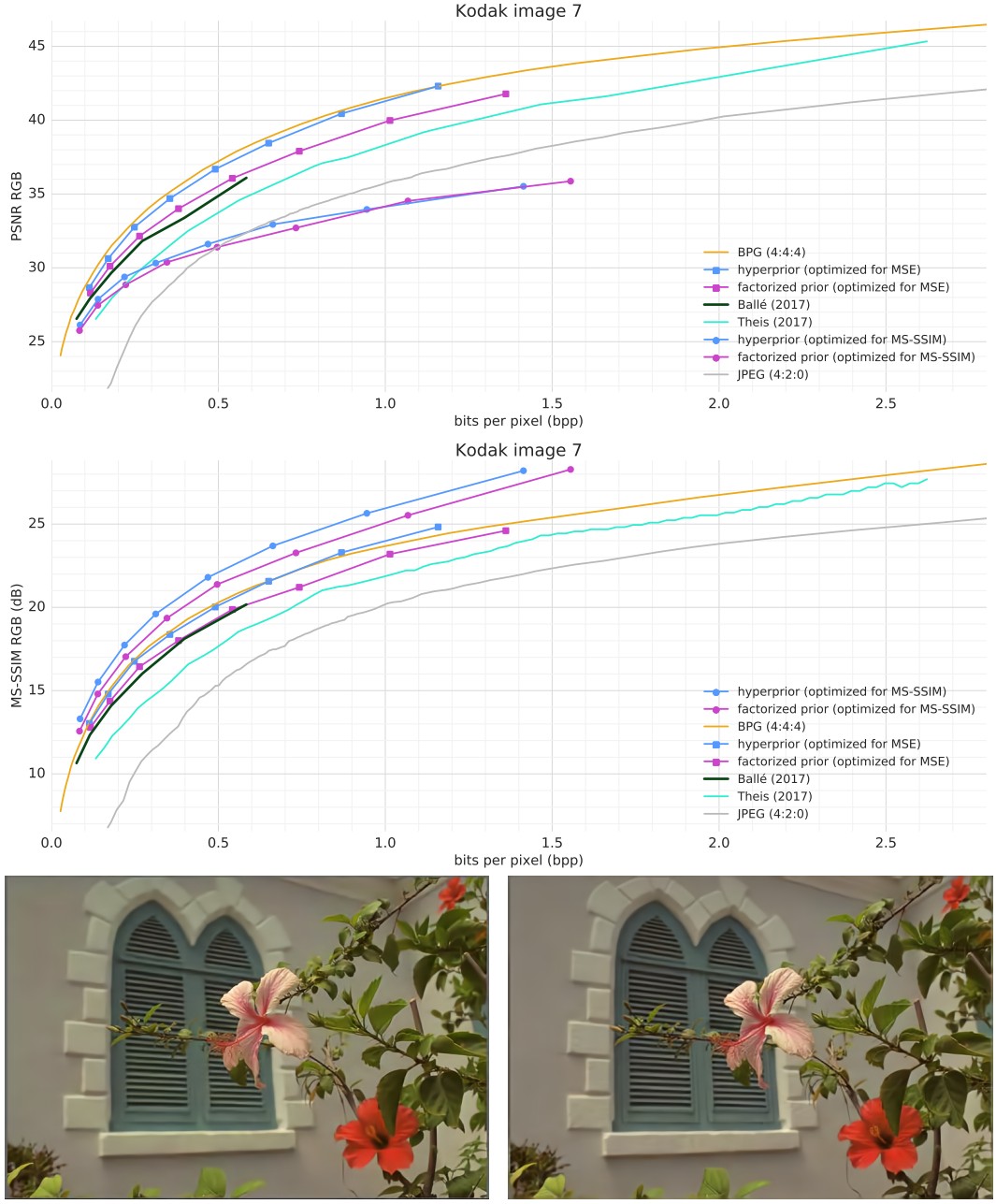

Figure 21: Results for Kodak image 07: PSNR and MS-SSIM rate-distortion curves (top), and example reconstructions for the hyperprior model optimized for squared error (bottom left) and MS-SSIM (bottom right). Images correspond to third rate–distortion point from the left of the blue curves (square and disc markers, respectively). Best viewed on a computer screen.

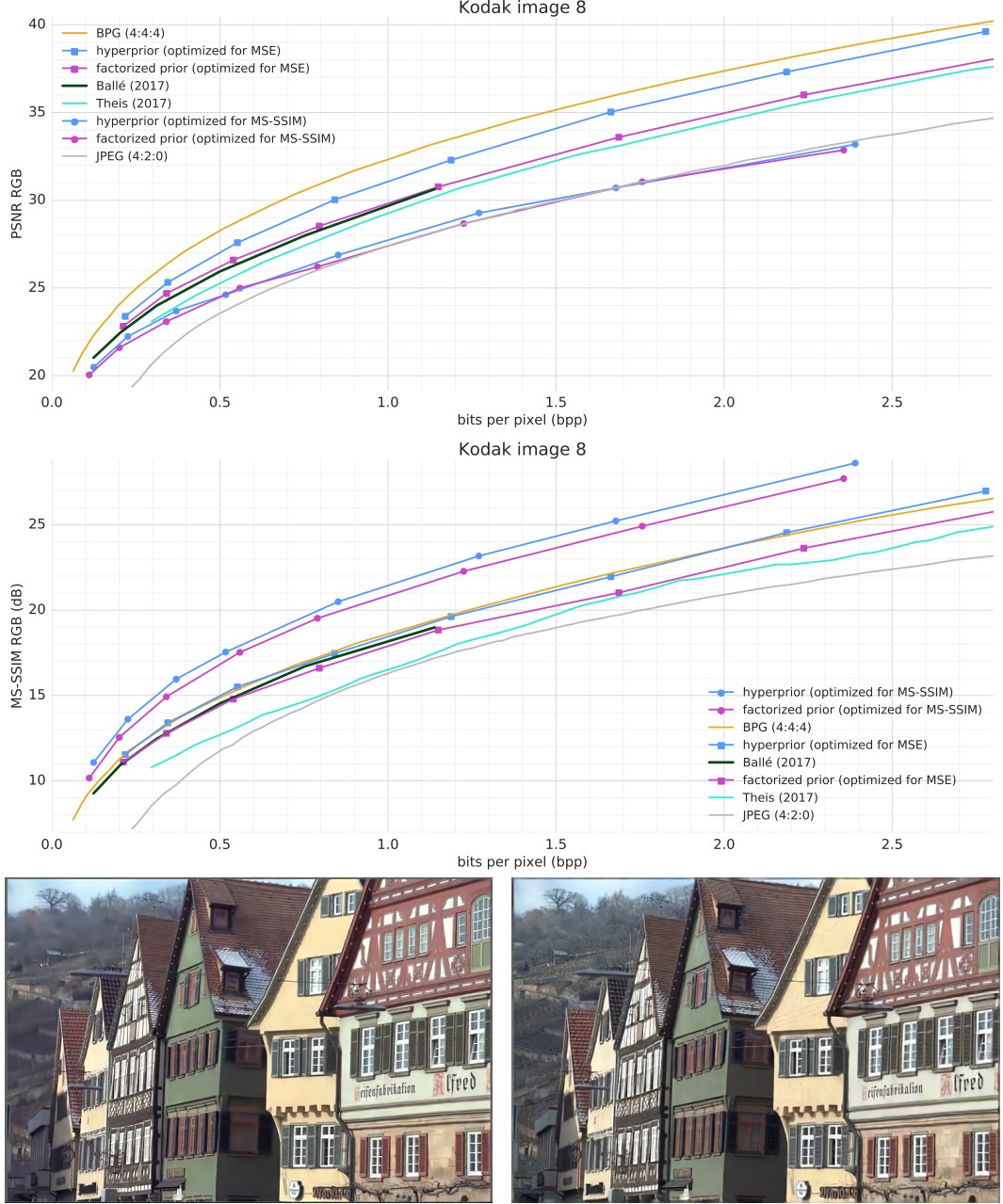

Figure 22: Results for Kodak image 08: PSNR and MS-SSIM rate-distortion curves (top), and example reconstructions for the hyperprior model optimized for squared error (bottom left) and MS-SSIM (bottom right). Images correspond to third rate–distortion point from the left of the blue curves (square and disc markers, respectively). Best viewed on a computer screen.

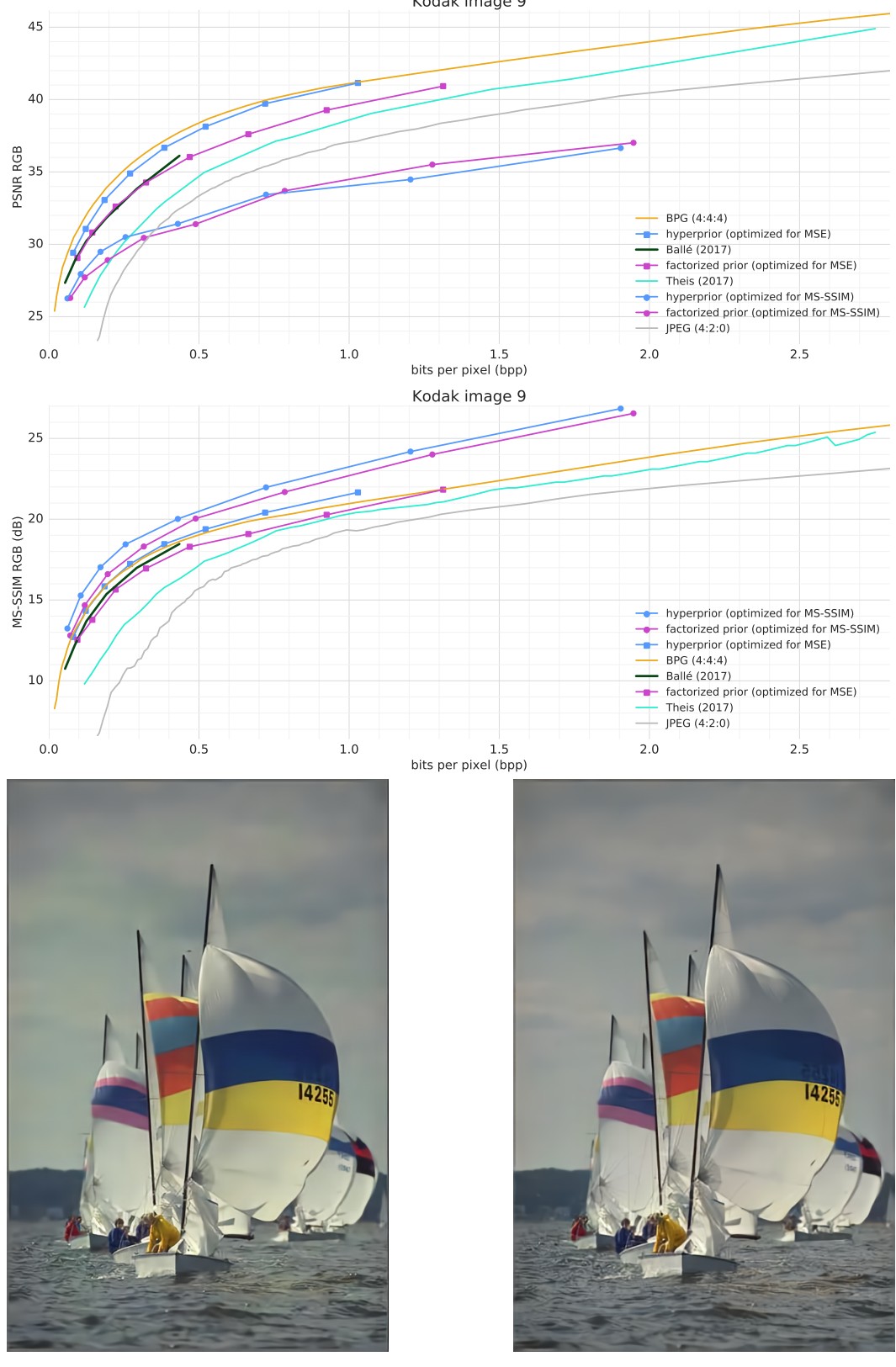

Figure 23: Results for Kodak image 09: PSNR and MS-SSIM rate-distortion curves (top), and example reconstructions for the hyperprior model optimized for squared error (bottom left) and MS-SSIM (bottom right). Images correspond to third rate–distortion point from the left of the blue curves (square and disc markers, respectively). Best viewed on a computer screen.

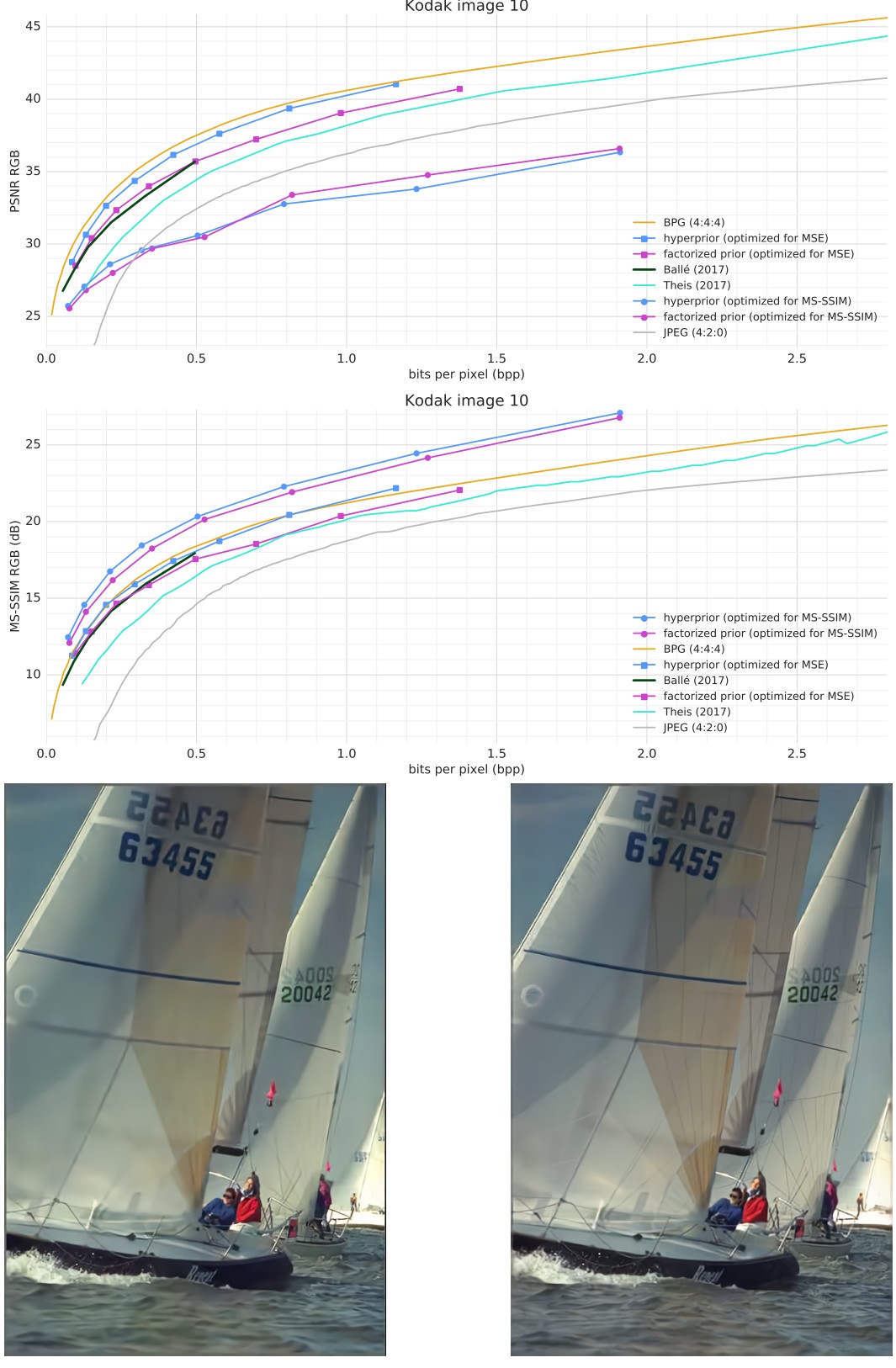

Figure 24: Results for Kodak image 10: PSNR and MS-SSIM rate-distortion curves (top), and example reconstructions for the hyperprior model optimized for squared error (bottom left) and MS-SSIM (bottom right). Images correspond to third rate–distortion point from the left of the blue curves (square and disc markers, respectively). Best viewed on a computer screen.

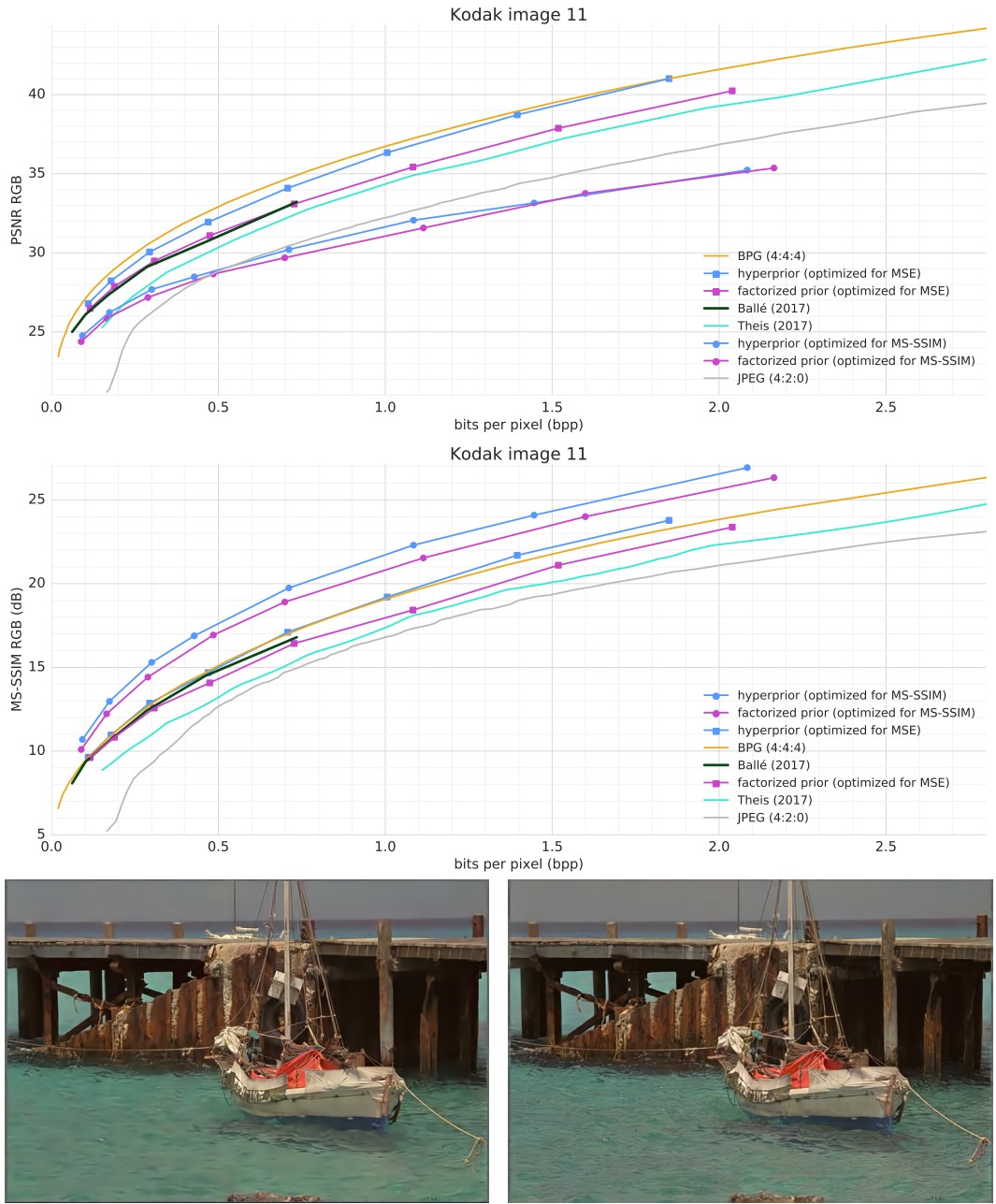

Figure 25: Results for Kodak image 11: PSNR and MS-SSIM rate-distortion curves (top), and example reconstructions for the hyperprior model optimized for squared error (bottom left) and MS-SSIM (bottom right). Images correspond to third rate–distortion point from the left of the blue curves (square and disc markers, respectively). Best viewed on a computer screen.

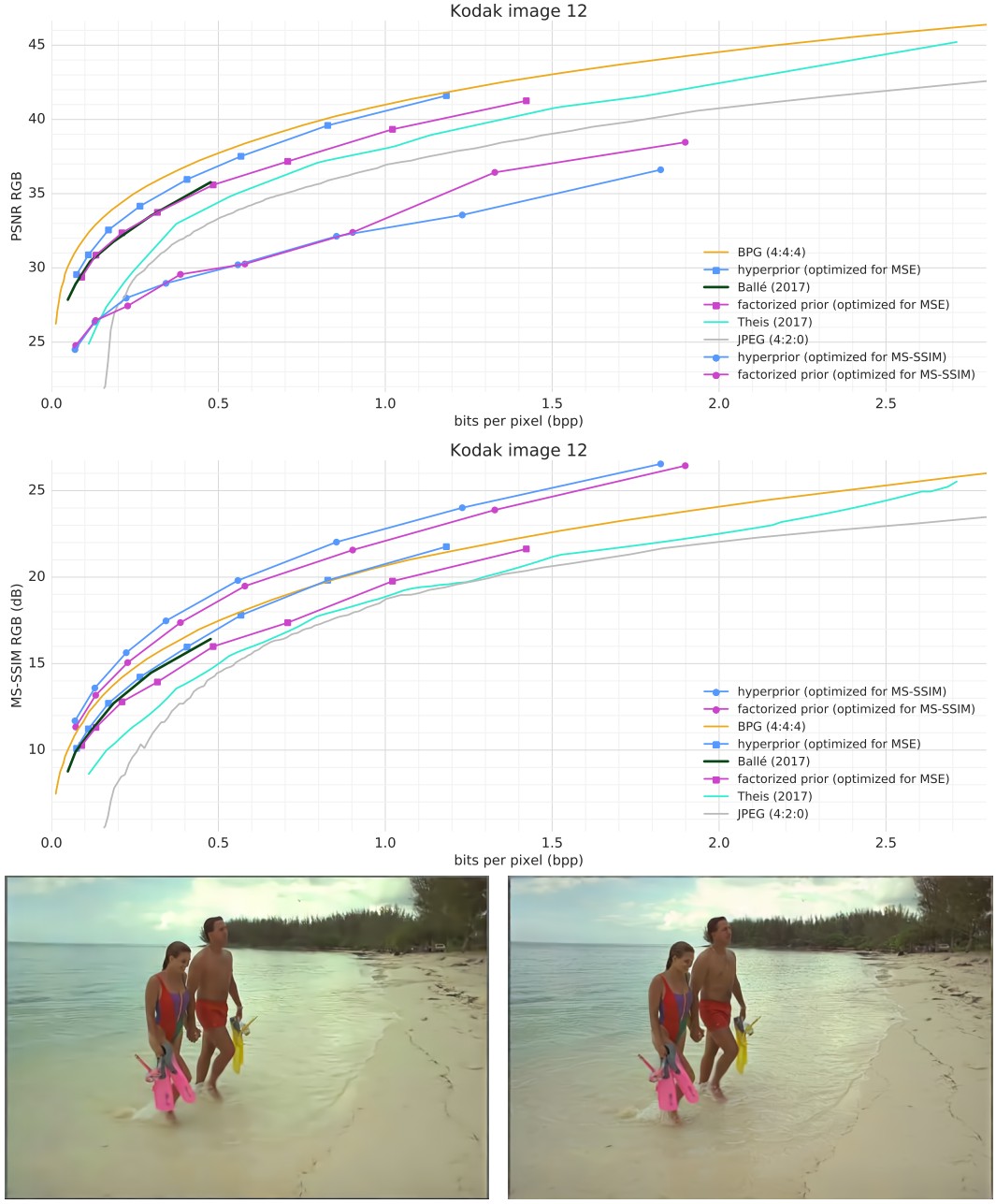

Figure 26: Results for Kodak image 12: PSNR and MS-SSIM rate-distortion curves (top), and example reconstructions for the hyperprior model optimized for squared error (bottom left) and MS-SSIM (bottom right). Images correspond to third rate–distortion point from the left of the blue curves (square and disc markers, respectively). Best viewed on a computer screen.

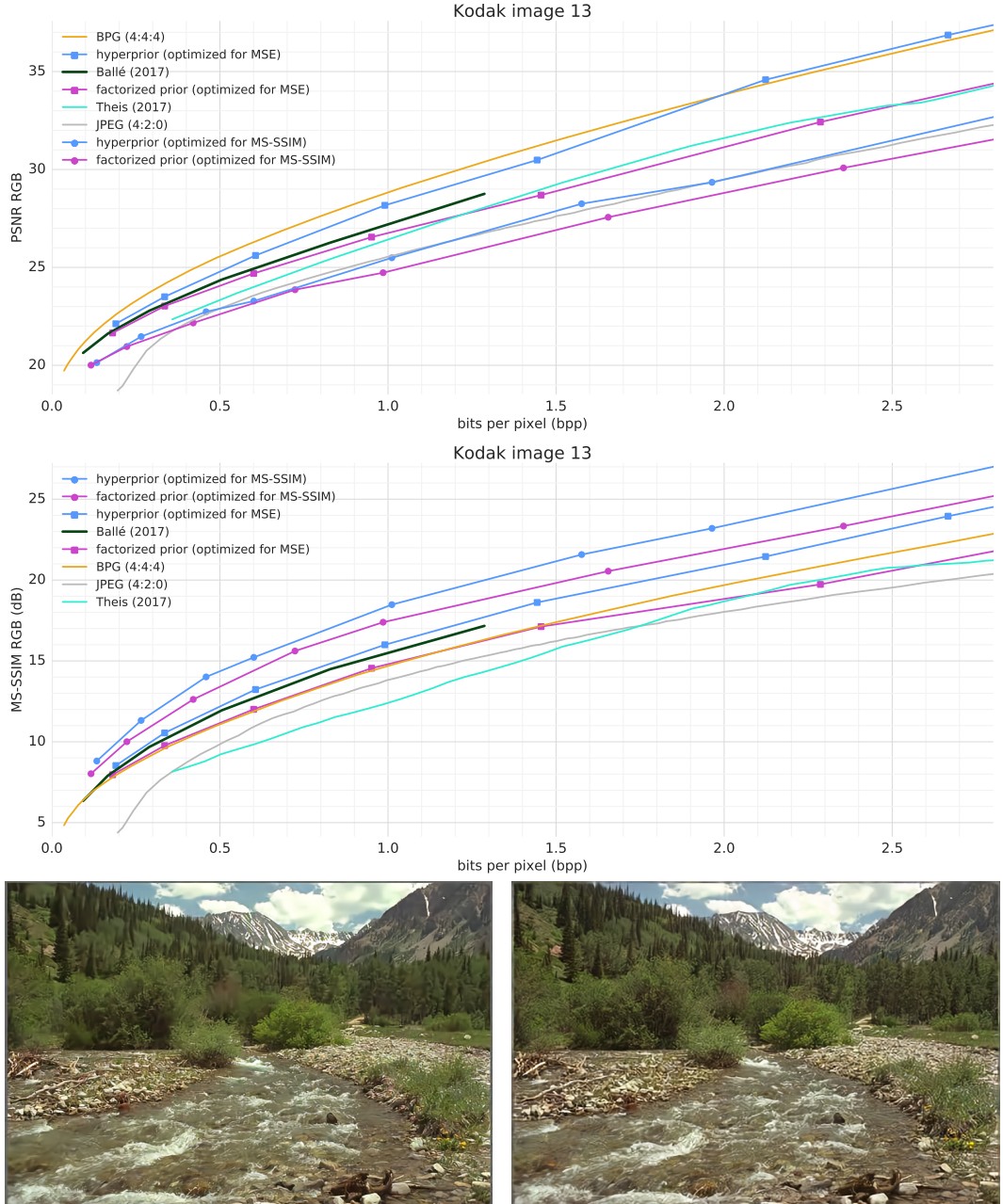

Figure 27: Results for Kodak image 13: PSNR and MS-SSIM rate-distortion curves (top), and example reconstructions for the hyperprior model optimized for squared error (bottom left) and MS-SSIM (bottom right). Images correspond to third rate–distortion point from the left of the blue curves (square and disc markers, respectively). Best viewed on a computer screen.

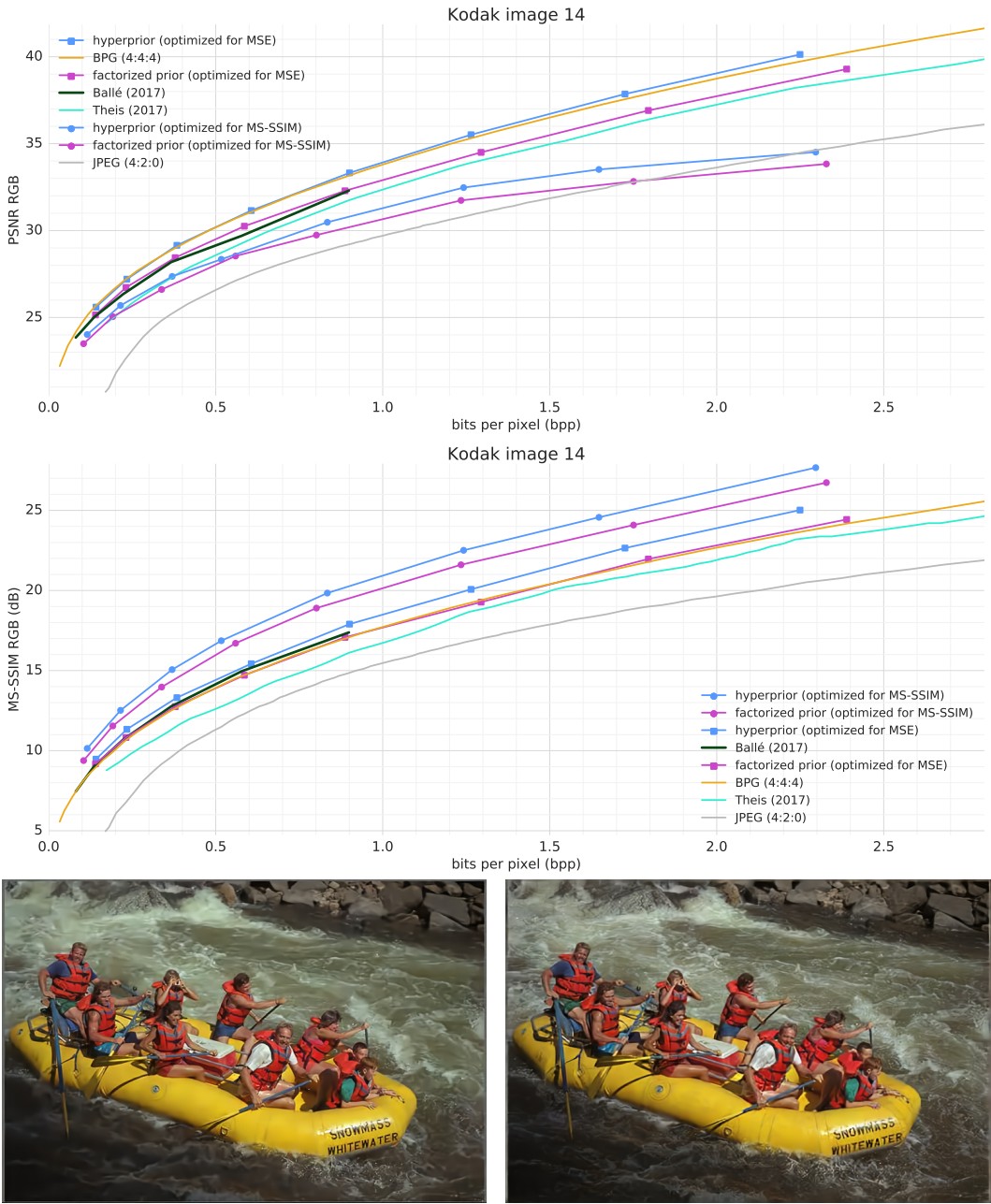

Figure 28: Results for Kodak image 14: PSNR and MS-SSIM rate-distortion curves (top), and example reconstructions for the hyperprior model optimized for squared error (bottom left) and MS-SSIM (bottom right). Images correspond to third rate–distortion point from the left of the blue curves (square and disc markers, respectively). Best viewed on a computer screen.

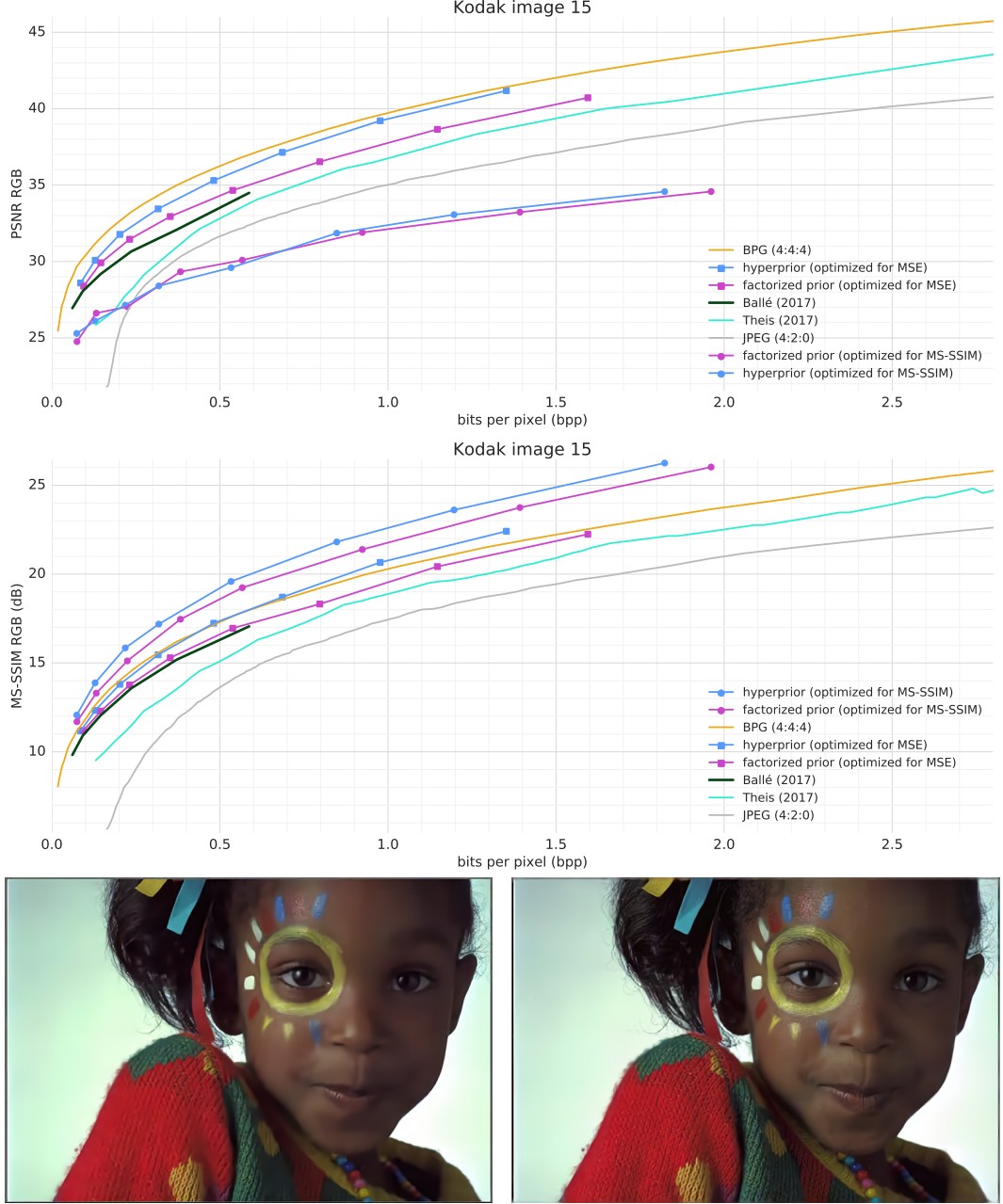

Figure 29: Results for Kodak image 15: PSNR and MS-SSIM rate-distortion curves (top), and example reconstructions for the hyperprior model optimized for squared error (bottom left) and MS-SSIM (bottom right). Images correspond to third rate–distortion point from the left of the blue curves (square and disc markers, respectively). Best viewed on a computer screen.

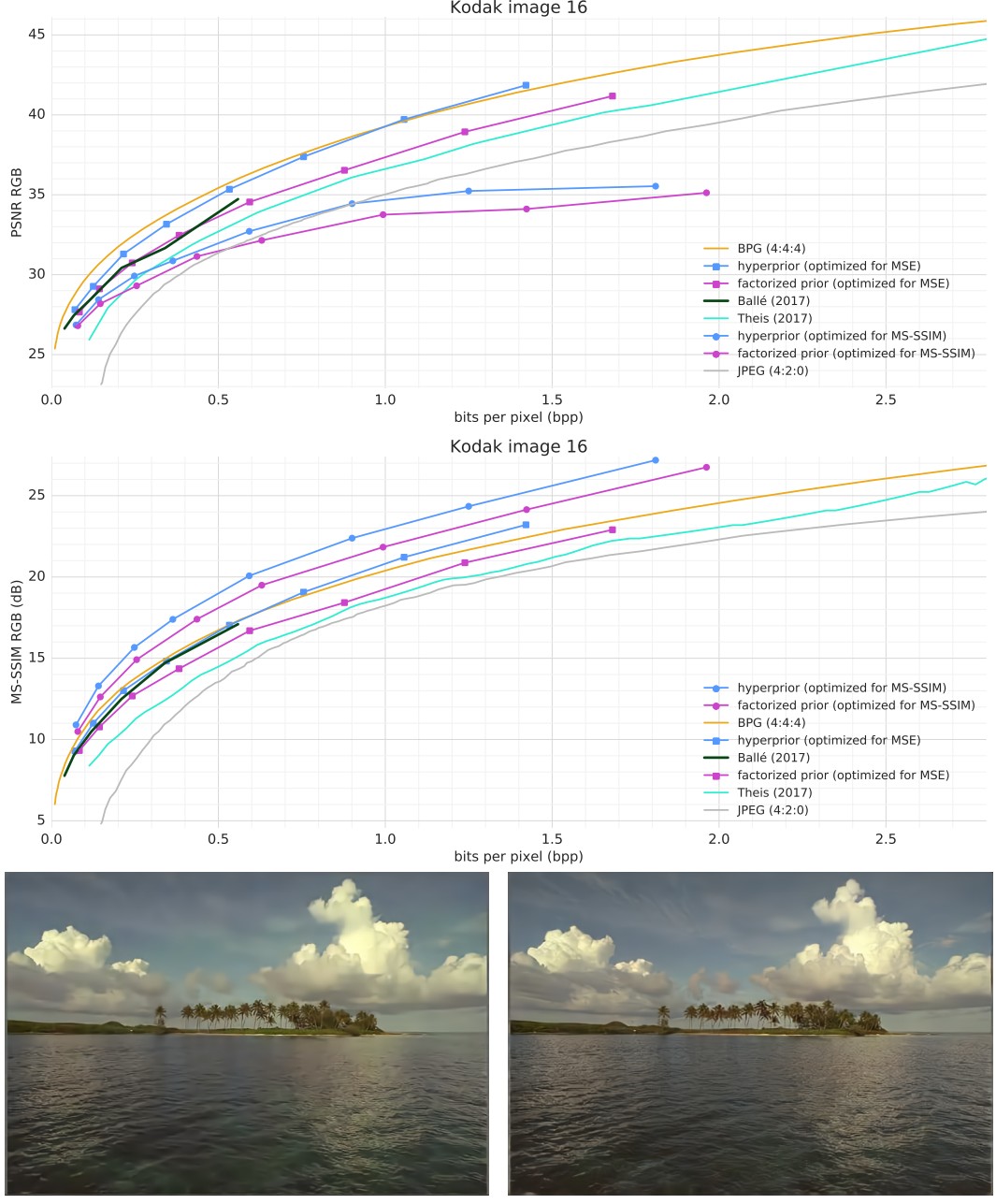

Figure 30: Results for Kodak image 16: PSNR and MS-SSIM rate-distortion curves (top), and example reconstructions for the hyperprior model optimized for squared error (bottom left) and MS-SSIM (bottom right). Images correspond to third rate–distortion point from the left of the blue curves (square and disc markers, respectively). Best viewed on a computer screen.

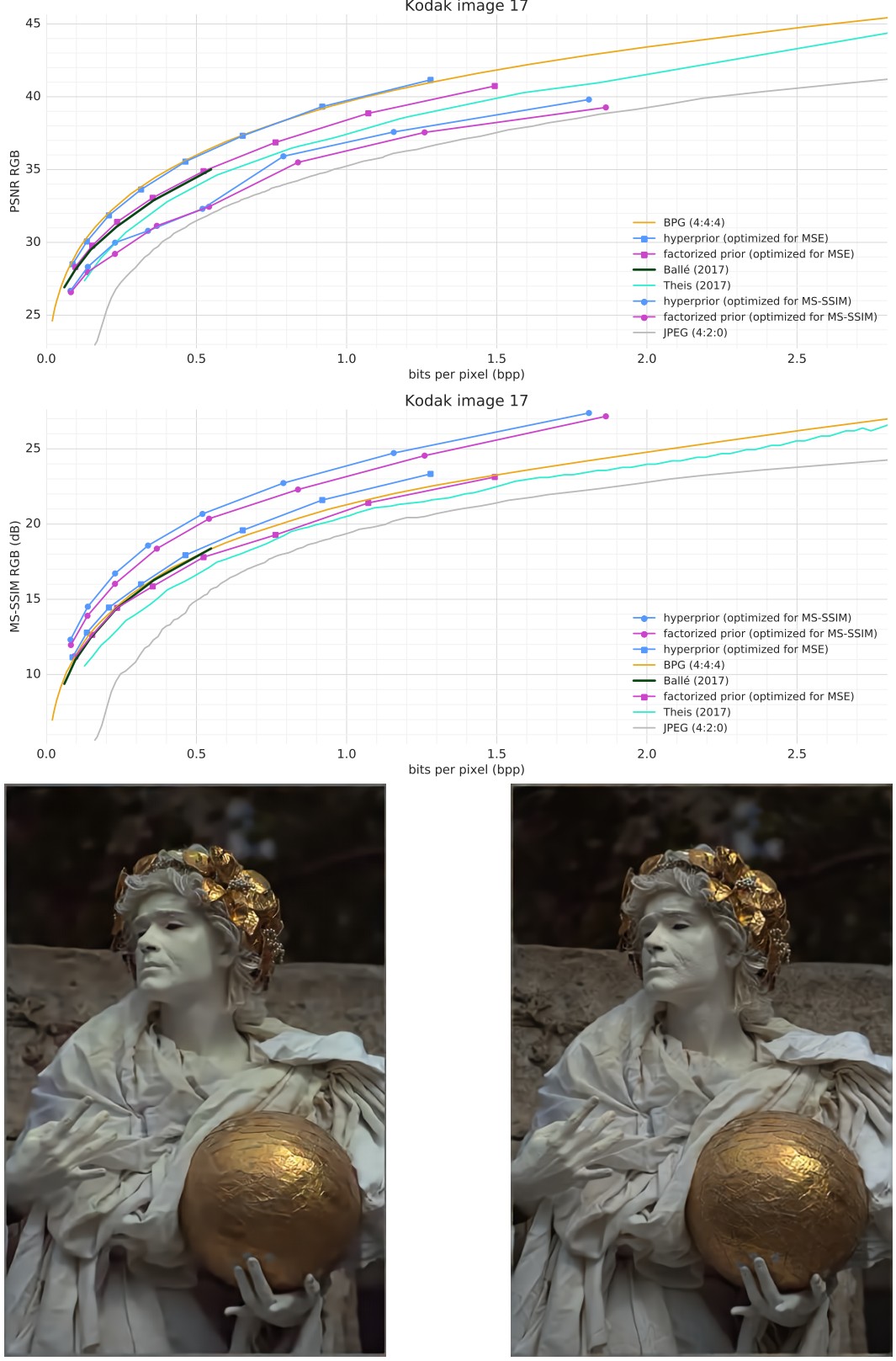

Figure 31: Results for Kodak image 17: PSNR and MS-SSIM rate-distortion curves (top), and example reconstructions for the hyperprior model optimized for squared error (bottom left) and MS-SSIM (bottom right). Images correspond to third rate–distortion point from the left of the blue curves (square and disc markers, respectively). Best viewed on a computer screen.

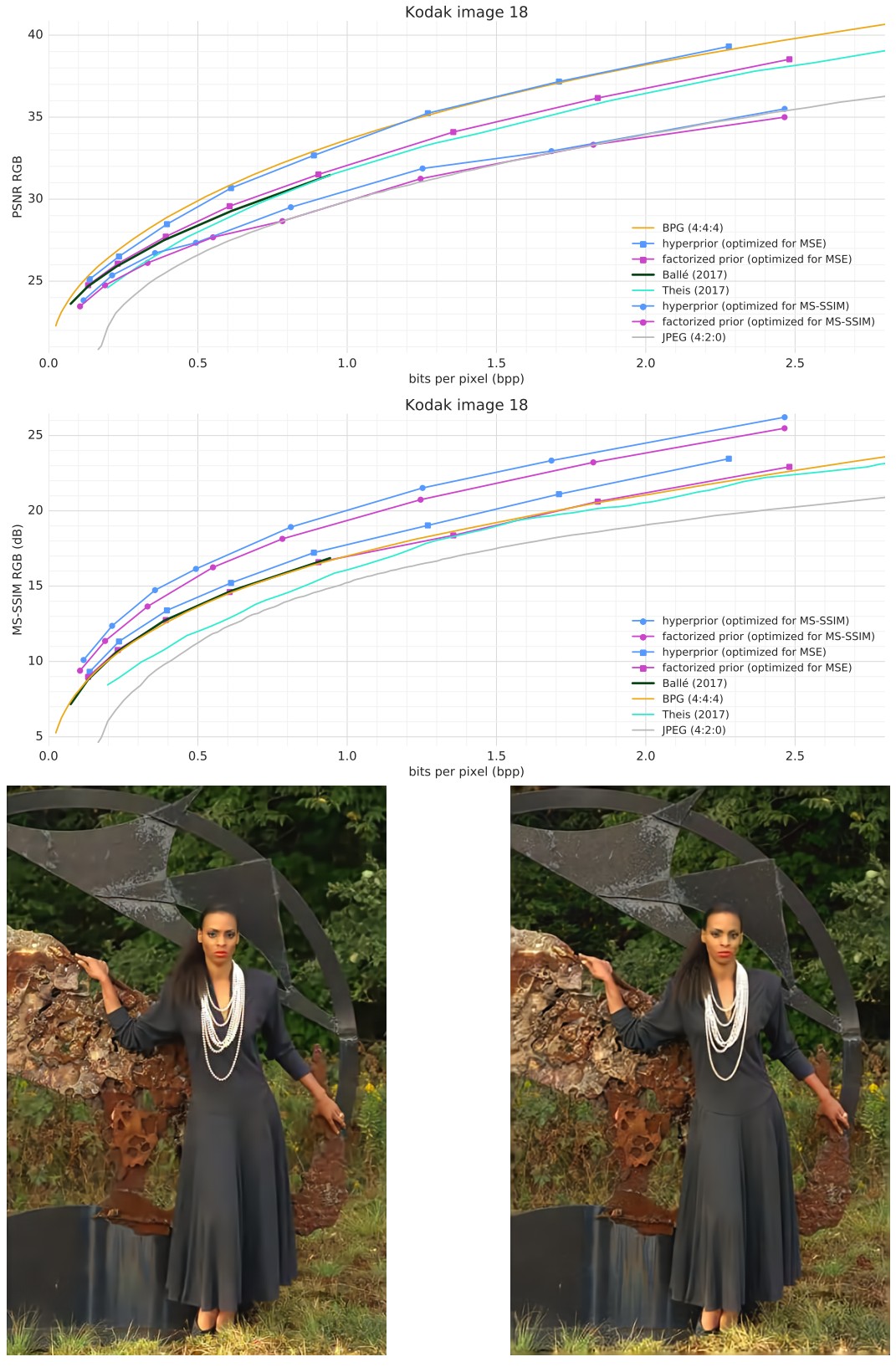

Figure 32: Results for Kodak image 18: PSNR and MS-SSIM rate-distortion curves (top), and example reconstructions for the hyperprior model optimized for squared error (bottom left) and MS-SSIM (bottom right). Images correspond to third rate–distortion point from the left of the blue curves (square and disc markers, respectively). Best viewed on a computer screen.

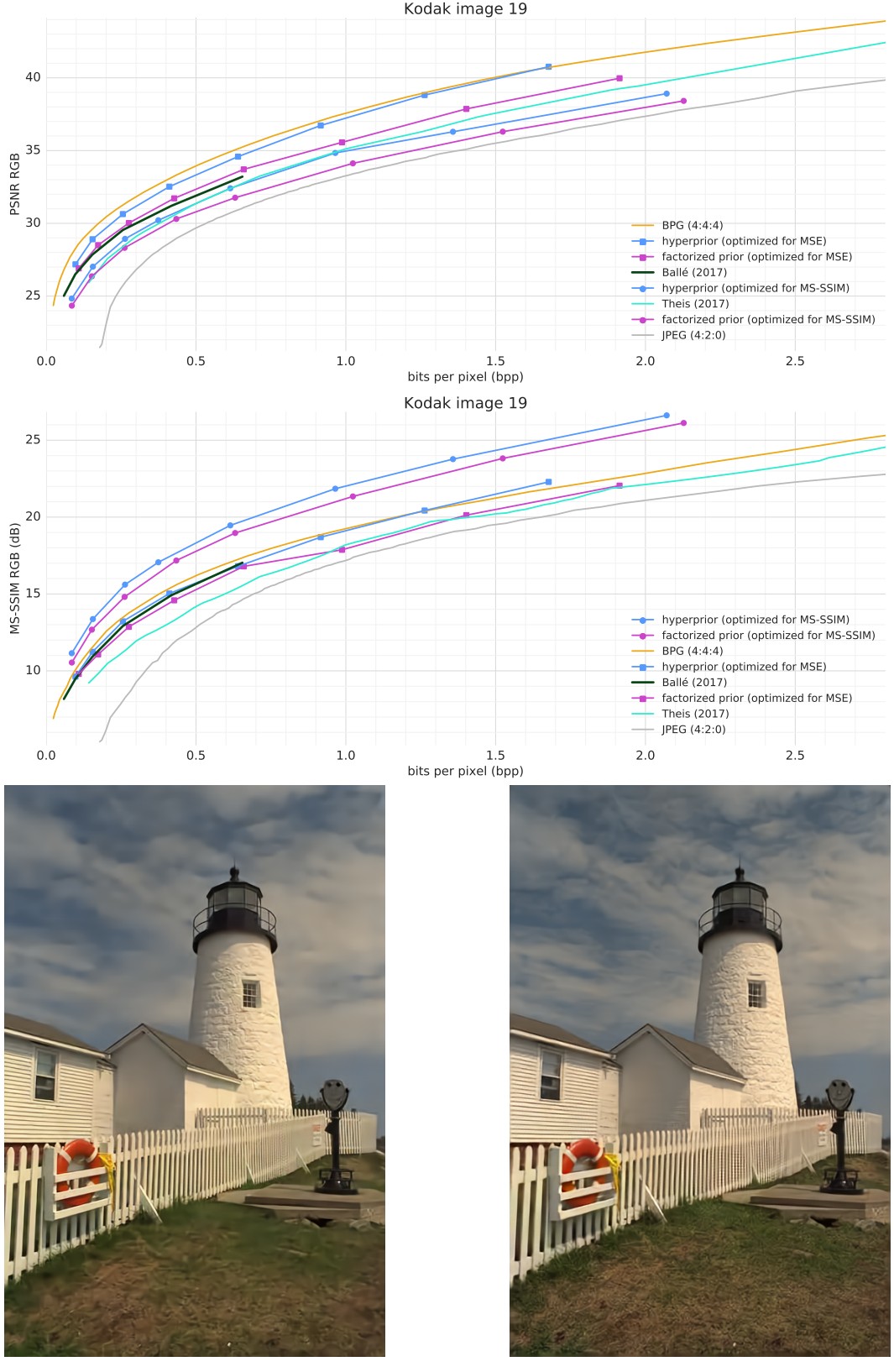

Figure 33: Results for Kodak image 19: PSNR and MS-SSIM rate-distortion curves (top), and example reconstructions for the hyperprior model optimized for squared error (bottom left) and MS-SSIM (bottom right). Images correspond to third rate–distortion point from the left of the blue curves (square and disc markers, respectively). Best viewed on a computer screen.

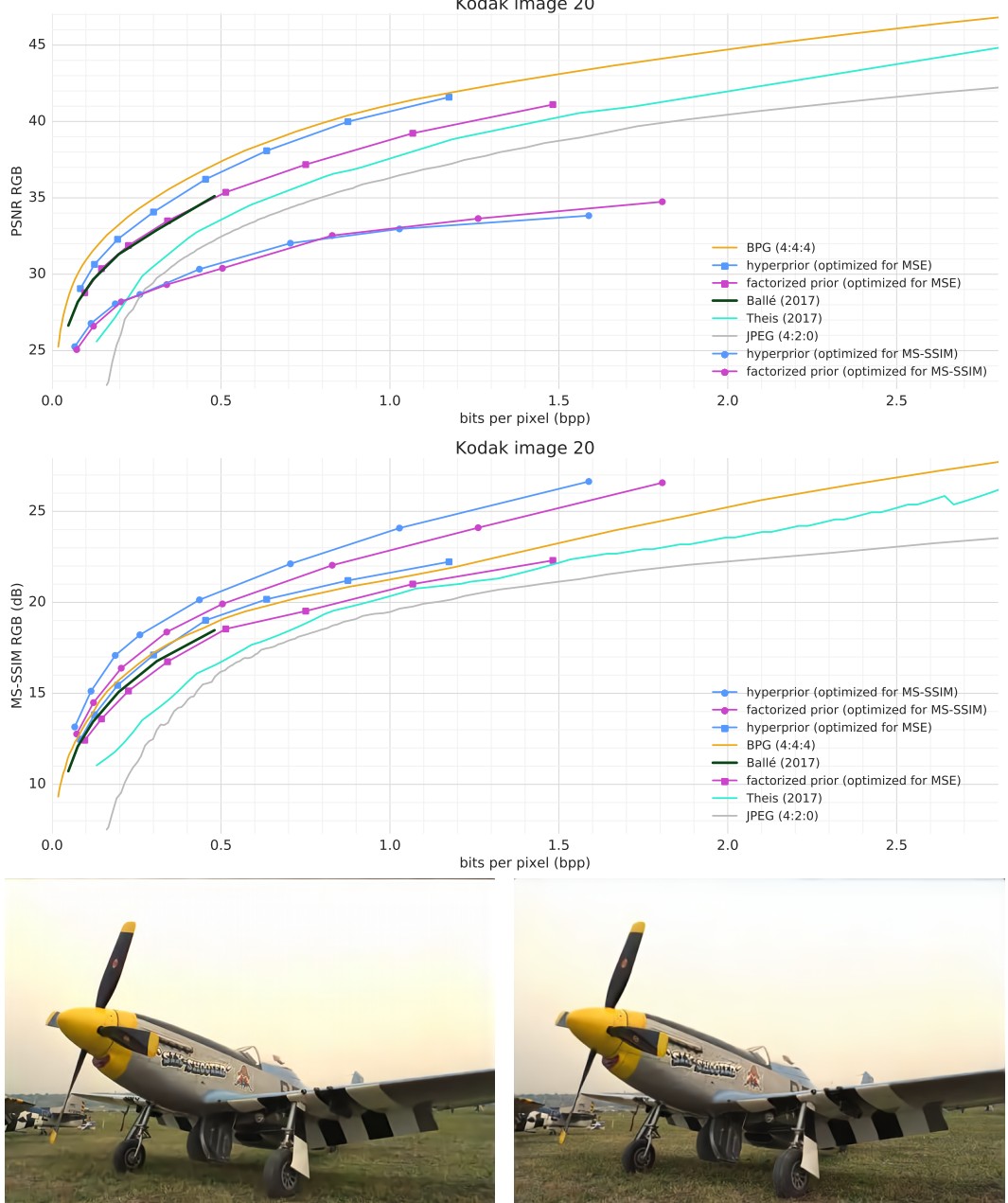

Figure 34: Results for Kodak image 20: PSNR and MS-SSIM rate-distortion curves (top), and example reconstructions for the hyperprior model optimized for squared error (bottom left) and MS-SSIM (bottom right). Images correspond to third rate–distortion point from the left of the blue curves (square and disc markers, respectively). Best viewed on a computer screen.

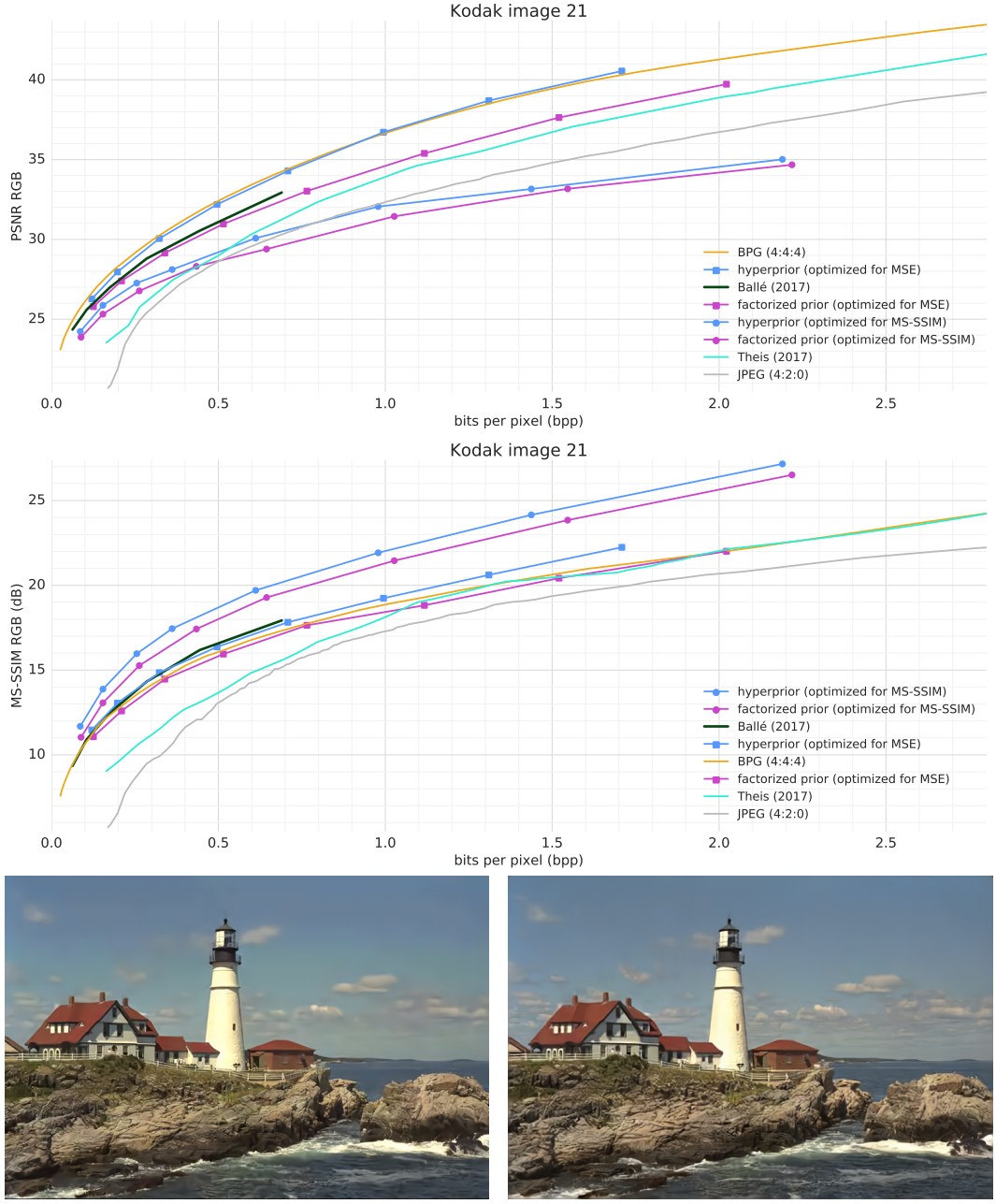

Figure 35: Results for Kodak image 21: PSNR and MS-SSIM rate-distortion curves (top), and example reconstructions for the hyperprior model optimized for squared error (bottom left) and MS-SSIM (bottom right). Images correspond to third rate–distortion point from the left of the blue curves (square and disc markers, respectively). Best viewed on a computer screen.

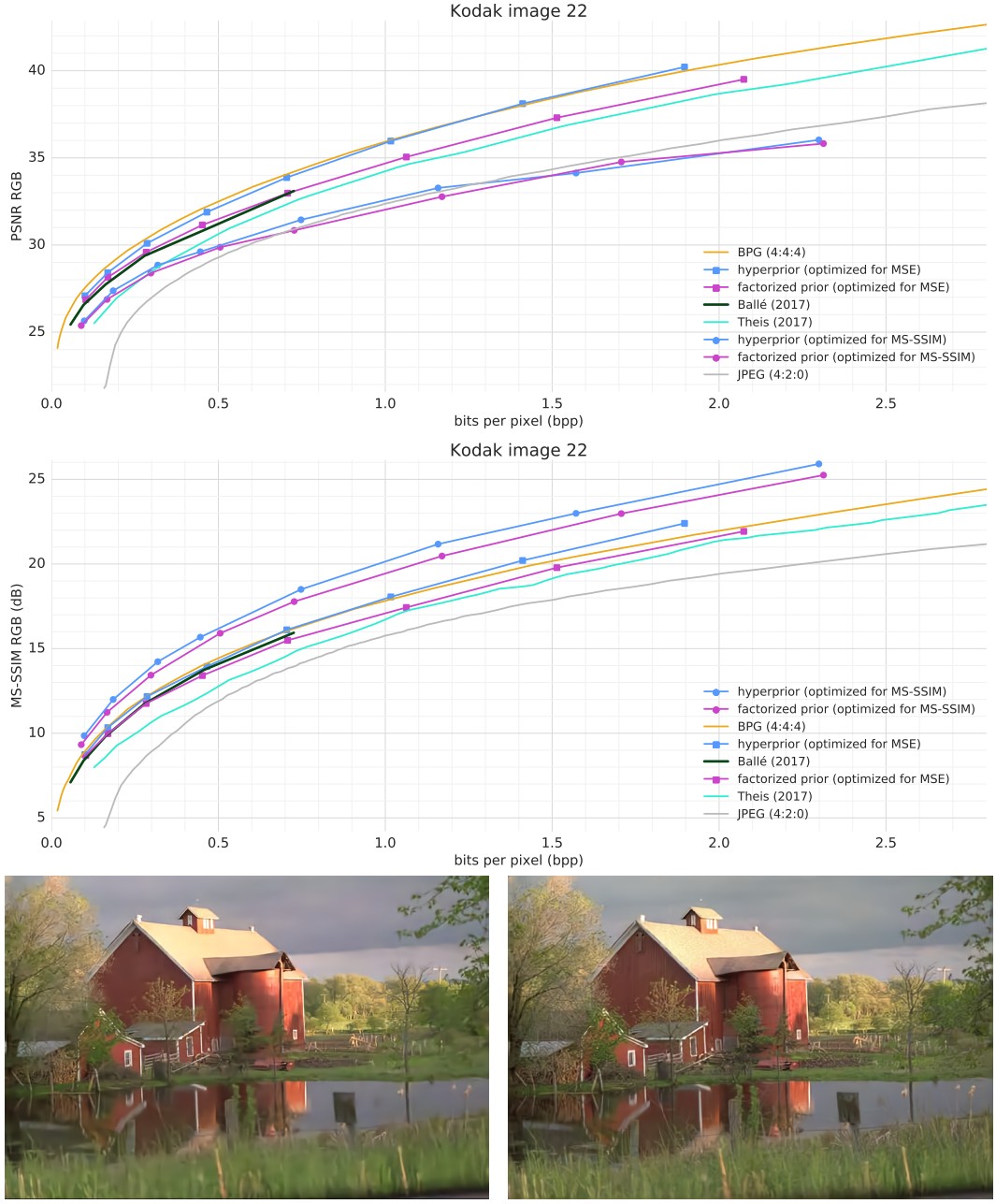

Figure 36: Results for Kodak image 22: PSNR and MS-SSIM rate-distortion curves (top), and example reconstructions for the hyperprior model optimized for squared error (bottom left) and MS-SSIM (bottom right). Images correspond to third rate–distortion point from the left of the blue curves (square and disc markers, respectively). Best viewed on a computer screen.

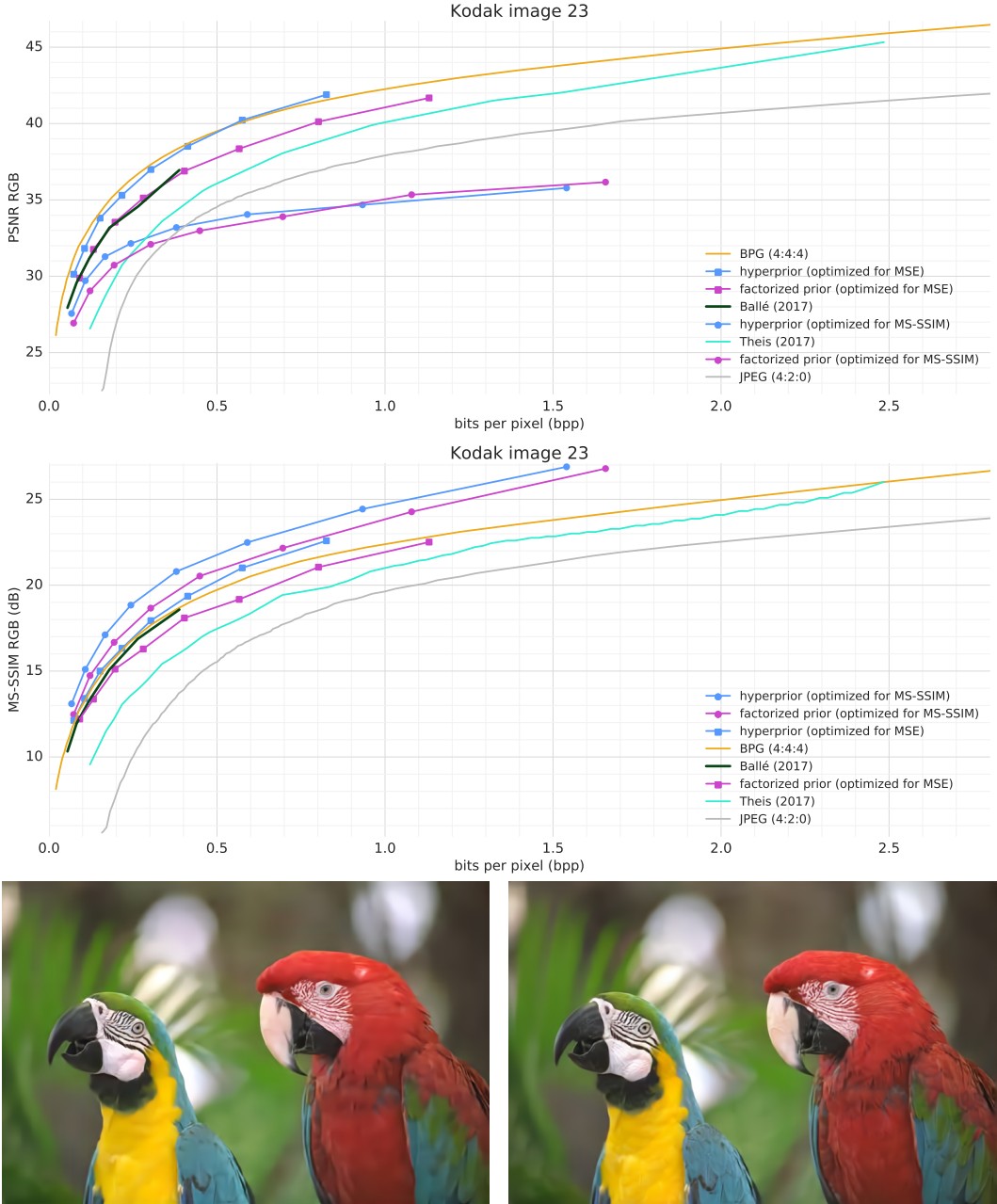

Figure 37: Results for Kodak image 23: PSNR and MS-SSIM rate-distortion curves (top), and example reconstructions for the hyperprior model optimized for squared error (bottom left) and MS-SSIM (bottom right). Images correspond to third rate–distortion point from the left of the blue curves (square and disc markers, respectively). Best viewed on a computer screen.

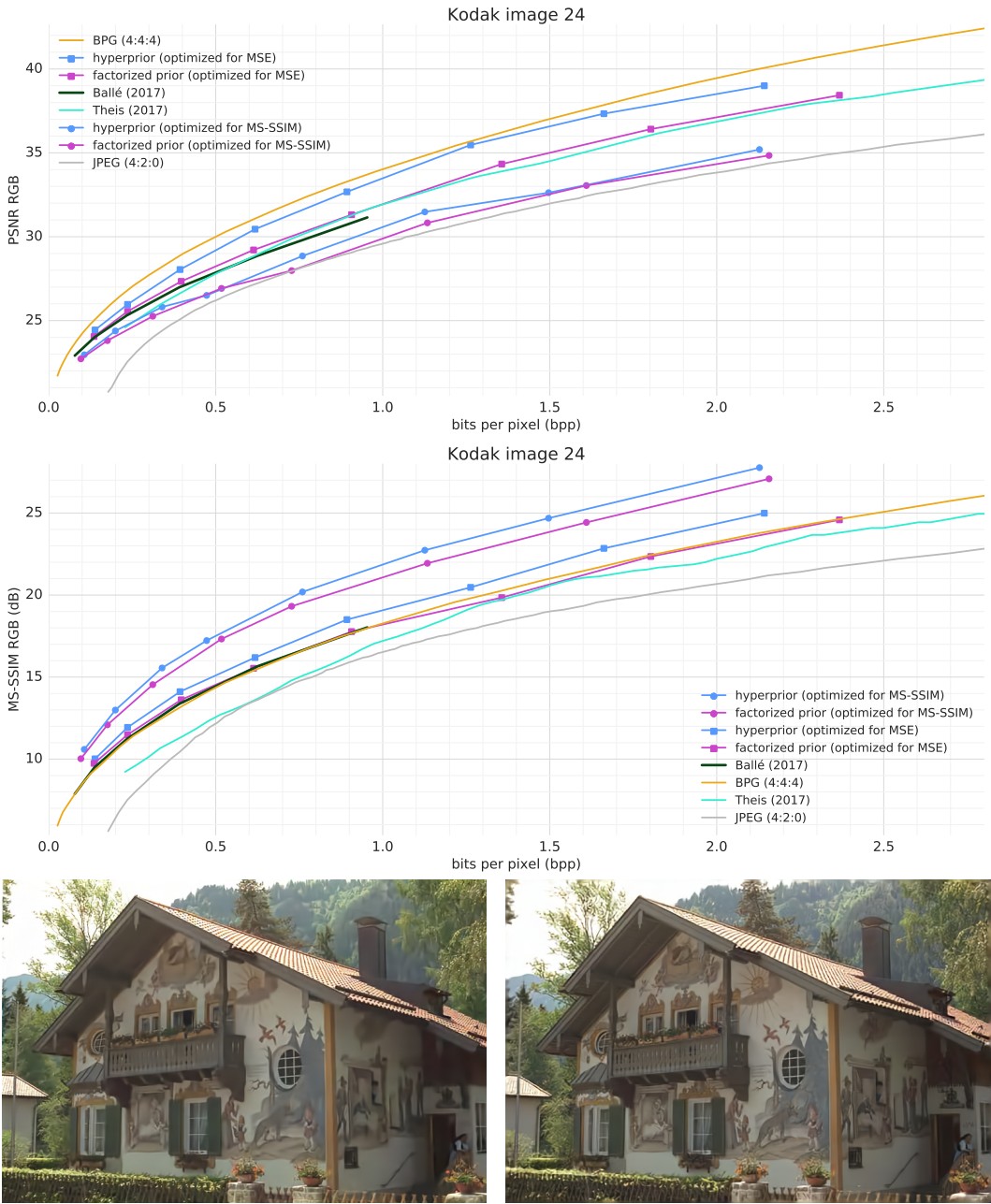

Figure 38: Results for Kodak image 24: PSNR and MS-SSIM rate-distortion curves (top), and example reconstructions for the hyperprior model optimized for squared error (bottom left) and MS-SSIM (bottom right). Images correspond to third rate–distortion point from the left of the blue curves (square and disc markers, respectively). Best viewed on a computer screen.

