# OpenReview forum: "Variational image compression with a scale hyperprior"
_ICLR.cc/2018/Conference — Accept (Poster)_

### Official Review · AnonReviewer1 · 2017-11-26
**Hierarchical entropy model a good idea; value of MS-SSIM improvement not clear**

**Rating:** 7
**Confidence:** 4

**Review:**

Summary:

This paper extends the work of Balle et al. (2016, 2017) on using certain types of variational autoencoders for image compression. After encoding pixels with a convolutional net with GDN nonlinearities, the quantized coefficients are entropy encoded. Where before the coefficients were independently encoded, the coefficients are now jointly modeled using a latent variable model. In particular, the model exploits dependencies in the scale of neighboring coefficients. The additional latent variables are used to efficiently represent these scales. Both the coefficients and the representation of the scales are quantized and encoded in the binary image representation.

Review:

Lossy image compression using neural networks is a rapidly advancing field and of considerable interest to the ICLR community. I like the approach of using a hierarchical entropy model, which may inspire further work in this direction. It is nice to see that the variational approach may be able to outperform the more complicated state-of-the-art approach of Rippel and Bourdev (2017). That said, the evaluation is in terms of MS-SSIM and only the network directly optimized for MS-SSIM outperformed the adversarial approach of R&B. Since the reconstructions generated by a network optimized for MSE tend to look better than those of the MS-SSIM network (Figure 6), I am wondering if the proposed approach is indeed outperforming R&B or just exploiting a weakness of MS-SSIM. It would have been great if the authors included a comparison based on human judgments or at least a side-by-side comparison of reconstructions generated by the two approaches.

It might be interesting to relate the entropy model used here to other work involving scale mixtures, e.g. the field of Gaussian scale mixtures (Lyu & Simoncelli, 2007).

Another interesting comparison might be to other compression approaches where scale mixtures were used and pixels were encoded together with scales (e.g., van den Oord & Schrauwen, 2017).

The authors combine their approach using MS-SSIM as distortion. Is this technically still a VAE? Might be worth discussing.

I did not quite follow the motivation for convolving the prior distributions with a uniform distribution.

The paper is mostly well written and clear. Minor suggestions:

– On page 3 the paper talks about “the true posterior” of a model which hasn’t been defined yet. Although most readers will not stumble here as they will be familiar with VAEs, perhaps mention first that the generative model is defined over both $x$ and $\tilde y$.

– Below Equation 2 it sounds like the authors claim that the entropy of the uniform distribution is zero independent of its width.

– Equation 7 is missing some $\tilde z$.

– The operational diagram in Figure 3 is missing a “|”.

---

> ### Author Response · Authors · 2017-12-27
> **Response to AnonReviewer1**
>
> Thank you for the review and suggestions.
>
> > Lossy image compression using neural networks is a rapidly advancing field and of considerable interest to the ICLR community. I like the approach of using a hierarchical entropy model, which may inspire further work in this direction. It is nice to see that the variational approach may be able to outperform the more complicated state-of-the-art approach of Rippel and Bourdev (2017). That said, the evaluation is in terms of MS-SSIM and only the network directly optimized for MS-SSIM outperformed the adversarial approach of R&B. Since the reconstructions generated by a network optimized for MSE tend to look better than those of the MS-SSIM network (Figure 6), I am wondering if the proposed approach is indeed outperforming R&B or just exploiting a weakness of MS-SSIM. It would have been great if the authors included a comparison based on human judgments or at least a side-by-side comparison of reconstructions generated by the two approaches.
>
> Thank you for thinking critically about distortion metrics. This is precisely one of the points we wanted to make with this paper - none of the metrics available today are perfect, and it is easy for ANN-based methods to overfit to whatever metric is used, resulting in good performance numbers but a loss of visual quality. That said, we would like to point out that neither we nor Rippel (2017) provide an evaluation based on human judgements. As such, it is unclear whether the adversarial loss they blend with an MS-SSIM loss is actually helping in terms of visual quality. Unfortunately, we can't systematically compare our method to theirs using human judgements, because they did not make their images available to us.
>
> Regarding MS-SSIM vs. squared loss, we think it depends on the image which one is visually better. Because MS-SSIM has been very popular, we wanted to show an example that is challenging for MS-SSIM. Note that many of the images shown in the appendix (side by side, one optimized for MS-SSIM and one for squared loss) are compressed to roughly similar bit rates, allowing a crude comparison (it is difficult in our current approach to match bit rates exactly). We lowered the bit rate of the images for this revision, to make the differences more visible.
>
> > It might be interesting to relate the entropy model used here to other work involving scale mixtures, e.g. the field of Gaussian scale mixtures (Lyu & Simoncelli, 2007).
>
> Thanks, we included this reference.
>
> > Another interesting comparison might be to other compression approaches where scale mixtures were used and pixels were encoded together with scales (e.g., van den Oord & > Schrauwen, 2017).
>
> We were unable to pinpoint this paper, could you please provide a more detailed reference?
>
> > The authors combine their approach using MS-SSIM as distortion. Is this technically still a VAE? Might be worth discussing.
>
> We don't know, and unfortunately, currently don't have much to say about this point.
>
> > I did not quite follow the motivation for convolving the prior distributions with a uniform distribution.
>
> We tried to improve the explanation in appendix 6.2.
>
> > – On page 3 the paper talks about “the true posterior” of a model which hasn’t been defined yet. Although most readers will not stumble here as they will be familiar with > VAEs, perhaps mention first that the generative model is defined over both $x$ and $\tilde y$.
>
> We hope to have fixed this with the current revision.
>
> > – Below Equation 2 it sounds like the authors claim that the entropy of the uniform distribution is zero independent of its width.
>
> That was not our intention, and it should be fixed now.
>
> > – Equation 7 is missing some $\tilde z$.
>
> Fixed.
>
> > – The operational diagram in Figure 3 is missing a “|”.
>
> Fixed.

---

### Official Review · AnonReviewer3 · 2017-11-27
**In my opinion the work has a good quality to be presented at the ICLR. However, I think it could be excellent if some parts are improved.**

**Rating:** 7
**Confidence:** 5

**Review:**


Authors propose a transform coding solution by extending the work in Balle 2016. They define an hyperprior for the entropy coder to model the spatial relation between the transformed coefficients.

The paper is well written, although I had trouble following some parts. The results of the proposal are state-of-the-art, and there is an extremely exhaustive comparison with many methods.

In my opinion the work has a good quality to be presented at the ICLR. However, I think it could be excellent if some parts are improved. Below I detail some parts that I think could be improved.


*** MAIN ISSUES

I have two main concerns about motivation that are related. The first refers to hyperprior motivation. It is not clear why, if GDN was proposed to eliminate statistical dependencies between pixels in the image, the main motivation is that GDN coefficients are not independent. Perhaps this confusion could be resolved by broadening the explanation in Figure 2. My second concern is that it is not clear why it is better to modify the probability distribution for the entropy encoder than to improve the GDN model. I think this is a very interesting issue, although it may be outside the scope of this work. As far as I know, there is no theoretical solution to find the right balance between the complexity of transformation and the entropy encoder. However, it would be interesting to discuss this as it is the main novelty of the work compared to other methods of image compression based on deep learning.

*** OTHER ISSUES

INTRODUCTION

-"...because our models are optimized end-to-end, they minimize the total expected code length by learning to balance the amount of side information with the expected improvement of the entropy model."
I think this point is very interesting, it would be good to see some numbers of how this happens for the results presented, and also during the training procedure. For example, a simple comparison of the number of bits in the signal and side information depending on the compression rate or the number of iterations during model training.


COMPRESSION WITH VARIATIONAL MODELS

- There is something missing in the sentence: "...such as arithmetic coding () and transmitted..."

- Fig1. To me it is not clear how to read the left hand schemes. Could it be possible to include the distributions specifically? Also it is strange that there is a \tiled{y} in both schemes but with different conditional dependencies. Another thing is that the symbol ψ appears in this figure and is not used in section 2.

- It would be easier to follow if change the symbols of the functions parameters by something like \theta_a and \theta_s.

- "Distortion is the expected difference between..." Why is the "expected" word used here?

- "...and substituting additive uniform noise..." is this phrase correct? Are authors is Balle 2016 substituting additive uniform noise?

- In equation (1), is the first term zero or constant? when talking about equation (7) authors say "Again, the first term is constant,...".

- The sentence "Most previous work assumes..." sounds strange.

- The example in Fig. 2 is extremely important to understand the motivation behind the hyperprior but I think it needs a little more explanation. This example is so important that it may need to be explained at the beginning of the work. Is this a real example, of a model trained without and with normalization? If so specify it please. Why is GDN not able to eliminate these spatial dependencies? Would these dependencies be eliminated if normalization were applied between spatial coefficients? Could you remove dependencies with more layers or different parameters in the GDN?

INTRODUCTION OF A SCALE HYPERPRIOR

- TYPO "...from the center pane of..."

- "...and propose the following extension of the model (figure 3):" there is nothing after the colon. Maybe there is something missing, or maybe it should be a dot instead of a colon. However to me there is a lack of explanation about the model.


RESULTS

- "...,the probability mass functions P_ŷi need to be constructed “on the fly”..."
How computationally costly is this?

- "...batch normalization or learning rate decay were found to have no beneficial effect (this may be due to the local normalization properties of GDN, which contain global normalization as a special case)."

This is extremely interesting. I see the connection for batch normalization, but not for decay of the learning rate. Please, clarify it. Does this mean that when using GDN instead of regular nonlinearity we no longer need to use batch normalization? Or in other words, do you think that batch normalization is useful only because it is special case of GSN? It would be useful for the community to assess what are the benefits of local normalization versus global normalization.

- "...each of these combinations with 8 different values of λ in order to cover a range of rate–distortion tradeoffs."

Would it be possible with your methods including \lambda as an input and the model parameters as side information?

- I guess you included the side information when computing the total entropy (or number of bits), was there a different way of compressing the image and the side information?

- Using the same metrics to train and to evaluate is a little bit misleading. Evaluation plots using a different perceptual metric would be helpful.

-"Since MS-SSIM yields values between 0 (worst) and 1 (best), and most of the compared methods achieve values well above 0.9, we converted the quantity to decibels in order to improve legibility."
Are differences of MS-SSIM with this conversion significant? Is this transformation necessary, I lose the intuition. Besides, probably is my fault but I have not being able to "unconvert" the dB to MS-SSIM units, for instance 20*log10(1)= 20 but most curves surpass this value.

- "..., results differ substantially depending on which distortion metric is used in the loss function during training."
It would be informative to understand how the parameters change depending on the metric employed, or at least get an intuition about which set of parameters adapt more g_a, g_s, h_a and h_s.

- Figs 5, 8 and 9. How are the curves aggregated for different images? Is it the mean for each rate value? Note that depending on how this is done it could be totally misleading.

- It would be nice to include results from other methods (like the BPG and Rippel 2017) to compare with visually.

RELATED WORK

Balle et al. already published a work including a perceptual metric in the end-to-end training procedure, which I think is one of the main contributions of this work. Please include it in related work:

"End-to-end optimization of nonlinear transform codes for perceptual quality." J. Ballé, V. Laparra, and E.P. Simoncelli. PCS: Picture Coding Symposium, (2016)

DISCUSSION

First paragraphs of discussion section look more like a second section of "related work".
I think it is more interesting if the authors discuss the relevance of putting effort into modelling hyperprior or the distribution of images (or transformation). Are these things equivalent? Or is there some reason why we can't include hyperprior modeling in the g_a transformation? For me it is not clear why we should model the distribution of outputs as, in principle, the g_a transformation has to enforce (using the training procedure) that the transformed data follow the imposed distribution. Is it because the GDN is not powerful enough to make the outputs independent? or is it because it is beneficial in compression to divide the problem into two parts?

REFERENCES

- Balle 2016 and Theis 2017 seem to be published in the same conference the same year. Using different years for the references is confusing.

- There is something strange with these references

Ballé, J, V Laparra, and E P Simoncelli (2016). “Density Modeling of Images using a Generalized
Normalization Transformation”. In: Int’l. Conf. on Learning Representations (ICLR2016). URL :
https://arxiv.org/abs/1511.06281.
Ballé, Valero Laparra, and Eero P. Simoncelli (2015). “Density Modeling of Images Using a Gen-
eralized Normalization Transformation”. In: arXiv e-prints. Published as a conference paper at
the 4th International Conference for Learning Representations, San Juan, 2016. arXiv: 1511.
06281.
– (2016). “End-to-end Optimized Image Compression”. In: arXiv e-prints. 5th Int. Conf. for Learn-
ing Representations.

---

> ### Author Response · Authors · 2017-12-27
> **Response to AnonReviewer3, part 1**
>
> Thank you for the review and suggestions.
>
> > I have two main concerns about motivation that are related. The first refers to hyperprior motivation. It is not clear why, if GDN was proposed to eliminate statistical > dependencies between pixels in the image, the main motivation is that GDN coefficients are not independent. Perhaps this confusion could be resolved by broadening the > explanation in Figure 2. My second concern is that it is not clear why it is better to modify the probability distribution for the entropy encoder than to improve the GDN model> . I think this is a very interesting issue, although it may be outside the scope of this work. As far as I know, there is no theoretical solution to find the right balance > between the complexity of transformation and the entropy encoder. However, it would be interesting to discuss this as it is the main novelty of the work compared to other > methods of image compression based on deep learning.
>
> Thank you very much for pointing this out! Our intention was to enable factorization of the latent representation as much as possible. However, the hyperprior models still significantly outperform the factorized prior models. We think of that result as an indication that statistical dependencies in the latent representation, at least for compression models, may actually be desirable. Some of our intuitions were not conveyed well in the original draft. We have rewritten large parts of the paper to make this much clearer. Please refer to the revised discussion, as well as the new section 6.3 in the appendix for details.
>
> >  -"...because our models are optimized end-to-end, they minimize the total expected code length by learning to balance the amount of side information with the expected > improvement of the entropy model."
> > I think this point is very interesting, it would be good to see some numbers of how this happens for the results presented, and also during the training procedure. For > example, a simple comparison of the number of bits in the signal and side information depending on the compression rate or the number of iterations during model training.
>
> We included a new plot about this, and a paragraph describing it, in the experimental results section. Generally, the amount of side information used is very low compared to the total bit rate.
>
> > - There is something missing in the sentence: "...such as arithmetic coding () and transmitted..."
>
> Fixed.
>
> > - Fig1. To me it is not clear how to read the left hand schemes. Could it be possible to include the distributions specifically? Also it is strange that there is a \tiled{y} > in both schemes but with different conditional dependencies. Another thing is that the symbol ψ appears in this figure and is not used in section 2.
>
> The schemes on the left hand are "graphical models" that are quite standard in the literature on Bayesian modeling (for instance, refer to "Pattern Recognition and Machine Learning" by Christopher Bishop). They are not crucial for the understanding of the paper, but might provide a quick overview for someone familiar with them. Unfortunately, we think there isn't enough space in the paper to provide more detail. Regarding the symbol psi, we reordered sections 2 and 3 to address the problem.
>
> > - It would be easier to follow if change the symbols of the functions parameters by something like \theta_a and \theta_s.
>
> We are following an established convention in the VAE literature to name the parameters of the generative model theta and the parameters of the inference model phi. We understand that this may decrease readability for people with other backgrounds, but currently we think this is the best solution.
>
> > - "Distortion is the expected difference between..." Why is the "expected" word used here?
>
> This is meant in the sense of taking the expectation of the difference over the data distribution. We tried to clarify this in the current revision and hope it is clearer now.
>
> > - "...and substituting additive uniform noise..." is this phrase correct? Are authors is Balle 2016 substituting additive uniform noise?
>
> Yes, that is correct.
>
> > - In equation (1), is the first term zero or constant? when talking about equation (7) authors say "Again, the first term is constant,...".
>
> They are zero *and* constant in both cases. We changed the language to be more precise.
>
> > - The sentence "Most previous work assumes..." sounds strange.
>
> We rewrote parts of the paper, which should have fixed this.

---

> ### Author Response · Authors · 2017-12-27
> **Response to AnonReviewer3, part 2**
>
> > - The example in Fig. 2 is extremely important to understand the motivation behind the hyperprior but I think it needs a little more explanation. This example is so important > that it may need to be explained at the beginning of the work. Is this a real example, of a model trained without and with normalization? If so specify it please.
>
> Yes, this is a real example. We made the description more precise, and we hope that our edits to the main text helped to convey the motivation better.
>
> > Why is GDN not able to eliminate these spatial dependencies? Would these dependencies be eliminated if normalization were applied between spatial coefficients? Could you remove dependencies with more layers or different parameters in the GDN?
>
> We think that GDN is capable of removing more dependencies than what we observe remain, but a certain amount of dependency may actually be desirable in the context of rate--distortion optimization. Unfortunately, it's impossible to fully control for all other possible causes of the remaining statistical dependencies, but we are interested in researching this further.
>
> > - TYPO "...from the center pane of..."
>
> Fixed.
>
> > - "...and propose the following extension of the model (figure 3):" there is nothing after the colon. Maybe there is something missing, or maybe it should be a dot instead of > a colon. However to me there is a lack of explanation about the model.
>
> Fixed.
>
> > - "...,the probability mass functions P_ŷi need to be constructed “on the fly”..."
> > How computationally costly is this?
>
> We are investigating this currently. Our implementation at this point is naive, in that it pre-generates the probability tables and fully stores them in memory before doing the arithmetic coding. The memory requirements can be substantial, slowing the process down artificially. A better way would be to inline these computations. We're also working on a method to collect accurate timing data, and will update the paper once we have them.
>
> > - "...batch normalization or learning rate decay were found to have no beneficial effect (this may be due to the local normalization properties of GDN, which contain global > normalization as a special case)."
> > This is extremely interesting. I see the connection for batch normalization, but not for decay of the learning rate. Please, clarify it. Does this mean that when using GDN > instead of regular nonlinearity we no longer need to use batch normalization? Or in other words, do you think that batch normalization is useful only because it is special > case of GSN? It would be useful for the community to assess what are the benefits of local normalization versus global normalization.
>
> We think that GDN has the potential to subsume the effects of batch normalization, as it implements local normalization, which is a generalization of global normalization. The Tensorflow implementation of GDN uses different default constants compared to the one described by Ballé (2017), which we suspect may have something to do with the fact that we couldn't get much gains out of applying a learning rate decay. However, this is speculative, and we are still researching these effects.
>
> > - "...each of these combinations with 8 different values of λ in order to cover a range of rate–distortion tradeoffs."
> > Would it be possible with your methods including \lambda as an input and the model parameters as side information?
>
> Yes, we could treat lambda as side information and have the decoder switch between different sets of model parameters based on that. All that would be required is an encoding scheme for lambda.
>
> > - I guess you included the side information when computing the total entropy (or number of bits), was there a different way of compressing the image and the side information?
>
> Yes, the reported rates are total bit rates for encoding y and z. We included a new figure in the experimental results section to show the fraction of side information compared to total bit rate.
>
> > - Using the same metrics to train and to evaluate is a little bit misleading. Evaluation plots using a different perceptual metric would be helpful.
>
> Why do you think it is misleading to train and evaluate on the same metric, could you elaborate?

---

> ### Author Response · Authors · 2017-12-27
> **Response to AnonReviewer3, part 3**
>
> > -"Since MS-SSIM yields values between 0 (worst) and 1 (best), and most of the compared methods achieve values well above 0.9, we converted the quantity to decibels in order to > improve legibility."
> > Are differences of MS-SSIM with this conversion significant? Is this transformation necessary, I lose the intuition. Besides, probably is my fault but I have not being able to > "unconvert" the dB to MS-SSIM units, for instance 20*log10(1)= 20 but most curves surpass this value.
>
> We updated the paper to include the exact formula we used in the figure caption, thanks for pointing out this oversight. The rationale for using this transformation is that a difference of, say, 0.01 in the 0.99 MS-SSIM range is much more significant than the same difference around a value of 0.91, for example. In a plot, the difference becomes harder and harder to see the closer the values approach 1. The logarithm serves to provide a visually more balanced presentation.
>
> > - "..., results differ substantially depending on which distortion metric is used in the loss function during training."
> > It would be informative to understand how the parameters change depending on the metric employed, or at least get an intuition about which set of parameters adapt more g_a, > g_s, h_a and h_s.
>
> We agree that this would be interesting, but lack a good way of measuring it. We will likely do more research in this direction in the future.
>
> > - Figs 5, 8 and 9. How are the curves aggregated for different images? Is it the mean for each rate value? Note that depending on how this is done it could be totally > misleading.
>
> Thank you for pointing this out. We have updated the paper to use interpolated rate aggregation for the MS-SSIM plots, in order to match Rippel (2017), and to use lambda-aggregation for the PSNR plots, in order to effectively compare to HEVC (the results did not change much). We discuss this in appendix 6.4.
>
> > - It would be nice to include results from other methods (like the BPG and Rippel 2017) to compare with visually.
>
> We agree this would be desirable, but this is limited in practice as Rippel (2017) have not made their reconstructed images available to us. For visual comparisons, we need to match bit rates for the compared image, which is not easy given models trained for a discrete set of lambdas. (Many images provided in the appendix approximately match, but not all of them.) We’ll attempt to prepare a visual comparison to BPG for the final paper, if time permits, or make it available online later.
>
> > Balle et al. already published a work including a perceptual metric in the end-to-end training procedure, which I think is one of the main contributions of this work. Please > include it in related work:
> >
> > "End-to-end optimization of nonlinear transform codes for perceptual quality." J. Ballé, V. Laparra, and E.P. Simoncelli. PCS: Picture Coding Symposium, (2016)
>
> Thanks - we fixed this. Note that their results are quite limited, as they use block transforms which don't adapt to the data as well as deeper models.
>
> > First paragraphs of discussion section look more like a second section of "related work".
> > I think it is more interesting if the authors discuss the relevance of putting effort into modelling hyperprior or the distribution of images (or transformation). Are these things equivalent? Or is there some reason why we can't include hyperprior modeling in the g_a transformation? For me it is not clear why we should model the distribution of outputs as, in principle, the g_a transformation has to enforce (using the training procedure) that the transformed data follow the imposed distribution. Is it because the GDN is not powerful enough to make the outputs independent? or is it because it is beneficial in compression to divide the problem into two parts?
>
> We think that it may be beneficial to divide the problem into two parts, as you say, and that our results provide a bit of evidence regarding that. However, we didn't do a good job of presenting this intuition in the first draft. We hope that the current revision is much clearer.
>
> > - Balle 2016 and Theis 2017 seem to be published in the same conference the same year. Using different years for the references is confusing.
>
> Fixed.
>
> > - There is something strange with these references
> >
> > Ballé, J, V Laparra, and E P Simoncelli (2016). “Density Modeling of Images using a Generalized
> > Normalization Transformation”. In: Int’l. Conf. on Learning Representations (ICLR2016). URL :
> > https://arxiv.org/abs/1511.06281.
> > Ballé, Valero Laparra, and Eero P. Simoncelli (2015). “Density Modeling of Images Using a Gen-
> > eralized Normalization Transformation”. In: arXiv e-prints. Published as a conference paper at
> > the 4th International Conference for Learning Representations, San Juan, 2016. arXiv: 1511.
> > 06281.
> > – (2016). “End-to-end Optimized Image Compression”. In: arXiv e-prints. 5th Int. Conf. for Learn-
> > ing Representations.
>
> Fixed.

---

### Official Review · AnonReviewer2 · 2017-11-28
**Variational image compression with a scale hyperprior**

**Rating:** 7
**Confidence:** 5

**Review:**

The paper is a step forward for image deep compression, at least when departing from the (Balle et al., 2017) scheme.
The proposed hyperpriors are especially useful for medium to high bpp and optimized for L2/ PSNR evaluation.

I find the description of the maths too laconic and hard to follow. For example, what’s the U(.|.) operator in (5)?

What’s the motivation of using GDN as non linearity instead of e.g. ReLU?

I am not getting the need of MSSSIM (dB).  How exactly was it defined/computed?

Importance of training data? The proposed models are trained on 1million images while others like (Theis et al, 2017) and [Ref1,Ref2] use smaller datasets for training.

I am missing a discussion about Runtime / complexity vs. other approaches?

Why MSSSIM is a relevant measure? The Fig. 6 seem to show better visual results for L2 loss (PSNR) than when optimized for MSSSIM, at least in my opinion.

What's the reason to use 4:4:4 for BPG and 4:2:0 for JPEG?

What is the relation between hyperprior and importance maps / content-weights [Ref1] ?

What about reproducibility of the results? Will be the codes/models made publicly available?

Relevant literature:
[Ref1] Learning Convolutional Networks for Content-weighted Image Compression (https://arxiv.org/abs/1703.10553)
[Ref2] Soft-to-Hard Vector Quantization for End-to-End Learned Compression of Images and Neural Networks  (https://arxiv.org/abs/1704.00648)

---

> ### Author Response · Authors · 2017-12-27
> **Response to AnonReviewer2**
>
> Thank you for the review and suggestions.
>
> > I find the description of the maths too laconic and hard to follow. For example, what’s the U(.|.) operator in (5)?
>
> We have completely rewritten some sections of the paper in order to improve clarity. U(.|.) indicates a uniform distribution (this is stated in the text). We hope that the paper is now easier to follow. Please let us know if there are any other (or new) parts which you find hard to read, we are happy to make further improvements.
>
> > What’s the motivation of using GDN as non linearity instead of e.g. ReLU?
>
> We have found that GDN nonlinearities, while keeping all other architecture parameters constant, provides significantly better performance than ReLU in g_a and g_s. We haven't done any systematic experiments regarding nonlinearities used in h_a and h_s, and went with ReLU as a "default" choice (note that the amount of side information overall is very small, so we might not benefit much by optimizing this part of the model).
>
> > I am not getting the need of MSSSIM (dB).  How exactly was it defined/computed?
>
> MS-SSIM is defined in Wang, Simoncelli, et al. (2003). It is one of the most widely used perceptual image quality metrics. Thank you for pointing out that we didn’t define how we converted to decibels. We included the exact definition in the figure caption.
>
> > Importance of training data? The proposed models are trained on 1million images while others like (Theis et al, 2017) and [Ref1,Ref2] use smaller datasets for training.
>
> We think this can likely be ruled out as a source of performance gains. Compared to the factorized-prior model in Ballé (2017), which was trained on ~7000 images and squared error, our squared-error factorized-prior model matches its performance on PSNR (figure 10) and even underperforms on MS-SSIM (figure 11).
>
> > I am missing a discussion about Runtime / complexity vs. other approaches?
>
> Our main goal here was to optimize for compression performance, and to control for the effect of capacity limitations of the model (as a result of fewer filters), which may cause unnecessary statistical dependencies in the representation. We realize this intention wasn't sufficiently clear in our first draft, which has lead to some confusion. We have rewritten parts of the paper, added a paragraph to the discussion, and added a supporting section in the appendix (6.3) to clarify.
>
> Comparing the runtime of the encoding and decoding process is important when evaluating compression methods for deployment. To make a fair comparison, all of the components involved must be appropriately optimized, which has not been a priority in our research so far. In particular, we have only implemented the arithmetic coder in a naive way, writing a very large probability table in memory, which is simple to implement, but unnecessarily slows down the computation. An optimized implementation would inline the computation of the probability tables in eq. (11). Some idea of complexity can be gathered from the architecture. Unfortunately, we omitted the number of filters in the transforms, which was an oversight. We now state this in the caption of the figure showing the architecture.
>
> We are working on improving our methods to make accurate runtime measurements, and will be happy to provide them in the final paper or here, as soon as we have them. To give you an estimate for the current implementation: the factorized-prior model, which does not suffer from the naive implementation mentioned above, can encode one of the Kodak images in ~70ms (corresponding to a throughput of ~5.5 megapixels per second). We estimate the hyperprior model to require a longer runtime, mostly due to h_a and h_s. Note that due to further subsampling, however, the complexity of h_a and h_s should be significantly lower than g_a and g_s.

---

> ### Author Response · Authors · 2017-12-27
> **Response to AnonReviewer2, part 2**
>
>
> > Why MSSSIM is a relevant measure? The Fig. 6 seem to show better visual results for L2 loss (PSNR) than when optimized for MSSSIM, at least in my opinion.
>
> MS-SSIM is a widely used image quality index, and has been popular in previous papers presenting ANN-based compression methods. We wanted to understand how visual quality differs when optimizing for different metrics. The image we show here represents a challenge for MS-SSIM, which we felt was important to talk about given how popular the metric is. Other images, such as Kodak 15, tend to be more challenging for squared-error optimized models. Note that in this revision of the paper, we lowered the bit rates of the example images in the appendix, to make artifacts more visible and to demonstrate this effect more clearly across a range of different images.
>
> > What's the reason to use 4:4:4 for BPG and 4:2:0 for JPEG?
>
> Because we optimized our models for squared error in the RGB representation (rather than a luma--chroma colorspace), BPG 4:4:4 is the appropriate method to compare to, as it optimizes the same metric. With respect to JPEG, the 4:4:4 format is not widely used, and we found that it also appears to perform much worse than 4:2:0 (indicating it may not have been optimized as well).
>
> > What is the relation between hyperprior and importance maps / content-weights [Ref1] ?
>
> The importance maps of Li et al. (2017) are primarily designed to provide an embedded code (i.e., a compressed representation of the image which allows accessing lower-quality versions of the image by decoding only a part of the bitstream). To do this, they employ binary quantization rather than integer (i.e. multi-level) quantization, among other techniques. Their entropy model corresponds to a Markov-style prior, similar to the ones used in Johnston et al. (2017) and Rippel et al. (2017).
>
> > What about reproducibility of the results? Will be the codes/models made publicly available?
>
> We are striving to publish at least the full results and parts of the code/model parameters, but due to possible legal constraints, we cannot make any promises at this point. We hope that the description in the paper is self-contained and detailed enough to be useful. We're also happy to answer any further questions.

---

### Public Comment · (anonymous) · 2017-11-24
**No mention at all of runtime**

The authors compare their approach to a number of existing compression algorithms. However, for a fair comparison to these, the authors must calibrate for the same runtime.

As it currently stands, it is impossible to disambiguate the factors driving the approach: is it in fact a better architecture for compression, or simply a large number of filters per layer? In the paper, it is mentioned that "N filters" are used per layer, but N is not mentioned anywhere: what is N? What is the runtime of the algorithm? What is the number of multiplications per pixel?

In compression, speed is critical to ensure viability. Based on my understanding, this is a constraint that many of the approaches compared against have been taking into consideration. As such, for an appropriate comparison speed must be taken into account.

---

> ### Author Response · Authors · 2017-12-27
> **Response to anonymous comment**
>
> Thank you for your comments, and thank you for bringing our oversight of defining N to our attention, which we corrected.
>
> We have not yet optimized our compression method for runtime. However, we have taken care to match model capacities between the factorized-prior model and the hyperprior model, and not to choose a number of filters that would limit transform capacity (see the new section 6.3 in the appendix for details), which would limit the ability of the models to factorize the representation. Runtime is only a very crude proxy for model capacity, and as such, we agree that calibrating for runtime is useful in a deployment context, but not necessarily for establishing whether powerful priors, that are trained end-to-end, are a good thing or not. This was one of the main intuitions driving this paper, and we hope that the presentation of this paper is much clearer in the revision.
>
> Although not the main focus of our paper, we do think it would be useful to have runtime comparisons to other methods, and we are working on a method to get accurate measurements (despite all the difficulties associated with runtime measurements across different hardware). We hope to provide these in another revision. To give you an estimate for the current implementation: the factorized-prior model, which does not suffer from the naive entropy coding implementation mentioned above, can encode one of the Kodak images in ~70ms (corresponding to a throughput of ~5.5 megapixels per second). We estimate the hyperprior model to require a longer runtime, mostly due to h_a and h_s. Note that due to further subsampling, however, the complexity of h_a and h_s should be significantly lower than g_a and g_s.

---

### Public Comment · (anonymous) · 2017-12-27
**Updated revision of our paper**

Dear commenters and reviewers,

thank you for your detailed critique of our paper. We have worked hard to revise our paper and address all of the points you have raised. Unfortunately we cannot currently upload an official revision. There seems to be some contradicting information whether revisions are allowed during the review process (http://www.iclr.cc/doku.php?id=iclr2018:conference_cfp indicates yes, https://openreview.net/group?id=ICLR.cc/2018/Conference indicates no). We have worked under the assumption that they are, and we think that sharing our revision is crucial to provide you with more data that helps us make our points. Therefore, we have shared our revision temporarily (and anonymously) at https://drive.google.com/file/d/1-gP0iFJtgqZ-DIm4kkOYKKz5mpCbm4uD/view.  We are contacting the organizers about this issue, and will update the official revision as soon as we are able to.

Thank you - the authors.

---

> ### Author Response · Authors · 2017-12-27
> **Update**
>
> Hello,
>
> it seems that our inability to upload a revision was a glitch in the website. Once we sent our comments, a button labeled "modifiable original" appeared, which we used to officially upload the revision.
>
> We confirmed that we are getting the updated version when clicking on the PDF symbol. However, it doesn't seem to appear in the revision history. We are unsure what the reason for this is.
>
> We are leaving the Google Drive version online for now, so you can confirm that it is the same version as provided by OpenReview.
>
> - the authors

---

### Author Response · Authors · 2018-04-30
**Updated revision**

Hello everyone,

as requested, we updated the appendix of the paper with a table of runtime measurements, additional plots of density model fits, and additional results on the Tecnick dataset.

Thank you, and see you at the conference!

---

### Decision · Program_Chairs · 2018-01-29
**ICLR 2018 Conference Acceptance Decision**

**Decision:**

Accept (Poster)

**Comment:**

Thank you for submitting you paper to ICLR. The reviewers and authors have engaged well and the revision has improved the paper. The reviewers are all in agreement that the paper substantially expands the prior work in this area,  e.g. by Balle et al. (2016, 2017), and is therefore suitable for publication. Although I understand that the authors have not optimised their compression method for runtime yet, a comment about this prospect in the main text would be a sensible addition.